# On Measuring Excess Capacity in Neural Networks

**Florian Graf**
University of Salzburg
florian.graf@plus.ac.at

**Sebastian Zeng**
University of Salzburg
sebastian.zeng@plus.ac.at

**Bastian Rieck**
Institute for AI and Health
Helmholtz Munich
bastian@rieck.me

**Marc Niethammer**
UNC Chapel Hill
mn@cs.unc.edu

**Roland Kwitt**
University of Salzburg
roland.kwitt@plus.ac.at

## Abstract

We study the *excess capacity* of deep networks in the context of supervised classification. That is, given a capacity measure of the underlying hypothesis class – in our case, empirical Rademacher complexity – to what extent can we (a priori) constrain this class while retaining an empirical error on a par with the unconstrained regime? To assess excess capacity in modern architectures (such as *residual networks*), we extend and unify prior Rademacher complexity bounds to accommodate function *composition* and *addition*, as well as the structure of convolutions. The capacity-driving terms in our bounds are the Lipschitz constants of the layers and an $(2, 1)$ group norm distance to the initializations of the convolution weights. Experiments on benchmark datasets of varying task difficulty indicate that (1) there is a substantial amount of excess capacity per task, and (2) capacity can be kept at a surprisingly similar level across tasks. Overall, this suggests a notion of *compressibility* with respect to weight norms, complementary to classic compression via weight pruning. Source code is available at https://github.com/rkwitt/excess_capacity.

## 1   Introduction

Understanding the generalization behavior of deep networks in supervised classification is still a largely open problem, despite a long history of theoretical advances. The observation that (over-parametrized) models can easily fit—i.e., reach zero training error—to randomly permuted training labels [45, 46] but, when trained on unpermuted labels, yield good generalization performance, has fueled much of the progress in this area. Recent works range from relating generalization to weight norms [3, 14, 26, 37, 38], measures of the distance to initializations [34], implicit regularization induced by the optimization algorithm [8, 41], or model compression [2, 6, 42]. Other works study connections to optimal transport [9], or generalization in the neural tangent kernel setting [1, 23].

When seeking to establish generalization guarantees within the classic uniform convergence regime, bounding a capacity measure, such as the Rademacher complexity [4], of the hypothesis class is the crucial step. While the resultant generalization bounds are typically vacuous and can exhibit concerning behavior [36], the *capacity bounds* themselves offer invaluable insights through the behavior of the bound-driving quantities, such as various types of weight norms or Lipschitz constants.

Particularly relevant to our work is the observation that the bound-driving quantities tend to increase with *task difficulty*. Fig. 1 illustrates this behavior in terms of Lipschitz constants per layer and the distance of each layer's weight to its initialization (measured via a group norm we develop in Section 3.1).

36th Conference on Neural Information Processing Systems (NeurIPS 2022).

This raises two immediate questions: First (**Q1**), can a network maintain empirical testing performance at a substantially lower capacity? Second (**Q2**), is the level of this lowered capacity inevitably tied to task difficulty, i.e., would a reduced-capacity model for an easy task fail on a difficult task? Both of these questions aim at the amount of "unneeded" capacity, which we refer to as *excess capacity* in the remainder of this work.

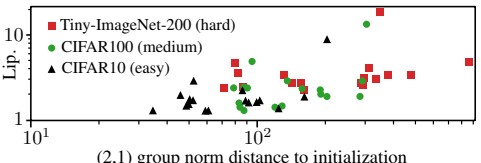

**Figure 1:** Layerwise Lipschitz constants and distance to initialization of a ResNet18 model (see Section 4), trained on different datasets.

We will address questions (**Q1**) and (**Q2**) by means of controlling the empirical Rademacher complexity of a neural network. To this end, we consolidate and extend prior results from the literature on Rademacher complexity bounds to accommodate a broad range of network components in a unified way, including convolutions and skip connections, two ubiquitous elements in state-of-the-art models.

Our **contributions** can be summarized as follows:

1. We establish two bounds (in Theorem 3.5) for the empirical Rademacher complexity of neural networks that use *convolutions* and implement functions built from *composition* and *addition*. Specifically, we introduce two novel, convolution-specific, single-layer covering number bounds in Section 3.2 and contrast them to prior art, then modularize the single-layer to multi-layer covering approach of Bartlett et al. [3] in Section 3.3, and eventually present one incarnation of our framework for convolutional residual networks in Sections 3.4 and 3.5.

2. We present an extensive set of experiments (in Section 4) with a ResNet18 model across benchmark datasets of varying task difficulty, demonstrating that model capacity, when measured via our weight norm based bound, (1) can be kept surprisingly small per task, and (2) can be kept at *roughly the same level* regardless of task difficulty. Both observations suggest *compressibility* of neural networks with respect to weight norms, complementary to the well-known compressibility property of neural networks with respect to the number of parameters [2, 42].

## 2 Related Work

Many prior works establish uniform-convergence type generalization bounds for neural networks through Rademacher complexity analysis. We review such approaches, highlighting challenges that arise with modern network architectures and the peculiarities of convolutional layers.

One direct approach to bound the empirical Rademacher complexity is via a *layer-peeling* strategy [14, 38, 44] where the Rademacher complexity of $L$-layer networks is expressed by a factor times the Rademacher complexity of $(L-1)$-layer networks; in other words, the last layer is peeled off. This factor is typically a matrix $(p, q)$ group norm, and thus the bounds usually scale with the product of the latter. Notably, the nonlinearities need to be *element-wise* operations, and some approaches only work for specific nonlinearities, such as ReLUs or asymmetric activations. A second strategy is to bound the empirical Rademacher complexity via a *covering numbers* approach [3, 26, 29, 47], typically achieved via Dudley's entropy integral [11]. This strategy is particularly flexible as it allows for arbitrary (but fixed) nonlinearities and various paths to bounding covering numbers of network parts, e.g., via Maurey's sparsification lemma or via parameter counting. The corresponding whole-network bounds typically scale with the product of each layer's Lipschitz constant or local empirical estimates thereof [43]. Irrespective of the particular proof strategy, most formal arguments only hold for neural networks constructed from function composition, i.e., maps of the form

$$x \mapsto \sigma_L(A_L \sigma_{L-1}(A_{L-1} \ldots \sigma_1(A_1 x) \ldots)) \ , \tag{1}$$

where $\sigma_i : \mathbb{R}^{d_{i-1}} \to \mathbb{R}^{d_i}$ are nonlinearities and $A_i$ are weight matrices specifying the $i$-th linear map. However, modern architectures often rely on operations specifically tailored to the data, such as convolutions, and typically incorporate skip connections as in residual networks [20], rendering many results inapplicable or suboptimal for such models. In this work, we handle convolutions and skip connections, thus increasing the applicability and utility of such bounds.

For example, while residual networks have been studied extensively, theory mostly focuses on expressivity or optimization aspects [3, 18, 28, 44]. Yun et al. [44] provide a Rademacher complexity bound via layer-peeling for fully-connected layers and element-wise activations. He et al. [19]

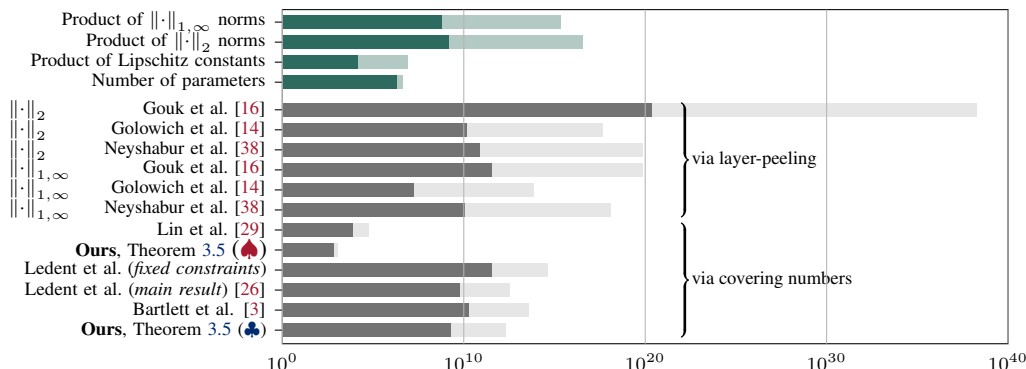

**Figure 2:** Empirical Rademacher complexity bounds (grouped by proof strategy; lower is better), for a 6- (■■) and an 11-layer (■ ■) convolutional network, trained on CIFAR10. Bounds are listed in Section A.2 and quantities that typically appear in these bounds are highlighted in **green** (top part of figure) for reference.

establish a generalization bound for residual networks via covering number arguments, resting upon earlier work by Bartlett et al. [3] for linear maps. However, when directly applied to convolutions, both bounds scale unfavorably w.r.t. the spatial input size (see Section 3.2). Other works provide generalization guarantees *specifically* tailored to convolutional networks, cf. [16, 26, 29, 31], and, although such bounds scale benignly with input size, they only apply for models as in Eq. (1).

**An initial numerical comparison.** Bounds on the empirical Rademacher complexity differ in their dependence on various quantities, such as matrix $(p, q)$ group norms, Lipschitz constants, or the number of parameters. Thus, a precise formal comparison is challenging and, depending on the setting, different bounds may be preferable. To provide some intuition about magnitude differences, we evaluated several existing bounds (including ours from Section 3.5) on two convolutional (ReLU) networks with 6 and 11 layers, see Fig. 2 and Section A.2 for details.

## 3 Rademacher Complexity Analysis

To derive bounds on the empirical Rademacher complexity, we follow the margin-based multiclass learning formalism and the flexible proof strategy of Bartlett et al. [3]. Section 3.2 introduces novel single-layer covering number bounds for convolutions. Section 3.3 modularizes and extends the single- to multi-layer covering step to account for architectures such as residual networks (Section 3.4). Last, Section 3.5 presents and discusses our Rademacher complexity bounds.

### 3.1 Preliminaries

In a $\kappa$-class classification task, we are given $n$ instance/label pairs $S = ((x_1, y_1), \ldots, (x_n, y_n))$, drawn iid from a probability measure $\mathcal{D}$ on $\mathbb{R}^d \times \{1, \ldots, \kappa\}$. For a neural network $f$ in a hypothesis class $\mathcal{F} \subset \{f : \mathbb{R}^d \to \mathbb{R}^\kappa\}$, a class label for input $x$ is obtained by taking the argmax over the components of $f(x) \in \mathbb{R}^\kappa$. The *margin operator* $\mathcal{M} : \mathbb{R}^\kappa \times \{1, \ldots, \kappa\} \to \mathbb{R}, (v, y) \mapsto v_y - \max_{i \neq y} v_i$ leads, with margin $\gamma > 0$, to the *ramp loss* $\ell_\gamma$ and the *empirical ramp risk* $\hat{R}_\gamma$, defined as

$$\ell_\gamma : \mathbb{R} \to \mathbb{R}^+, r \mapsto (1 + r/\gamma)\mathbb{1}_{r \in [-\gamma, 0]} + \mathbb{1}_{r > 0} \quad \text{and} \quad \hat{R}_\gamma(f) = \frac{1}{n} \sum_{i=1}^n \ell_\gamma(-\mathcal{M}(f(x_i), y_i)) \ . \quad (2)$$

To derive generalization bounds via classical Rademacher complexity analysis [33], without having to resort to vector-contraction inequalities [32], we consider the hypothesis class

$$\mathcal{F}_\gamma = \{(x, y) \mapsto \ell_\gamma(-\mathcal{M}(f(x), y)) : f \in \mathcal{F}\} \ . \quad (3)$$

Then, defining the *empirical Rademacher complexity* of any class $\mathcal{H}$ of real-valued functions as

$$\hat{\mathfrak{R}}_S(\mathcal{H}) = \mathbb{E}_\sigma \left[ \sup_{h \in \mathcal{H}} \frac{1}{n} \sum_{i=1}^n \sigma_i h(x_i, y_i) \right] \ , \quad (4)$$

with iid Rademacher variables $\sigma = (\sigma_1, \ldots, \sigma_n)$ from a uniform distribution on $\{\pm 1\}$, facilitates to study $\mathcal{F}_\gamma$ via Eq. (4). The following lemma [3, Lemma 3.1] establishes the link to a margin-based multiclass generalization bound for any $f \in \mathcal{F}$.

**Lemma 3.1.** *Given a hypothesis class $\mathcal{F}$ of functions $f : \mathbb{R}^d \to \mathbb{R}^\kappa$ and a margin parameter $\gamma > 0$, then, with probability of at least $1 - \delta$ over the choice of $S \sim \mathcal{D}^n$, for any $f \in \mathcal{F}$, it holds that*

$$\mathbb{P}[\arg\max_{i \in \{1,\ldots,\kappa\}} f(x)_i \neq y] \leq \hat{R}_\gamma(f) + 2\hat{\mathfrak{R}}_S(\mathcal{F}_\gamma) + 3\sqrt{\frac{\log\left(\frac{2}{\delta}\right)}{2n}} \quad . \tag{5}$$

To obtain a computable expression for the right-hand side of Lemma 3.1, we seek a bound on $\hat{\mathfrak{R}}_S(\mathcal{F}_\gamma)$ tied to some measurable quantities of the network realizing $\mathcal{F}$. For our purposes, the relationship of $\hat{\mathfrak{R}}_S(\mathcal{F}_\gamma)$ and the covering number of $\mathcal{F}_\gamma$ turns out to be a flexible approach. In general, given a normed space $(\mathcal{G}, \|\cdot\|_\mathcal{G})$, the *covering number* $\mathcal{N}(\mathcal{G}, \epsilon, \|\cdot\|_\mathcal{G})$ is the cardinality of the smallest $\epsilon$-cover of $\mathcal{G}$, i.e., of the smallest subset $\mathcal{U} \subset \mathcal{G}$ such that, for any $g \in \mathcal{G}$, there exists $u \in \mathcal{U}$ with $\|g - u\|_\mathcal{G} \leq \epsilon$. In our setting, $\mathcal{G}$ is a class of functions $g : (\mathcal{X}, \|\cdot\|_\mathcal{X}) \to (\mathcal{Y}, \|\cdot\|_\mathcal{Y})$ between normed spaces $\mathcal{X}$ and $\mathcal{Y}$. Given data $X = (x_1, \ldots, x_n) \in \mathcal{X}^n$, we define a *data-dependent norm* on $\mathcal{G}|_X$ as

$$\|g\|_X = \sqrt{\sum_i \|g(x_i)\|_\mathcal{Y}^2} \quad . \tag{6}$$

In other words, Eq. (6) is the $l_2$ norm on the restriction of $\mathcal{G}$ to $X$. Specifically, we seek to bound $\log \mathcal{N}(\mathcal{F}_\gamma, \epsilon, \|\cdot\|_S)$, as this facilitates to control the empirical Rademacher complexity of $\mathcal{F}_\gamma$ by means of Dudley's entropy integral. Typically, such covering number bounds depend on the norm of the data itself, i.e., $\|X\| = \sqrt{\|x_1\|_\mathcal{X}^2 + \cdots \|x_n\|_\mathcal{X}^2}$.

## 3.2 Covering number bounds for convolutions

We consider 2D convolutions, acting on images with $c_{in}$ channels of width $w$ and height $h$, i.e., $x \in \mathbb{R}^{c_{in} \times h \times w}$. For readability only, we discuss convolutions of stride 1 and input-size preserving padding; this is not an assumption required for Theorem 3.2. Formally, a convolutional layer is a linear map $\phi_K : \mathbb{R}^{c_{in} \times h \times w} \to \mathbb{R}^{c_{out} \times h \times w}$ (as we omit bias parameters), where $c_{in}$ and $c_{out}$ denote the number of input and output channels. The map is parametrized by a tensor $K \in \mathbb{R}^{c_{out} \times c_{in} \times k_h \times k_w}$ of spatial extension/kernel size $(k_h, k_w)$. Since convolutions are linear maps, they can be specified by matrices which act on the (reshaped) inputs and one could invoke existing covering number bounds. However, this is suboptimal, as any structure specific to convolutions is ignored. In particular, norm-based generalization bounds agnostic to this structure incur unfavorable scaling behavior w.r.t. the dimensionality of the input. To be more specific, the weight tensor $K$ of a convolutional layer does not directly specify the corresponding matrix; instead, it parametrizes $c_{out}$ filters, i.e., local linear maps, which are applied to the $(c_{in} \times k_h \times k_w)$-sized pixel neighborhoods of the input. Hence, the matrix $M_K$ corresponding to the *global* linear map consists of many copies of the elements of this tensor, one for each of the $hw$ patches the filters are applied to. Thus, the $l_p$ norm of the matrix $M_K$ is $\|M_K\|_p = (hw)^{1/p} \|K\|_p$ (see Section A.1). We mitigate this scaling issue by tying the covering number of a convolutional layer to a variant of the (2,1) group norm on the *tensor* $K$ itself. We define this norm as the sum over the $l_2$ norms taken along the *input channels* of $K$, i.e.,

$$\|K\|_{2,1} = \sum_{ikl} \|K_{i\cdot kl}\|_2 = \sum_{ikl} \sqrt{\sum_j K_{ijkl}^2} \quad . \tag{7}$$

For the special case of inputs of size $(h, w) = (1, 1)$ and kernel size $(k_h, k_w) = (1, 1)$, convolution is just matrix multiplication along the channels. In this case, $M_K = K_{\cdot\cdot 11}$ and our norm from Eq. (7) agrees with the standard $(2, 1)$ group norm on $M_K^\top$, i.e., $\|K\|_{2,1} = \|M_K^\top\|_{2,1}$. Theorem 3.2 establishes two covering number bounds for convolutions.

**Theorem 3.2.** *Let $b > 0$ and $\mathcal{F} = \{\phi_K | K \in \mathbb{R}^{c_{out} \times c_{in} \times k_h \times k_w}, \|K\|_{2,1} \leq b\}$ denote the class of 2D convolutions with $c_{in}$ input channels, $c_{out}$ output channels and kernel size $k_h \times k_w$, parametrized by tensors $K$ with $W = c_{out} c_{in} k_h k_w$ parameters. Then, for any $X \in \mathbb{R}^{n \times c_{in} \times h \times w}$ and covering radius $\epsilon > 0$,*

$$\log \mathcal{N}(\mathcal{F}, \epsilon, \|\cdot\|_X) \leq \begin{cases} \left\lceil \frac{\|X\|^2 b^2}{\epsilon^2} \right\rceil \log(2W) & \text{(8a)} \\[3mm] 2W \log\left(1 + \left\lceil \frac{\|X\|^2 b^2}{\epsilon^2} \right\rceil\right) & \text{. (8b)} \end{cases}$$

Eq. (8a) is analogous to the single-layer bound of Bartlett et al. [3, Lemma 3.2] for fully-connected layers, but replaces the $(2,1)$ group norm constraint on matrices $M^\top$ with a constraint on tensors $K$. This is tighter than invoking [3, Lemma 3.2] directly on $M_K^\top \in \mathbb{R}^{c_{in}hw \times c_{out}hw}$, as $K$ has only $c_{out}c_{in}k_hk_w$ parameters and $\left\|M_K^\top\right\|_{2,1} \geq {hw}/{\sqrt{k_hk_w}}\|K\|_{2,1}$, see Section A.1. A thorough comparison between the two bounds in Theorem 3.2 is nuanced, though, as preferring one over the other depends on the ratio between the number of parameters $W$ and $\|X\|^2 b^2/\epsilon^2$. The latter, in turn, requires to consider all covering radii $\epsilon$. Hence, we defer this discussion to Section 3.5, where differences manifest more clearly in the overall empirical Rademacher complexity bounds.

**Proof sketch.** The statement of Theorem 3.2 follows from an application of Maurey's sparsification lemma, which guarantees the existence of an $\epsilon$-cover of $\mathcal{F}$ (of known cardinality) if there is a finite subset $\{V_1, \ldots, V_d\} \subset \mathcal{F}$ s.t. every $f \in \mathcal{F}$ is a convex combination of the $V_i$. We show that one can find such a finite subset of cardinality $d = 2c_{in}c_{out}k_hk_w = 2W$. The cardinality of the cover is then determined by a *combinatorial quantity* which additionally depends on $\|X\|$ and the norm constraint $b$. Bounding this quantity, i.e., a binomial coefficient, in two different ways, establishes the bounds.

**Relation to prior work.** Closely related is recent work by Ledent et al. [26] who derive $l_\infty$ covering number bounds for convolutional layers based on a classic result by Zhang [47]. Similar to Eq. (8a), their bound depends on the square of a weight norm directly on the tensor $K$, the square of a data norm, as well as a logarithmic term. The data norm is the maximal $l_2$ norm of a single patch. Compared to our result, this implicitly removes a factor of the spatial dimension $hw$. However, when transitioning to multi-layer bounds, this factor reenters in the form of the spatial dimension of the output (after subsequent pooling) via the Lipschitz constant. Overall, the quadratic terms across both results scale similarly (with our data norm being less sensitive to outliers), but we improve on Ledent et al. [26] in the logarithmic term. By contrast, the use of $l_\infty$ covers in [26] yields whole-network bounds with improved dependency on the number of classes; see Section A.6 for an in-depth comparison. In other related work, Lin et al. [29] derive an $l_2$ covering number bound for convolutional layers similar to Eq. (8b), which depends linearly on the number of parameters and logarithmically on norms. In their proof, Lin et al. [29] show that every cover of a convolutional layer's weight space (a subset of a Euclidean space) induces a cover of the corresponding function space w.r.t. the data dependent norm defined in Eq. (6). However, their approach incurs an additional factor inside the logarithm that corresponds to the number of how often each filter is applied, i.e., the spatial dimension of the output. Importantly, non-convolution specific approaches can equally mitigate undesirable scaling issues, e.g., by utilizing $(1, \infty)$ group norms on the matrices representing the linear map [14, 16, 38]; as differences primarily manifest in the resulting bounds on the Rademacher complexity, we refer to our discussion in Section 3.5.

## 3.3 Covering number bounds for composition & addition

As many neural networks are built from composition and summation of layers, we study covering numbers under these operations. The key building blocks are the following, easy to verify, inequalities.

**Lemma 3.3.** *Let $\mathcal{F}_1, \mathcal{F}_2$ be classes of functions on normed spaces $(\mathcal{X}, \|\cdot\|_{\mathcal{X}}) \to (\mathcal{Y}, \|\cdot\|_{\mathcal{Y}})$ and let $\mathcal{G}$ be a class of $c$-Lipschitz functions $(\mathcal{Y}, \|\cdot\|_{\mathcal{Y}}) \to (\mathcal{Z}, \|\cdot\|_{\mathcal{Z}})$. Then, for any $X \in \mathcal{X}^n$ and $\epsilon_{\mathcal{F}_1}$, $\epsilon_{\mathcal{F}_2}, \epsilon_{\mathcal{G}} > 0$, it holds that*

$$\mathcal{N}(\{f_1 + f_2 \mid f_1 \in \mathcal{F}_1, f_2 \in \mathcal{F}_2\}, \epsilon_{\mathcal{F}_1} + \epsilon_{\mathcal{F}_2}, \|\cdot\|_X) \leq \mathcal{N}(\mathcal{F}_1, \epsilon_{\mathcal{F}_1}, \|\cdot\|_X)\mathcal{N}(\mathcal{F}_2, \epsilon_{\mathcal{F}_2}, \|\cdot\|_X) \quad (9)$$

*and*

$$\mathcal{N}(\{g \circ f \mid g \in \mathcal{G}, f \in \mathcal{F}_2\}, \epsilon_{\mathcal{G}} + c\epsilon_{\mathcal{F}_2}, \|\cdot\|_X) \leq \mathcal{N}(\mathcal{F}_2, \epsilon_{\mathcal{F}_2}, \|\cdot\|_X) \sup_{f \in \mathcal{F}_2} \mathcal{N}(\mathcal{G}, \epsilon_{\mathcal{G}}, \|\cdot\|_{f(X)}) . \quad (10)$$

To establish these inequalities, one chooses minimal covers of the original function spaces and links their elements via the considered operation, i.e., addition or composition. The resulting functions correspond to tuples of elements of the original covers. Hence, the right-hand side of the inequalities is a product of covering numbers. The crucial step is to determine a preferably small radius $\epsilon$ such that these functions form an $\epsilon$-cover. In Lemma 3.3, this is achieved via standard properties of norms. Notably, iterative application of Lemma 3.3 allows bounding the covering numbers of any function class built from compositions and additions of simpler classes.

In Section C.3, we apply Lemma 3.3 on two examples, i.e., (1) $f \in \mathcal{F} = \{f_L \circ \cdots \circ f_1\}$ and (2) $h \in \mathcal{H} = \{g + h_L \circ \cdots \circ h_1\}$. Instantiating the first example for $f_i = \sigma_i \circ \phi_i$, with $\sigma_i$ fixed and $\phi_i$ from a family of linear maps, yields covering number bounds for networks as in Eq. (1). As the second example corresponds to residual blocks (with $g$ possibly the identity map), the combination of (1) and (2) yields covering number bounds for residual networks; see Example C.3.

Overall, this strategy not only allows to derive covering number bounds for a broad range of architectures, but also facilitates integrating linkings between function spaces in a modular way. For instance, Lemma C.14 provides a variant of Lemma 3.3 for *concatenation*, used in DenseNets [22].

**Relation to prior work.** He et al. [19] investigate covering number bounds for function spaces as considered above. They present covering number bounds for residual networks and show that the covering number $\mathcal{N}(\mathcal{F}, \epsilon, \|\cdot\|_X)$ of such models with layers $\mathcal{F}_\alpha$ is bounded by the product $\prod_\alpha \sup_{\phi \in \mathcal{G}_\alpha} \mathcal{N}(\mathcal{F}_\alpha, \epsilon_{\mathcal{F}_\alpha}, \|\cdot\|_{\phi(X)})$ for appropriately defined function spaces $\mathcal{G}_\alpha$. Yet, the dependency of the whole-network covering radius $\epsilon$ on the single-layer covering radii $\epsilon_{\mathcal{F}_\alpha}$ is *only* derived for a very specific residual network. Our addition to the theory is a more modular and structured way of approaching the problem, which we believe to be valuable on its own.

### 3.4 Covering number bounds for residual networks

We next state our whole-network covering number bounds for residual networks and then present the corresponding bounds on the empirical Rademacher complexity in Section 3.5. Accompanying generalization guarantees (obtained via Lemma 3.1) are given in Section C.5. The results of this section hold for a hypothesis class $\mathcal{F}$ of networks implementing functions of the form

$$f = \sigma_L \circ f_L \circ \cdots \circ \sigma_1 \circ f_1 \quad \text{with} \quad f_i(x) = g_i(x) + (\sigma_{iL_i} \circ h_{iL_i} \circ \cdots \circ \sigma_{i1} \circ h_{i1})(x) \ , \quad (11)$$

i.e., a composition of $L$ residual blocks. Here, the nonlinearities $\sigma_i$ and $\sigma_{ij}$ are fixed and $\rho_i$-, resp., $\rho_{ij}$-Lipschitz continuous with $\sigma_i(0) = 0$ and $\sigma_{ij}(0) = 0$. We further fix the shortcuts to maps with $g_i(0) = 0$. The map $h_{ij}$ identifies the $j$-th layer in the $i$-th residual block with Lipschitz constraints $s_{ij}$ and distance constraints $b_{ij}$ (w.r.t. reference weights $M_{ij}$). Specifically, if $h_{ij}$ is *fully-connected*, then

$$h_{ij} \in \left\{ \phi : x \mapsto A_{ij}x \ \middle| \ \mathrm{Lip}(\phi) \leq s_{ij}, \ \left\| A_{ij}^\top - M_{ij}^\top \right\|_{2,1} \leq b_{ij} \right\} \ , \quad (12)$$

and, in case $h_{ij}$ is *convolutional*, then

$$h_{ij} \in \left\{ \phi_{K_{ij}} \ \middle| \ \mathrm{Lip}(\phi_{K_{ij}}) \leq s_{ij}, \ \left\| K_{ij} - M_{ij} \right\|_{2,1} \leq b_{ij} \right\} \ . \quad (13)$$

In terms of notation, $s_i = \mathrm{Lip}(g_i) + \prod_{j=1}^{L_i} \rho_{ij} s_{ij}$ further denotes the upper bound on the Lipschitz constant of the $i$-th residual block $f_i$. The Lipschitz constants are w.r.t. Euclidean norms; for a fully-connected layer this coincides with the spectral norm of the weight matrix.

The covering number bounds in Theorem 3.4 below depend on three types of quantities: (1) the total number of layers $\bar{L} = \sum_i L_i$, (2) the numbers $W_{ij}$ of parameters of the $j$-th layer in the $i$-th residual block, their maximum $W = \max_{ij} W_{ij}$, and (3) terms $C_{ij}$ that quantify the part of a layer's capacity attributed to weight and data norms. With respect to the latter, we define

$$C_{ij}(X) = 2 \frac{\|X\|}{\sqrt{n}} \left( \prod_{l=1}^{L} s_l \rho_l \right) \frac{\prod_{k=1}^{L_i} s_{ik} \rho_{ik}}{s_i} \frac{b_{ij}}{s_{ij}} \quad (14)$$

and write $C_{ij} = C_{ij}(X)$ for brevity. Importantly, $\|X\| \leq \sqrt{n} \max_i \|x_i\|$ and so the $C_{ij}$ can be bounded independently of the sample size. Overall, this yields the following covering number bounds for residual networks.

**Theorem 3.4.** *The covering number of the class of residual networks $\mathcal{F}$ as specified above, satisfies*

$$\log \mathcal{N}(\mathcal{F}, \epsilon, \|\cdot\|_X) \leq \begin{cases} \log(2W) \left( \sum_{i=1}^{L} \sum_{j=1}^{L_i} \left\lceil C_{ij}^{2/3} \right\rceil \right)^3 \left\lceil \frac{n}{\epsilon^2} \right\rceil & (15a) \\[2em] \sum_{i=1}^{L} \sum_{j=1}^{L_i} 2W_{ij} \log \left( 1 + \left\lceil \bar{L}^2 C_{ij}^2 \right\rceil \left\lceil \frac{n}{\epsilon^2} \right\rceil \right) & (15b) \end{cases} \ .$$

## 3.5 Rademacher complexity bounds

In combination with Dudley's entropy integral, Theorem 3.4 implies the empirical Rademacher complexity bounds in Theorem 3.5. These bounds equally hold for non-residual networks as in Eq. (1), i.e., the special case of setting the shortcuts $g_i$ to the zero map (with $L = 1$ block).

**Theorem 3.5.** *Let $\gamma > 0$ and define $\tilde{C}_{ij} = 2C_{ij}/\gamma$. Further, let $H_{n-1} = \sum_{m=1}^{n-1} 1/m = \mathcal{O}(\log(n))$ denote the $(n-1)$-th harmonic number. Then, the empirical Rademacher complexity of $\mathcal{F}_\gamma$ satisfies*

$$\hat{\mathfrak{R}}_S(\mathcal{F}_\gamma) \leq \frac{4}{n} + \frac{12 H_{n-1}}{\sqrt{n}} \sqrt{\log(2W)} \left( \sum_{i=1}^{L} \sum_{i=j}^{L_i} \left\lceil \tilde{C}_{ij}^{2/3} \right\rceil \right)^{3/2} \tag{$\clubsuit$}$$

*and*

$$\hat{\mathfrak{R}}_S(\mathcal{F}_\gamma) \leq \frac{12}{\sqrt{n}} \sqrt{ \sum_{i=1}^{L} \sum_{i=j}^{L_i} 2W_{ij} \left( \log\left( 1 + \left\lceil \bar{L}^2 \tilde{C}_{ij}^2 \right\rceil \right) + \psi\left( \left\lceil \bar{L}^2 \tilde{C}_{ij}^2 \right\rceil \right) \right) } \,, \tag{$\spadesuit$}$$

*where $\psi$ is a monotonically increasing function, satisfying $\psi(0) = 0$ and $\forall x : \psi(x) < 2.7$.*

The theorem considers the function class $\mathcal{F}_\gamma$ as defined in Eq. (3). As a consequence, the bounds depend on the quotients $\tilde{C}_{ij} = 2C_{ij}/\gamma$, which measure a layer's capacity (with respect to weight and data norms) relative to a classification margin parameter $\gamma$. As we will see in the experiments, constraining the layers' Lipschitz constants and weight norms, allows to substantially reduce the quantities $C_{ij}$ while the margin parameter $\gamma$ decreases only moderately.

Theorem 3.5 also immediately implies generalization bounds for $\mathcal{F}_\gamma$ via Lemma 3.1. In a subsequent step one can gradually decrease the constraint strengths and invoke a union bound argument over the corresponding generalization bounds, as for example done in [3, Lemma A.9]. This yields a generalization bound which does not depend on a priori defined constraint strengths, but on the actual Lipschitz constants and group norms computed from a neural network's weights.

**Interpretation.** To facilitate a clean comparison between the bounds in Theorem 3.5, we disregard the ceiling function and apply Jensen's inequality to the first bound ($\clubsuit$), yielding

$$\hat{\mathfrak{R}}_S(\mathcal{F}_\gamma) \leq \frac{4}{n} + \frac{12 H_{n-1}}{\sqrt{n}} \sqrt{ \log(2W) \sum_{ij} \bar{L}^2 \tilde{C}_{ij}^2 } \,. \tag{16}$$

Denoting $\tilde{C} = \max_{ij} \tilde{C}_{ij}$, Eq. (16) reveals that the bounds essentially differ only in that ($\clubsuit$) depends on $(\log(2W) \bar{L}^2 \tilde{C}^2)^{1/2}$ and ($\spadesuit$) depends on $(2W \log(1 + \bar{L}^2 \tilde{C}^2))^{1/2}$. Thus, the question of *which one is tighter*, hinges on the ratio of $2W$ and $\bar{L}^2 \tilde{C}^2$, i.e., a tradeoff between the number of parameters and the weight norms. As we see in Fig. 2, for simple, unconstrained networks, our second bound ($\spadesuit$) is much tighter. However, due to the logarithmic dependency on $\tilde{C}$, it is less affected by constraining the distances $b$ to initialization and the Lipschitz constants $s$. In Section 4, we show that this effect causes ($\clubsuit$) to be a more faithful measure of excess capacity. As $\tilde{C}$ depends exponentially on the network depth via the product of Lipschitz constants, another perspective on the bounds is that Eq. ($\clubsuit$) favors shallow architectures whereas Eq. ($\spadesuit$) favors narrow architectures. Notably, replacing the function class $\mathcal{F}_\gamma$ with a class of networks composed with a Lipschitz augmented loss function [43] facilitates deriving Rademacher complexity- and generalization bounds, which do not suffer from the exponential depth dependency via the product of Lipschitz constants. Instead, such bounds depend on data dependent empirical estimates thereof, which are typically much smaller.

**Relation to prior work.** Prior works [14, 16, 38] that tie generalization to $(1, \infty)$ group norms of matrices of fully-connected layers are equally applicable to convolutional networks without unfavorable scaling w.r.t. input size. In particular, for $(1, \infty)$ group norms of $M_K^\top$, we have $\|M_K^\top\|_{1,\infty} = \max_o \|K_{o\cdots}\|_1$, i.e., the maximum $l_1$ norm over each (input channel, width, height) slice of $K$. Yet, due to the layer-peeling strategy common to these works, the bounds scale with the product of matrix group norms *vs.* the product of Lipschitz constants (as in the $\tilde{C}_{ij}$ in Theorem 3.5) for covering number based strategies. While one can construct settings where the product of $(1, \infty)$ group norms is smaller than the product of Lipschitz constants, this is typically not observed empirically, cf. Fig. 2. Alternatively, Long & Sedghi [31] derive a generalization bound which does not depend on $l_p$

norms or group norms, but only on the distance to initialization with respect to the spectral norm. Notably, an intermediate result in this reference yields a generalization bound of similar form as (♠), scaling with the logarithm of the product of Lipschitz constants and with the square root of the number of parameters, see Section A.5. The distance to initialization then enters the main result [31, Theorem 3.1] at the cost of a Lipschitz constraint on the initialization. We argue that (♠) incorporates the distance to initialization more naturally, as it comes without constraints on the initialization itself. Further, it holds for any sample size $n$ and the numerical constants are explicit. Last, in the special case of fully-connected layers and no skip connections, (♣) reduces to the Rademacher complexity bound from [3]. Yet, there are three differences to this result: (i) a different numerical constant, (ii) the logarithm is replaced with a harmonic number, and (iii) there are no ceiling functions. From our understanding, these modifications are equally necessary when proving the special case directly. Nevertheless, these differences are only of minor importance, as they do not affect the asymptotic behavior of the bound. For more details, see Section A.4.

## 4   Empirical Evaluation

To assess the excess capacity of a neural network trained via a standard protocol on some dataset, we seek a hypothesis class that contains a network of the same architecture with comparable testing error but *smaller* capacity. Controlling capacity via the bounds in Theorem 3.5 requires *simultaneously* constraining the Lipschitz constants per layer and the $(2, 1)$ group norm distance of each layer's weight to its initialization. We first discuss how to enforce the constraints. Then, we fix a residual network architecture and train on datasets of increasing difficulty while varying the constraint strengths.

**Capacity reduction.** Controlling hypothesis class capacity necessitates ensuring that optimization yields a network parametrization that satisfies the desired constraints. To this end, we implement a variant of *projected* stochastic gradient descent (SGD) where, after a certain number of update steps, we project onto the intersection of the corresponding constraint sets $\mathcal{C}_1$ and $\mathcal{C}_2$. For convolutional layers, parametrized by tensors $K$, these are the convex sets $\mathcal{C}_1 = \{K : \|K - K^0\|_{2,1} \leq b\}$ and $\mathcal{C}_2 = \{K : \text{Lip}(\phi_K) \leq s\}$. Hence, jointly satisfying the constraints is a convex feasibility problem of finding a point in $\mathcal{C} = \mathcal{C}_1 \cap \mathcal{C}_2$. To ensure $\mathcal{C} \neq \emptyset$, we initially (prior to optimization) scale each layer's weight $K^0$ so that $\text{Lip}(\phi_{K^0}) = s$. This starting point (per layer) resides in $\mathcal{C}$ by construction.

To project onto $\mathcal{C}$, we rely on alternating *orthogonal projections* which map $K$ to a tensor in $\mathcal{C}_1$, resp. $\mathcal{C}_2$ with minimal distance to $K$. Repeated application of these projections converges to a point in the intersection $\mathcal{C}$ [5]. To implement the orthogonal projections onto $\mathcal{C}_1$ and $\mathcal{C}_2$, we rely on work by Liu et al. [30] and Sedghi et al. [40], respectively. The latter requires certain architectural prerequisites, and in consequence, we need to use convolutions of stride 1 (though our bounds equally hold for strides >1) and to reduce spatial dimensionality only via max-pooling. Further, we use circular padding and kernel sizes not larger than the input dimensionalities. For details on the projection algorithm and a comparison to alternating *radial* projections, see Section B.2.

**Architecture.** We use a slightly modified (pre-activation) ResNet18 [21]. Modifications include: (1) the *removal of batch normalization* and biases; (2) skip connections for residual blocks where the number of channels doubles and spatial dimensionality is halved are implemented via a fixed map. Each half of the resulting channels is obtained via $2 \times 2$ spatial max-pooling (shifted by one pixel). This map has Lipschitz constant $\sqrt{2}$ and is similar to the shortcut variant (A) of [20]; finally, (3) we *fix* the weight vectors of the last (classification) layer at the vertices of a $\kappa - 1$ unit simplex. Fixing the classifier is motivated by [18] and the simplex configuration is inspired by [17, 48] who show that this configuration corresponds to the geometric weight arrangement one would obtain at minimal cross-entropy loss. By construction, this classifier has Lipschitz constant $\sqrt{\kappa/\kappa-1}$. Notably, modifications (2) and (3) *do not* harm performance, with empirical testing errors on a par with a standard ResNet18 without batch normalization. Modification (1), i.e., the omission of normalization layers, was done to ensure that the experiments are in the setting of Eq. (11) and therefore that (♣, ♠) are faithful capacity measures. However, it is accompanied by a noticeable increase in testing error. In principle, our theory could handle batch normalization, as, during evaluation, the latter is just an affine map parametrized by the running mean and variance learned during training. However, including normalization in our empirical evaluation is problematic, as normalizing batches of small variance requires the normalization layers to have a large Lipschitz constant. Consequently, considering normalization layers as affine maps and enforcing Lipschitz constraints on them could

prevent proper normalization of the data. Another strategy would be to consider normalization layers as fixed nonlinearities which normalize each batch to zero mean and unit variance. However, this map is not Lipschitz continuous, and again, modifications could hinder normalization (which defeats the very purpose of these layers). Hence, we decided to remove normalization layers in our empirical evaluation. Presumably, however, there is a middle ground where capacity is reduced, and normalization is still possible. If so, excess capacity could be assessed for very deep architectures, which are difficult to train without normalization layers.

**Datasets & Training.** We test on three benchmark datasets: CIFAR10/100 [25], and Tiny-ImageNet-200 [24], listed in order of increasing task difficulty. We minimize the cross-entropy loss using SGD with momentum (0.9) and small weight decay (1e-4) for 200 epochs with batch size 256 and follow a CIFAR-typical stepwise learning rate schedule, decaying the initial learning rate (of 3e-3) by a factor of $5\times$ at epochs 60, 120 & 160 [10]. *No data augmentation is used*. When projecting onto the constraint sets, we found one alternating projection step every 15th SGD update to be sufficient to remain close to $\mathcal{C}$. To ensure that a trained model is within the capacity-constrained class, we perform 15 *additional* alternating projection steps after the final SGD update. For consistency, all experiments are run with the same hyperparameters. Consequently, hyperparameters are chosen so that training converges for the strongest constraints we assess. In particular, we train for 200 epochs even though unconstrained and weakly constrained models can be trained much faster. Importantly, this affects the assessment of excess capacity only marginally, as we observe that the testing error does not deteriorate in case of more update steps. Similarly, the Lipschitz constant and the distance to initialization stay almost constant once the close-to-zero training error regime is reached, which may happen way before 200 epochs are completed.

### 4.1 Results

First, we assess the capacity-driving quantities in our bounds for models trained *without* constraints[1]. Table 1 (top) lists a comparison across datasets, along with the capacity measures (♣, ♠), the training/testing error, and the empirical generalization gap (i.e., the difference between testing and training error). In accordance with our motivating figure from Section 1 (Fig. 1), we observe an overall *increase* in both capacity-driving quantities as a function of task difficulty.

To assess excess capacity in the context of questions (**Q1**) and (**Q2**), we first identify, per dataset, the *most restrictive* constraint combination where the testing error[2] is *as close as possible* to the unconstrained regime. We refer to this setting as the *operating point* for the constraints, characterizing the function class $\mathcal{F}$ that serves as a reference to measure excess capacity. The operating points per dataset, as well as the corresponding results are listed in Table 1 (bottom).

With respect to (**Q1**) we find that networks can indeed maintain, or even improve, testing error at substantially lowered capacity (see performance comparison in Fig. 3 relative to ◆). Furthermore, the observation that the capacity of the constrained models (surprisingly) remains in the *same order* of magnitude across tasks of varying difficulty, suggests a negative answer to question (**Q2**). A reduced-capacity model from an easy task can perform well on a difficult task. In consequence, when comparing the top *vs.* bottom part of Table 1 with respect to column (♣), we do see that task difficulty primarily manifests as *excess* capacity. Another manifestation of task difficulty is evident from the more detailed analysis in Fig. 3 (bottom), where we see that tightening both constraints beyond the identified operating point leads to a more rapid deterioration of the testing error as the task difficulty increases. Interestingly, at the operating point, the constrained models do not only share similar capacity across datasets, but also similar empirical *generalization gaps*, primarily due to leaving the ubiquitous zero-training-error regime. The latter is particularly noteworthy, as strong regularization (e.g., via weight decay) can equally enforce this behavior, but typically at the cost of a large increase in testing error (which we do not observe). Finally, the parameter-counting variant of our bound (see Table 1, column ♠) is, by construction, much less affected by the constraints and apparently fails to capture the observations above. This highlights the relevance of tying capacity to weight norms and underscores their utility in our context.

---

[1]At evaluation time, Lipschitz constants are computed via a power iteration for convolutional layers [15, 27].

[2]We are primarily interested in *what is feasible* in terms of tolerable capacity reduction. Hence, leveraging the testing split of each dataset for this purpose is legitimate from this exploratory perspective.

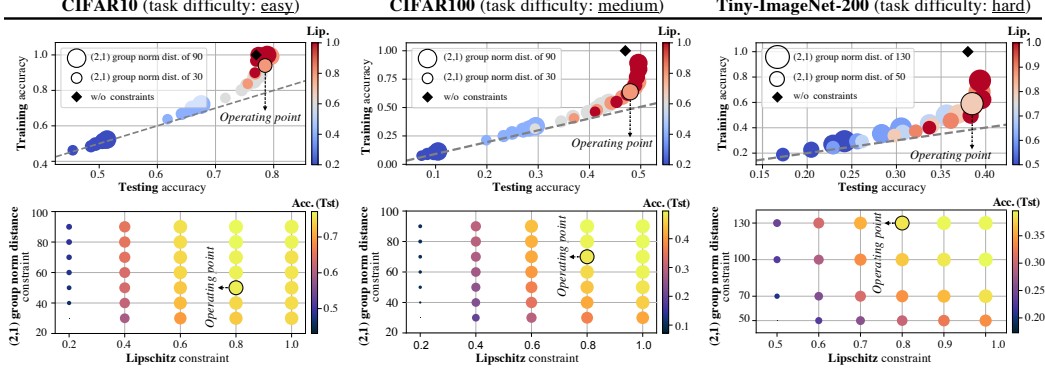

**Figure 3:** Fine-grained analysis of training/testing accuracy in relation to the Lipschitz constraint and the $(2, 1)$ group norm distance to initialization constraint. We see that testing accuracy can be retained (relative to ◆) for a range of fairly restrictive constraints (*top row*), compared to the unconstrained regime (cf. **Lip./Dist.** columns in the top part of Table 1). However, this range noticeably narrows with increasing task difficulty (*bottom row*) .

**Table 1:** Assessment of the capacity-driving quantities. We list the *median* over the Lipschitz constants (**Lip.**) and the $(2, 1)$ group norm distances (**Dist.**) across all layers. **Err.** denotes the training/testing error, **Capacity** denotes the measures (♣, ♠) from Theorem 3.5 and **Gap** the empirical generalization gap. The top part lists results in the unconstrained regime (see ◆ in Fig. 3), the bottom part lists results at the operating point of the most restrictive constraint combination where the testing error is not worse than in the unconstrained case. **Mar.** denotes the margin parameter $\gamma$ used for computing the capacity measures. As the constrained models do not fit the training data anymore, they do not have a positive classification margin. Thus, we choose $\gamma$ such that the unconstrained and constrained models have the same ramp loss value.

| | **Lip.** | **Dist.** | **Mar.** | **Err. (Tst)** | **Err. (Trn)** | **Capacity (♣, ♠)** | **Gap** |
|---|---|---|---|---|---|---|---|
| CIFAR10 | 1.63 | 60.3 | 11.2 | 0.24 | 0.00 | $1.0{\cdot}10^{10}$ / $8.8{\cdot}10^2$ | 0.24 |
| CIFAR100 | 2.17 | 129.1 | 23.4 | 0.54 | 0.00 | $1.7{\cdot}10^{11}$ / $9.3{\cdot}10^2$ | 0.54 |
| Tiny ImageNet | 3.05 | 287.3 | 24.7 | 0.62 | 0.00 | $4.5{\cdot}10^{13}$ / $1.1{\cdot}10^3$ | 0.62 |
| CIFAR10 | 0.80 | 50.0 | 1.00 | 0.21 | 0.06 | $1.8{\cdot}10^8$ / $7.8{\cdot}10^2$ | 0.15 |
| CIFAR100 | 0.80 | 70.0 | 1.00 | 0.52 | 0.36 | $2.6{\cdot}10^8$ / $7.9{\cdot}10^2$ | 0.16 |
| Tiny ImageNet | 0.80 | 130.0 | 1.00 | 0.62 | 0.41 | $8.9{\cdot}10^8$ / $8.9{\cdot}10^2$ | 0.21 |

# 5   Discussion

Studying the capacity of neural networks hinges crucially on the measure that is used to quantify it. In our case, capacity rests upon two bounds on the empirical Rademacher complexity, both depending on weight norms and the number of parameters, but to different extents. Hence, exerting control over the *weight norms* manifests in different ways: in case of the more weight norm dependent capacity measure, our results show substantial task-dependent *excess capacity*, while, when relying more on parameter counting, this effect is less pronounced. Although the latter measure yields tighter bounds, its utility in terms of explaining the observed empirical behavior is limited: in fact, capacity tied to weight norms not only better correlates with observed generalization gaps (both with and without constraints), but the amount of tolerable capacity reduction also reflects the smaller generalization gaps in the constrained regime. Note that our results rest upon carefully implementing constraint enforcement during optimization. Hence, numerical schemes to better account for this setting might potentially reveal an even more pronounced excess capacity effect.

In summary, our experiments, guided by the theoretical bounds, strongly suggest a notion of *compressibility* of networks with respect to weight norms. This compressibility only moderately reduces with task difficulty. We believe these observations to be particularly relevant and we foresee them sparking future work along this direction.

### Acknowledgments

This work was supported by the Austrian Science Fund (FWF) under project FWF P31799-N38 and the Land Salzburg under projects 0102-F1901166-KZP and 20204-WISS/225/197-2019.

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
