# Appendix (Supplementary Material) – Overview

In the appendix, we present (1) a more detailed comparison to prior work (Section A), (2) additional experiments (Section B) and (3) list all proofs which are left-out in the main manuscript (Section C). In particular, in Section C.2, we derive our *single-layer covering number bounds* from Theorem 3.2; Section C.3 presents the modularized strategy from Section 3.3 to obtain *whole-network* covering number bounds. This includes several examples (e.g., residual networks) and an extension to accommodate *concatenation*. Section C.4 then tailors our empirical Rademacher complexity bounds to networks with fully-connected and convolutional layers, and Section C.5 finally lists the accompanying generalization bounds.

# A Comparison with prior work on Rademacher complexity bounds

## A.1 Analysis of matrices corresponding to convolutions

We compare the norms of the matrices $M_K$ corresponding to the linear map realized by a convolutional layer with the norm of the corresponding weight tensor $K$. This facilitates studying the Rademacher complexity of convolutional layers via norm-based bounds for fully-connected layers.

In accordance with the definition of the $(2, 1)$ group norm in Eq. (7), we define the $(p, q)$ group norm of a weight tensor $K$ as

$$\|K\|_{p,q} = \left( \sum_{a,b=1}^{k} \sum_{o=1}^{c_{out}} \|K_{o \cdot ab}\|_p^q \right)^{1/q} . \tag{17}$$

For simplicity, we (i) consider only circular, input-size preserving paddings, (ii) assume that the spatial input dimension $h = w = d$ is a multiple of the convolution stride $t$, and (iii) assume that the kernel size $k_h = k_w = k \leq d$. In this setting, convolution corresponds to the application of a local map $\mathbb{R}^{c_{in} \times k \times k} \to \mathbb{R}^{c_{out}}$, specified by the weight tensor $K \in \mathbb{R}^{c_{out} \times c_{in} \times k \times k}$, to all $(d/t)^2$ patches. Each row $[M_K]_a$ of $M_K \in \mathbb{R}^{c_{out}(d/t)^2 \times c_{in}d^2}$ has as non-zero elements only the entries of the tensor $K_{o_a \cdots}$ for some $o_a = 1, \dots, c_{out}$. Thus, the $(p, q)$ group norm of $M_K^\top$ is

$$\left\|M_K^\top\right\|_{p,q} = \left( \sum_{a=1}^{c_{out}(d/t)^2} \|[M_K]_a\|_p^q \right)^{1/q} = \left( \sum_{a=1}^{c_{out}(d/t)^2} \|K_{o_a \cdots}\|_p^q \right)^{1/q} = \left( \frac{d}{t} \right)^{2/q} \left( \sum_{o=1}^{c_{out}} \|K_{o \cdots}\|_p^q \right)^{1/q} . \tag{18}$$

In particular,

$$\left\|M_K^\top\right\|_{2,1} = \left( \frac{d}{t} \right)^2 \sum_{o=1}^{c_{out}} \|K_{o \cdots}\|_2 , \tag{19}$$

$$\left\|M_K^\top\right\|_2 = \frac{d}{t} \|K\|_2 , \tag{20}$$

$$\left\|M_K^\top\right\|_{1,\infty} = \max_o \|K_{o \cdots}\|_1 . \tag{21}$$

Note the benefit of the $(1, \infty)$ group norm, which does not scale with the input dimension $d$.

We point out that for $p > q$ (Hölder inequality for $p/q$)

$$\|K_{o \cdots}\|_p^q = \left( \sum_{a,b=1}^{k} \|K_{o \cdot ab}\|_p^p \right)^{q/p} \geq \frac{1}{k^{2(1-q/p)}} \sum_{a,b=1}^{k} \|K_{o \cdot ab}\|_p^q \tag{22}$$

and so

$$
\begin{aligned}
\left\|M_K^\top\right\|_{p,q} &= \left( \left( \frac{d}{t} \right)^2 \sum_{o=1}^{c_{out}} \|K_{o \cdots}\|_p^q \right)^{1/q} \\
&\geq \left( \frac{d^2}{t^2 k^{2(1-q/p)}} \sum_{o=1}^{c_{out}} \sum_{a,b=1}^{k} \|K_{o \cdot ab}\|_p^q \right)^{1/q} \\
&= \left( \frac{d}{tk} \right)^{2/q} k^{2/p} \|K\|_{p,q} .
\end{aligned}
\tag{23}
$$

*This inequality quantifies the disadvantage of applying generalization bounds for fully-connected layers directly on the matrices that parametrize the linear maps corresponding to convolutions.*

## A.2 Comparison of bounds for convolutional networks

Table 3 lists various upper bounds on the empirical Rademacher complexity of convolutional networks as specified in the paragraph below, in a common notation (see Table 2). As in Section A.1, we

**Table 2:** Notation

| | | | |
|---|---|---|---|
| $n$ | Number of samples | $\kappa$ | Number of classes |
| $L$ | Number of layers | $c_i$ | Number of input channels to layer $i$ |
| $k_i$ | Kernel size at layer $i$ | $t_i$ | Stride at layer $i$ |
| $K_i$ | Weight of layer $i$, $K_i \in \mathbb{R}^{c_{i+1} \times c_i \times k_i \times k_i}$ | $K_i^{(0)}$ | Initialization of layer $i$ |
| $d_i$ | Input spatial width at layer $i$ | $s_i$ | Lipschitz constant of layer $i$ |
| $\gamma$ | Margin at output | $W_i$ | Number of parameters of layer $i$ |

**Table 3:** Comparison of empirical Rademacher complexity bounds (in our notation). When referring to sections or theorems from references, we underline the corresponding results.

| | |
|---|---|
| **Ours** (see Thm. 3.5 (♣)) | $\frac{4}{n} + \frac{12 H_{n-1}}{\sqrt{n}} \sqrt{\log(2W)} \left( \sum_{i=1}^{L} \left[ \left( \frac{4}{\gamma} \frac{\|X\|}{\sqrt{n}} \left( \prod_{l=1}^{L} s_l \right) \frac{\left\| K_i - K_i^{(0)} \right\|_{2,1}}{s_i} \right)^{2/3} \right] \right)^{3/2}$ |
| Bartlett et al. [3] (see Lem. A.8) | $\frac{4}{n} + \frac{48}{\gamma} \frac{\|X\|}{\sqrt{n}} \left( \prod_{i=1}^{L} s_i \right) \left( \sum_{i=1}^{L} \left( \log \left( \frac{2 W_i d_i^2}{t_i^2 k_i^2} \right) \frac{d_i^4}{t_i^4} \frac{\left( \sum_{o=1}^{c_{i+1}} \left\| [K_i - K_i^{(0)}]_{o\cdots} \right\|_2 \right)^2}{s_i^2} \right)^{1/3} \right)^{3/2} \frac{\log(n)}{\sqrt{n}}$ |
| Ledent et al. [26] (main result, see Thm. 11) | $768 R \sqrt{\log_2(32 \Gamma n^2 + 7 \bar{W} n)} \frac{\log(n)}{\sqrt{n}}$ 
 $R = \left( \sum_{i=1}^{L} r_i^{2/3} \right)^{3/2}, \qquad \Gamma = \max_i \left( r_i d_{i+1}^2 c_{i+1} \right), \qquad \bar{W} = \max_i d_i^2 c_i$ 
 $r_i = a_i B_{i-1}(X) \rho_{i+}$ 
 $a_i = \sum_o \left\| [K_i - K_i^{(0)}]_{o\cdots} \right\|_2, \qquad a_L = \left\| K_L - K_L^{(0)} \right\|_2$ 
 $\rho_{i+} = d_{i+1} \max_{U \leq L} \frac{\prod_{u=l+1}^{U} s_u}{B_U(X)}, \qquad \rho_{L+} = \frac{1}{\gamma}$ 
 $B_{i-1}(X) = $ maximal $l_2$ norm of a convolutional patch of the inputs to the $i$-th layer |
| Ledent et al. [26] (simpler result, see Sec. E) | $\frac{4}{n} + 768 R \sqrt{\log_2(32 \Gamma n^2 + 7 \bar{W} n)} \frac{\log(n)}{\sqrt{n}}$ 
 $R = \left( \sum_{i=1}^{L} r_i^{2/3} \right)^{3/2}, \qquad \Gamma = \max_i \left( r_i d_{i+1}^2 c_{i+1} \right), \qquad \bar{W} = \max_i d_i^2 c_i$ 
 $r_i = \frac{|X|_0}{\gamma} \max_o \| [K_L]_{o\cdots} \|_2 \left( \prod_{j=1}^{L-1} s_j \right) d_{i+1} \frac{\sum_o \left\| [K_i - K_i^{(0)}]_{o\cdots} \right\|_2}{s_i}$ 
 $|X|_0 = B_0(X)$ maximal $l_2$ norm of a convolutional patch on $X$ |
| **Ours** (see Thm. 3.5 (♠)) | $12 \sqrt{\sum_{i=1}^{L} 2 W_i \left( \log \left( 1 + \left\lceil L^2 \tilde{C}_i^2 \right\rceil \right) + \psi \left( \left\lceil L^2 \tilde{C}_i^2 \right\rceil \right) \right)} \frac{1}{\sqrt{n}}$ 
 $\tilde{C}_i = \frac{4}{\gamma} \frac{\|X\|}{\sqrt{n}} \left( \prod_{i=l}^{L} s_l \right) \frac{\left\| K_i - K_i^{(0)} \right\|_{2,1}}{s_i}, \qquad \psi(x) = \zeta \left( \frac{3}{2}, 1 \right)^{1/3} \zeta \left( \frac{3}{2}, 1 + 1/x \right)^{2/3} < 2.7$ |
| Lin et al. [29] (see Lem. 18) | $16 \left( \frac{2}{\gamma} \frac{\|X\|_2}{\sqrt{n}} L^2 \left( \prod_{i=1}^{L} s_i \right) \left( \sum_{i=1}^{L} W_i^2 \frac{d_i}{t_i} \frac{\|K_i\|_2}{s_i} \right) \right)^{1/4} \frac{1}{\sqrt{n}}$ |
| Neyshabur et al. [38] (see Cor. 2, with $l_{1,\infty}$) | $2^L \kappa \left( \prod_{i=1}^{L} \max_o \| [K_i]_{o\cdots} \|_1 \right) \log(2 c_1 d_1^2) \max_k \|x_k\|_\infty \frac{1}{\sqrt{n}}$ |
| Golowich et al. [14] (see Thm. 2, $l_{1,\infty}$) | $2 \kappa \sqrt{L + 1 + \log(c_1 d_1^2)} \left( \prod_{i=1}^{L} \max_o \| [K_i]_{o\cdots} \|_1 \right) \sqrt{\frac{\max_{abc} \sum_{k=1}^{n} [x_k]_{abc}^2}{n}} \frac{1}{\sqrt{n}}$ |
| Gouk et al. [16] (see Thm. 1, with $l_{1,\infty}$) | $2^{L+1} \kappa \sqrt{\log(2 c_1 d_1^2)} \left( \prod_{i=1}^{L} \max_o \| [K_i]_{o\cdots} \|_1 \right)$ 
 $\left( \sum_{i=1}^{L} \frac{\max_o \left\| [K_i - K_i^{(0)}]_{o\cdots} \right\|_1}{\max_o \| [K_i]_{o\cdots} \|_1} \right) \max_k \|x_k\|_\infty \frac{1}{\sqrt{n}}$ |
| Neyshabur et al. [38] (see Cor. 2, with $l_2$) | $2^{L-1} \kappa \frac{\|X\|_2}{\sqrt{n}} \left( \prod_{i=1}^{L} \frac{d_i}{t_i} \|K_i\|_2 \right) \frac{1}{\sqrt{n}}$ |
| Golowich et al. [14] (see Thm. 1, with $l_2$) | $\kappa \frac{\|X\|_2}{\sqrt{n}} \left( \prod_{i=1}^{L} \frac{d_i}{t_i} \|K_i\|_2 \right) (\sqrt{2 \log(2) L} + 1) \frac{1}{\sqrt{n}}$ |
| Gouk et al. [16] (see Thm. 2, with $l_2$) | $2^L \sqrt{2} \kappa \frac{\|X\|_2}{\sqrt{n}} \left( \prod_{i=1}^{L} \frac{d_i^2}{t_i} \sqrt{c_i} \|K_i\|_2 \right) \left( \sum_{i=1}^{L} \frac{\left\| K_i - K_i^{(0)} \right\|_2}{\|K_i\|_2 \prod_{j=1}^{i} d_j \sqrt{c_j}} \right) \frac{1}{\sqrt{n}}$ |

consider input-size preserving circular padding, convolutions with stride of $t$, and assume that (i) the spatial dimensions $h = w = d$ are a multiple of $t$ and (ii) the kernel size is $k_h = k_w = k \leq d$.

---

[3]The numerical constant (48) differs from the one (36) in [3] as discussed in Section A.4.

The empirical Rademacher complexity bounds in Table 3 are formulated in dependence of norms of the weights $K_i$. This is not entirely accurate, as the bounds typically refer to neural networks whose weights satisfy a priori defined norm *constraints*. We choose this abuse of notation so that we do not need to introduce additional variables for each norm constraint.

The listed bounds are used in the numerical comparison for unconstrained networks in Fig. 2. More specifically, we consider a hypothesis class $\mathcal{F}$ represented by a neural network of the form $f = \sigma_L \circ f_L \circ \cdots \circ \sigma_1 \circ f_1$, where $\sigma_i : x \mapsto \max(x, 0)$ denotes the ReLU activation function and $f_i$ identifies a convolutional layer. Note that fully-connected layers, e.g., a linear classifier at the last layer, can be handled by setting the spatial input dimension $d_i$, the kernel size $k_i$ and the stride $t_i$ all equal to 1.

Bounds designed for fully-connected networks are applied to the matrices $M_K$ that correspond to the weight tensor $K$ which parametrizes the convolution. To handle the multiclass regime, the covering number based bounds will be applied to $\mathcal{F}_\gamma$, see Eq. (3). Layer-peeling based bounds, originally presented for binary classification, are multiplied by the number of classes $\kappa$ (according to [32]) as done in [16].

### A.3 Details for the numerical comparison in Fig. 2

In Fig. 2, we evaluated several existing upper bounds (see Table 3) on the empirical Rademacher complexity of convolutional networks for two specific architectures.

The *first architecture* (a 6-layer network) consists of 5 convolutional layers with stride 2, kernel size 3, padding of 1 and 256 filters / output channels, so that a $(3 \times 32 \times 32)$-dimensional input image is mapped to a $(256 \times 1 \times 1)$-dimensional representation. The subsequent linear classifier is a convolutional layer with kernel size 1 and no padding. Its number of filters equals the number of classes of the classification problem. This classification layer is equivalent to applying a fully-connected layer to the flattened representations. The *second architecture* (an 11-layer network) only differs in that each convolutional layer with stride 2 is followed by an additional convolutional layer with stride 1 (kernel size 3, padding of 1 and 256 filters). All activation functions are ReLUs.

We trained both networks on the CIFAR10 dataset, minimizing the cross-entropy loss using stochastic gradient descent (SGD) with batch size 256, weight decay (1e-4), and momentum (0.9). During the 100 training epochs, the learning rate is gradually reduced following a cosine annealing schedule, starting with an initial learning rate of 1e-4. *Notably, we do not use any data augmentation*. Both networks fit the training data, achieving an accuracy of 72.5% (6-layer), resp. 77.9% (11-layer), on the test data.

To assess the different empirical Rademacher complexity bounds, we measured weight norms of the networks' layers and inserted them into the bounds from Table 3. Fig. 4 illustrates the results. Note that, following our discussion in Section A.2, Rademacher complexity bounds are typically formulated for networks with a priori specified weight norm constraints, whereas here, we train *unconstrained* networks and merely measure the weight norms at the end of training.

Fig. 4 highlights several aspects of the studied bounds. First of all, **all bounds are vacuous**, as they are larger than 1. Somewhat surprisingly, the bounds mainly driven by the number of parameters are clearly the smallest, i.e., the ones from Lin et al. [29] and Theorem 3.5 (♠). As expected, our bound from Theorem 3.5 (♣) is smaller than [3], as it accounts for the structure of convolutions. Furthermore, we see the benign scaling of the product of Lipschitz constants with the network depth compared to the product of $l_2$ norms, resp., $l_{1,\infty}$ norms. This is mirrored in the benign scaling of the covering number based bounds compared to the layer-peeling based ones.

### A.4 Comparison with Bartlett et al. [3]

Our Rademacher complexity bounds are based on the proof strategy of Bartlett et al. [3]. That is, we first derive single layer covering number bounds for *convolutional* layers. In a second step, we derive covering numbers for entire *residual* networks. Last, a combination of Dudley's entropy integral and [3, Lemma 3.1] implies the Rademacher complexity bounds (♣) and (♠). As already discussed in Section 3.2, our single-layer covering number bound for convolutional layers includes the single-layer covering number bound for fully-connected layers from [3, Lemma 3.2]. Consequently, in the special case of fully-connected layers and no skip connections, our main result (♣) reduces to

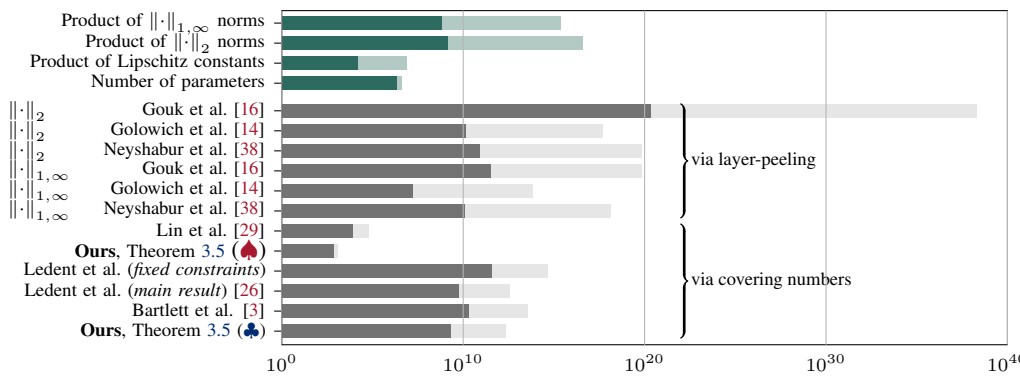

**Figure 4:** Empirical Rademacher complexity bounds (grouped by proof strategy), for a 6- (■■) and an 11-layer (■■) convolutional network, trained on CIFAR10. Quantities that typically appear in these bounds are shown in **green** (top part of figure) for reference.

the Rademacher complexity bound from [3]. To be more specific, we show that

$$\hat{\mathfrak{R}}_S(\mathcal{F}_\gamma) \leq \frac{4}{n} + \frac{12H_{n-1}}{\sqrt{n}}\sqrt{\log(2W)}\left(\sum_{i=1}^{L}\left\lceil \tilde{C}_i^{2/3}\right\rceil\right)^{3/2} \tag{24}$$

and Bartlett et al. [3] prove

$$\hat{\mathfrak{R}}_S(\mathcal{F}_\gamma) \leq \frac{4}{n} + \frac{9\log(n)}{\sqrt{n}}\sqrt{\log(2W)}\left(\sum_{i=1}^{L}\tilde{C}_i^{2/3}\right)^{3/2}. \tag{25}$$

Here, $L$ denotes the depth of the network and $\tilde{C}_i$ is the part of the capacity of the $i$-th layer due to weight and data norms, i.e.,

$$\tilde{C}_i(X) = \frac{4}{\gamma}\frac{\|X\|}{\sqrt{n}}\left(\prod_{l=1}^{L}s_l\rho_l\right)\frac{b_i}{s_i}, \tag{26}$$

with $s_i$ the layers' Lipschitz constraints, $b_i$ the layers' (2,1) norm constraints and $\rho_i$ the Lipschitz constants of the nonlinearities.

A closer look reveals that there are three differences between the results: (i) a different numerical constant, (ii) the logarithm is replaced with a harmonic number, and (iii) Eq. (26) contains no ceiling functions. Of course, the differences do not affect the asymptotic behavior of the bound and are thus only of minor importance. From our understanding, the differences are rooted in a lapse in the inequality chains of [3, Eq. (A.3)].

- The difference in the numerical constant appears, because proving the entire network covering number bound requires transitioning to *external* covering numbers, which manifests as an additional factor of 2 in the final result. This is because the single layer covering number bounds from Theorem 3.2, resp. [3, Lemma 3.2], hold for layers with only a $(2,1)$ group norm constraint, which form a *superset* of the layers with a $(2,1)$ group norm constraint *and* a Lipschitz constraint as considered in Theorem 3.5, resp [3, Theorem 3.3]. On the other hand, the parameter $\alpha$ in the proof of [3, Lemma A.8] can be chosen as $\alpha = 1/\sqrt{n}$, which improves the bound by a factor of $3/2$ (see proof of Theorem C.19). Overall, both effects lead to a factor of $4/3$, which is precisely the quotient of the numerical constants in Eq. (25) and Eq. (26).

- Our result in Eq. (25) contains ceiling functions and a harmonic number, which is a direct consequence of the ceiling function appearing in the single layer covering number bound of Theorem 3.2, resp. [3, Lemma 3.2]. In the chain of inequalities [3, Eq. (A.3)] in the proof of [3, Theorem 3.3], the single layer bound is inserted *without* the ceiling function.

## A.5 Comparison with Long & Sedghi [31]

Long & Sedghi [31] study generalization bounds for the class $\mathcal{F}$ of convolutional networks that realize functions of the form

$$f = \sigma_L \circ \phi_{K_L} \circ \cdots \circ \sigma_1 \circ \phi_{K_1} : \mathcal{X} \to \mathbb{R}$$

with Lipschitz/spectral-norm constraints, i.e., they assume that the initializations $\phi_{K_i^{(0)}}$ per layer are $(1 + \nu)$-Lipschitz and that the distances

$$\beta_i = \text{Lip}(\phi_{K_i} - \phi_{K_i^{(0)}})$$

to initialization satisfy $\sum_i \beta_i \leq \beta$ for some given constant $\beta > 0$. They show [31, Theorem 3.1] that for $\lambda$-Lipschitz loss functions $\ell$, the generalization gap is (with probability $1 - \delta$) uniformly bounded over the class $\mathcal{F}_\ell = \{(x, y) \mapsto \ell(f(x), y) \mid f \in \mathcal{F}\}$ by

$$CM\sqrt{\frac{\bar{W}(\beta + \nu L + \log(\lambda\beta\chi)) + \log\left(\frac{1}{\delta}\right)}{n}} \quad, \tag{27}$$

assuming that $\lambda\beta\chi(1 + \nu + \beta/L)^L \geq 5$ and $n$ large enough. Here, $C$ denotes an unspecified constant and $M$ the maximum of the loss function $\ell$. Further, $\bar{W} = \sum_i W_i$ is the total number of network parameters and $\|x\|_2 \leq \chi$ is an upper bound on the Euclidean norm of the data. As can be seen, this bound depends on the square root of parameters and the distance $\beta$ to initialization. In contrast to other results (e.g., [3, 26, (♣), (♠)]), it also depends on a Lipschitz constraint $(1 + \nu)$ *directly* on the initialization.

Eq. (27) is based on [31, Lemma 2.3], which requires the class $\mathcal{F}_\ell$ to be $(B, d)$-*Lipschitz parametrized*, i.e., that there exists $d \in \mathbb{N}$ and a norm $\|\cdot\|$ on $\mathbb{R}^d$, together with a $B$-Lipschitz continuous and surjective map $\phi : \mathcal{B}(1, \|\cdot\|) \to \mathcal{F}_\ell$ from the $\|\cdot\|$-unit ball in $\mathbb{R}^d$ onto $\mathcal{F}_\ell$, which is $B$-Lipschitz. The latter means that for every $\theta, \theta' \in \mathcal{B}(1, \|\cdot\|)$, it holds that $\|\phi(\theta) - \phi(\theta')\|_\infty \leq B \|\theta - \theta'\|$. In this situation, the generalization gap is bounded by

$$CM\sqrt{\frac{d \log B + \log\left(\frac{1}{\delta}\right)}{n}} \quad. \tag{28}$$

In a series of lemmas [31, Lemma 3.2–3.4], the authors show that $\mathcal{F}_\ell$ is indeed $(B, d)$-Lipschitz parametrized with $d = \bar{W}$ and $B = \lambda\chi\beta(1 + \nu + \beta/L)^L$. We will repeat the argument and show that it implies an intermediate result which scales similarly to our result (♠) from Theorem 3.5, i.e., with the square root of

$$\bar{W} \log\left(\prod_j s_j\right) \quad,$$

where $s_i$ denote Lipschitz constraints on the layers $\phi_{K_i}$.

Let $\mathbf{K} = (K_1, \ldots, K_L)$ and $\tilde{\mathbf{K}} = (\tilde{K}_1, \ldots, \tilde{K}_L)$ be tuples of weight tensors and denote the corresponding networks by $f_{\mathbf{K}}$, resp. $f_{\tilde{\mathbf{K}}}$. If $\mathbf{K}$ and $\tilde{\mathbf{K}}$ differ in only one layer, say $K_j \neq \tilde{K}_j$, then for all $x \in \mathcal{X}$ (see proof of [31, Lemma 3.2]),

$$|f_{\mathbf{K}}(x) - f_{\tilde{\mathbf{K}}}(x)| \leq \|x\|_2 \left(\prod_{i \neq j} \text{Lip}(\phi_{K_i})\right) \text{Lip}(\phi_{K_j} - \phi_{\tilde{K}_j}) \leq \chi \left(\prod_{i=1}^{L} s_i\right) \text{Lip}(\phi_{K_j} - \phi_{\tilde{K}_j}) \quad. \tag{29}$$

Consequently, if $\mathbf{K}$ and $\tilde{\mathbf{K}}$ differ in *all* layers, it holds that

$$|f_{\mathbf{K}}(x) - f_{\tilde{\mathbf{K}}}(x)| \leq \chi \left(\prod_{i=1}^{L} s_i\right) \sum_{j=1}^{L} \text{Lip}(\phi_{K_j} - \phi_{\tilde{K}_j}) \quad. \tag{30}$$

As $\sum_{j=1}^{L} \text{Lip}(\phi_{K_j})$ defines a norm $\|\cdot\|$ on $\mathbb{R}^{\bar{W}}$ (in [31] this norm is denoted $\|\cdot\|_\sigma$), the inequality above implies that the surjective map

$$\mathcal{B}(1, \|\cdot\|) \to \mathcal{F} \quad, \quad \frac{\mathbf{K}}{\sum_i s_i} \mapsto f_{\mathbf{K}}$$

is $\left(\chi\left(\prod_{i=1}^{L} s_i\right)\sum_i s_i\right)$-Lipschitz, i.e., the class $\mathcal{F}_\ell = \{(x,y) \mapsto \ell(f(x),y)\mid f \in \mathcal{F}\}$ is $\left(\lambda\chi\left(\prod_{i=1}^{L} s_i\right)\sum_i s_i, \bar{W}\right)$-Lipschitz parametrized. Thus, Eq. (28) implies a generalization bound of the form

$$CM\sqrt{\frac{\bar{W}\log\left(\lambda\chi\left(\prod_{i=1}^{L} s_i\right)\sum_i s_i\right) + \log\left(\frac{1}{\delta}\right)}{n}} . \tag{31}$$

Similarly to our result (♠) from Theorem 3.5, this bound scales with the square root of the number of parameters and with the logarithm of the product of Lipschitz constants. However, as Eq. (31) and Eq. (27) are proven via an asymptotic bound from Giné and Guillou [13], the constant $C$ and the minimal sample size $n$ required for Eq. (31) and Eq. (27) to hold are not readily available. This makes further comparisons difficult.

Eq. (31) differs from the main result in [31], i.e., Eq. (27), as, instead of constraints on the layers' Lipschitz constants $\mathrm{Lip}(\phi_{K_i}) \leq s_i$, Long & Sedghi consider constraints on the Lipschitz constants of the initialization $\mathrm{Lip}(\phi_{K_i^{(0)}}) \leq 1 + \nu$ and on the distance to initialization $\mathrm{Lip}(\phi_{K_i} - \phi_{K_i^{(0)}}) \leq \beta_i$ with $\sum_i \beta_i = \beta$. Starting from Eq. (29), these constraints enter via the triangle inequality, i.e.,

$$\mathrm{Lip}(\phi_{K_i}) \leq \mathrm{Lip}(\phi_{K_i^{(0)}}) + \mathrm{Lip}(\phi_{K_i} - \phi_{K_i^{(0)}}) \leq 1 + \nu + \beta_i .$$

Maximizing $\prod_i(1 + \nu + \beta_i)$ subject to $\sum_i \beta_i = \beta$, yields

$$|f_{\mathbf{K}}(x) - f_{\tilde{\mathbf{K}}}(x)| \leq \chi\left(1 + \nu + \beta/L\right)^L \mathrm{Lip}(\phi_{K_j} - \phi_{\tilde{K}_j}) \tag{32}$$

$$\leq \chi\exp\left(\nu L + \beta\right)\mathrm{Lip}(\phi_{K_j} - \phi_{\tilde{K}_j}) . \tag{33}$$

[31, Lemma 3.3 & 3.4] then imply that $\mathcal{F}_\ell$ is $(B,d)$-Lipschitz parametrized with $d = \bar{W}$ and $B = \lambda\chi\beta\exp\left(\nu L + \beta\right)$, which in turn implies Eq. (27).

Obviously, every bound that depends on weight norms can be transferred to a bound that depends on the norm of the initialization and the distance to it, simply by application of the triangle inequality. We argue, that utilizing the translation invariance of covering numbers, as done in, e.g., (♣), (♠), as well as in [3, 26], is a more natural way of incorporating the distance to initialization, as it allows for bounds which do not depend on norm constraints on the initialization.

### A.6 Comparison with Ledent et al. [26]

In [26], Ledent et al. derive generalization/Rademacher complexity bounds via $l_\infty$ coverings of convolutional networks. These bounds incorporate weight sharing and thus directly depend on the norms of the weight tensors, instead of depending on the norms of the matrix that parametrizes the linear (convolutional) map. This results in an improved scaling with respect to the spatial input width.

In general, the bounds in [26] scale similarly to our bound (♣) from Theorem 3.5 in that they depend on the square root of the product of Lipschitz constants (or empirical estimates thereof). In particular, just as our result (♣), [26, Theorem 16] is based on Rademacher complexity bounds for function classes $\mathcal{F}_\gamma$, i.e., the composition of Lipschitz- and distance-constrained convolutional networks with the ramp loss at margin $\gamma > 0$. The main result [26, Theorem 3], as well as [26, Theorem 20], adapts techniques from [43] and [35] to replace the product of Lipschitz constants with empirical equivalents, which are typically much smaller. To this end, they study the composition of convolutional networks with an *augmented* loss function, see for example [26, Eq. (26)].

In this part of the appendix, we compare our norm-driven bound (♣) with the main results in [26]. As mentioned in Section 3.2, we find that both results exhibit similar scaling behavior, but *we improve in the logarithmic term and in that our dependency on data norms is less sensitive to outliers*. On the other hand, the main bounds in [26] exhibit an improved dependency on the number of classes. The latter pays off, e.g., for shallow networks or in extreme multiclass problems with a large number of classes. All three effects are due to the use of $l_2$ *vs.* $l_\infty$ covering numbers.

Central to all Rademacher complexity bounds [26] is the single-layer $l_\infty$ covering number bound restated in the proposition below.

**Proposition A.1** ([26, Proposition 6]). *Let positive reals $(a, b, \epsilon)$ and positive integer $m$ be given. Let the tensor $X \in \mathbb{R}^{n \times U \times d'}$ be given with $\forall i \in \{1, \dots, n\}, \forall u \in \{1, \dots, U\}, \|X_{iu\cdot}\|_2 \leq b$. For*

*any fixed $M$:*

$$\log \mathcal{N}\left(\left\{XA\colon A \in \mathbb{R}^{d'\times m}, \|A - M\|_{2,1} \leq a\right\}, \epsilon, \|\cdot\|_*\right) \leq \frac{64a^2b^2}{\epsilon^2}\log_2\left[\left(\frac{8ab}{\epsilon} + 7\right)mnU\right] \quad (34)$$

*with the norm $\|\cdot\|_*$ over the space $\mathbb{R}^{n\times U\times m}$ defined by $\|Y\|_* = \max_{i\leq n}\max_{j\leq U}(\sum_{k=1}^m Y_{ijk}^2)^{1/2}$.*

Some remarks regarding the notation. Here, $X$ does *not* denote the input data $(x_1, \ldots, x_n)$, but the $nU$-tuple of all $d'$-sized convolutional patches of the input data. Thus, $d' = k^2c_{in}$ is the square of the kernel size times the number of input channels and $U = \lceil d/t\rceil^2$ is the number of patches per image, which is computed as the square of the spatial width divided by the stride. The matrix $A \in \mathbb{R}^{k^2c_{in}\times c_{out}}$ then is the local linear map acting on the convolutional patches, i.e., $A$ is a reshaping of the weight tensor $K$ and $XA$ is the output of the convolutional layer, i.e., $n$ images with $c_{out}$ channels with $U$ pixels each. Further, $\|A\|_{2,1}$ is the *standard* matrix $(2,1)$ group norm which differs from $\|K\|_{2,1}$ defined in Eq. (7).

As the single-layer bound in Eq. (34) and our single-layer bound in Eq. (54) are the fundamental building blocks of all inferred results (and we did not study augmented loss functions), we will focus on them for the comparison. For ease of reference, we restate the relevant part of Theorem 3.2.

**Theorem A.2.** *Let $b > 0$ and $\mathcal{F} = \{\phi_K |\ K \in \mathbb{R}^{c_{out}\times c_{in}\times k\times k}, \|K\|_{2,1} \leq b\}$ denote the class of 2D convolutions with $c_{in}$ input channels, $c_{out}$ output channels and kernel size $k\times k$, parametrized by tensors $K$ with $W$ parameters. Then,*

$$\log\mathcal{N}(\mathcal{F}, \epsilon, \|\cdot\|_X) \leq \left\lceil\frac{\|X\|^2 b^2}{\epsilon^2}\right\rceil \log(2W)\ . \quad (35)$$

There is a clear similarity between Eq. (34) and Eq. (35). Both depend quadratically on weight and data norms divided by the covering radius $\epsilon$, as well as on a logarithmic term. Consequently, differences between both bounds are nuanced and, ignoring the constant in Eq. (34), it is a priori not clear which bound is preferable. We will discuss these nuances theoretically and provide a empirical comparison in Fig. 5.

(Diff-1) **Data norms.** Our work assumes a bound on the $l_2$ norm of the *whole input* $x$ (a $c_{in}d^2$-tuple), whereas [26] only assumes a bound on the $l_2$ norm of every *single patch* $p$ ($c_{in}k^2$-tuples). This potentially improves Eq. (34) over Eq. (35) by a factor of $(k/d)^2$, as

$$\max_{p\in\text{patches}}\|p\| \leq \|x\| \lesssim d/k \max_{p\in\text{patches}}\|p\|\ . \quad (36)$$

The left inequality is obvious. The right inequality follows from considering the sum of all patch norms. As every pixel $x_{ijk}$ appears in at least $\lfloor k/s\rfloor^2$ patches and there are at most $\lceil d/s\rceil^2$ patches in total, it holds that

$$\lfloor k/s\rfloor^2 \|x\|^2 = \sum_{i=1}^c \sum_{j,l=1}^d |x_{ijl}|^2 \lfloor k/s\rfloor$$

$$\leq \sum_{i=1}^c \sum_{j,l=1}^d |x_{ijl}|^2\,\text{card}(\{p \in \text{patches} \mid x_{ijl} \in p\})$$

$$= \sum_{p\in\text{patches}} \|p\|^2 \leq \lceil d/s\rceil^2 \max_{p\in\text{patches}} \|p\|^2\ ,$$

and $\lfloor k/s\rfloor/\lceil d/s\rceil \approx k/d$. Notably, the maximum in Eq. (34) is over the patches on *all* of the input data, which is quite *sensitive to outliers*. Hence, the improvement over Eq. (35) is typically smaller than $d/k$, especially at hidden layers, see top row of Fig. 5.

(Diff-2) **Weight norms.** The $(2,1)$ group norms on the weights are applied differently, i.e., we compute a $(2,1)$ norm via Eq. (7), whereas [26] computes the $(2,1)$ group norm of the matrix corresponding to the local linear map, which is applied to each patch, i.e., $\sum_i \|K_{i\cdots}\|_2$. As

$$\sum_i \|K_{i\cdots}\|_2 \leq \|K\|_{2,1} \leq k\sum_i \|K_{i\cdots}\|_2\ , \quad (37)$$

this potentially improves Eq. (34) over Eq. (35) by a factor of $1/k^2$. Empirically, we observe that $\|K\|_{2,1} \approx k \sum_i \|K_{i\cdots}\|_2$, see bottom row of Fig. 5.

Thus, considering norm constraints only, i.e., (Diff-1) and (Diff-2), we find that Eq. (34) is potentially better by a factor $(k/d)^2 \cdot (1/k)^2 = 1/d^2$, i.e., the reciprocal of the squared height/width of the input images. However, the comparison is more intricate, as the coverings are with respect to different ($l_2$ vs. $l_\infty$) norms and, more importantly, the considered function classes differ. As, ultimately, we want to get Rademacher complexity bounds for whole networks, we need to consider effects that appear when transitioning to whole-network bounds.

(Diff-3) **Lipschitz constants.** In whole-network bounds, contributions of all layers are summed. These contributions are the (logarithmic) single-layer bounds, scaled by a factor corresponding to the Lipschitz constant of the remainder of a network after the layer. Typically, the Lipschitz constant of the part before a layer additionally enters as an estimate of the norm of the layer's input. Notably, in [26], the Lipschitz constant of the network's remaining layers incurs an additional factor $\lceil d/t \rceil^2$, i.e., the spatial dimensionality of the output (denoted by $w_l$ in the reference). This counterbalances the improvements by (Diff-1) and (Diff-2).

Specifically, in [26], the Lipschitz constants are with respect to the norms $\|\cdot\|_{\infty,r}$ on the domain and $|\cdot|_s$ on the codomain, see, e.g., the definition of $\rho_{l_1 \to l_2}^{\mathcal{A}}$ in the statement of [26, Proposition 10]. There,

$$\|x\|_{\infty,r} = \max_{j \leq d} \max_{k \leq d} \sqrt{\sum_{i=1}^{c} x_{ijk}^2}$$

is the maximum $l_2$ norm of a slice of the image $x$ along the channels, i.e., at fixed spatial position. The norm $|y|_s = \max_{p \in \text{patches}} \|p\|$ is the maximal norm of a convolutional patch on $y$. Transitioning to Lipschitz constants with respect to $l_2$ norms, i.e., spectral norms, as done for the main result in [26, Theorem 3], incurs an additional factor $d$ (the spatial dimension of $x$), since

$$\frac{|f(x)|_s}{\|x\|_{\infty,r}} = \underbrace{\frac{|f(x)|_s}{\|f(x)\|_2}}_{\leq 1} \frac{\|f(x)\|_2}{\|x\|_2} \underbrace{\frac{\|x\|_2}{\|x\|_{\infty,r}}}_{\leq d} \leq d \frac{\|f(x)\|_2}{\|x\|_2} \quad.$$

In this inequality, $x$ denotes the output of the considered layer and so $d$ is its spatial width. Notably, in [26], $d$ can actually be reduced to the output's spatial width after a subsequent max-pooling operation. In our whole-network bound, the Lipschitz constant is already with respect to $l_2$ norms and thus no additional factors appear.

(Diff-4) **Dependency on number of classes.** The use of $l_\infty$ covering numbers in [26] improves the dependency on the number of classes for whole-network bounds. This is because the weights of the classification layer do not enter via a $(2,1)$ group norm constraint, but a Frobenius norm constraint. This implicitly improves the log covering number of this layer by a factor of the number of classes. Since, for whole-network bounds, the contribution of all layers are summed, we expect this effect to be significant if the contribution of the classification is a substantial fraction of the whole-network bound. This would be the case, e.g., for shallow networks or in extreme multiclass settings.

Finally, we discuss the logarithmic terms and constants.

(Diff-5) **Logarithmic terms.** Our bound in Eq. (35) depends logarithmically on the number of parameters, denoted by $W$. By contrast, Eq. (34) depends on $\log_2 \left[ \left( \frac{8ab}{\epsilon} + 7 \right) mnU \right]$. When transitioning to Rademacher complexity bounds via Dudley's entropy integral (cf. [26, Eq. (29)]), the covering radius $\epsilon$ in the $\log_2$ term is replaced by $\frac{1}{n}$. So, considering the definitions of $U$ and $m$, we need to compare $8abmUn^2 = 8abc_{out}d^2n^2/t^2$ (Ledent et al.) with $2W = 2c_{in}c_{out}k^2$ (Ours). As, typically, $k \leq d/t$ and $c_{in} < n^2$, we improve over [26] in the logarithmic term (recall that $a$ and $b$ denote the weight and data norm constraints, respectively).

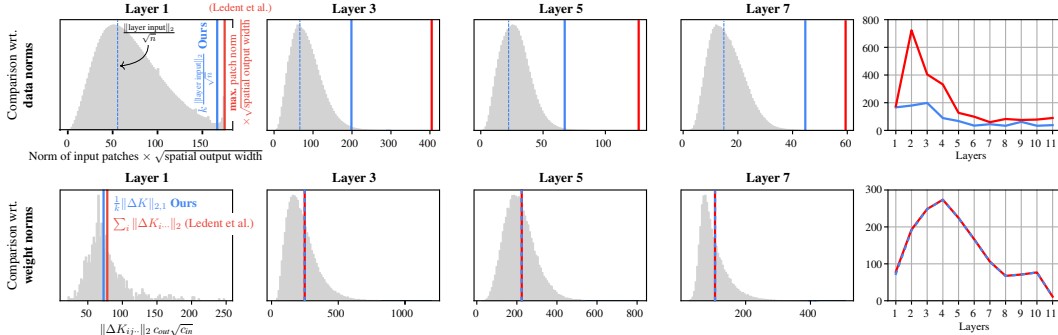

**Figure 5:** Comparison of factors (w.r.t. *data norms* and *weight norms*) in the single-layer covering number bound of Ledent et al. [26, Proposition 6] and our result from Eq. (35). Shown are detailed results for four exemplary layers (from the 11-layer convolutional network described in Section A.3), as well as a summary plot across all layers (rightmost). The first row presents histograms of patch norms; the second row presents histograms of norms of $k \times k$ slices of tensors $\Delta K$, i.e. the difference $\Delta K$ between a weight tensor and its initialization.

(Diff-6) **Multiplicative constants.** The single-layer bound by Ledent et al. Eq. (34) has a rather large multiplicative constant $64$ (compared to $1$ in Eq. (35)). This constant enters mainly via a previous theorem by Zhang [47, Theorem 4]. Notably, a remark in [47] highlights that the constants in this theorem are not optimized. Thus, improving Eq. (34) in this regard might be possible, and the difference in numerical constants might be less pronounced than it appears at first sight. Yet, from our understanding, some constants are unavoidable, e.g., the factor $2$ which enters the proof sketch of [26, Proposition 6].

Overall, (Diff-1) – (Diff-3) lead to several effects, which can potentially compensate each other, especially if, for each layer, the coordinates of its input and of the weights have roughly equal norm. Notably, in this situation, our single-layer bound can be improved by a factor of $1/t^2$, with $t$ denoting the stride of the convolution, see Remark C.9. Thus, in the absence of pooling (e.g., when downsampling is handled directly by the stride of the convolutional layer), the scaling is precisely the same. As Eq. (34) depends on the maximum norm of a patch over *all of the input data* (i.e., a quantity which is sensitive to outliers), we do not expect (Diff-1) – (Diff-3) to fully compensate each other, but rather expect an advantage of our bound from Theorem A.2. A detailed investigation is shown in Fig. 5, which highlights weight and data norms for layers of a trained network (an 11-layer convolutional network as used for Fig. 4). To incorporate the effects (Diff-1) – (Diff-3) and to allow for a cleaner comparison, we multiply the $l_2$ norms of the patches by the square root of spatial dimensionality of the output and shift a factor of the kernel size from our weight norms to our data norms. We see that the (rescaled) weight norms across all layers are essentially the same, whereas, due to the maximum being sensitive to outliers, our data norms are substantially smaller at the hidden layers.

As a last comparison, we illustrate the magnitudes of all factors appearing in the bounds of Eqs. (34) and (35) and of the spatial dimensionality of the output, see (Diff-3) in the enumeration above. In both bounds, we discard the denominator $\epsilon^2$ and, in Eq. (34), we replace the factor $1/\epsilon$ in the logarithm by $n$, just as it enters the Rademacher complexity bounds.

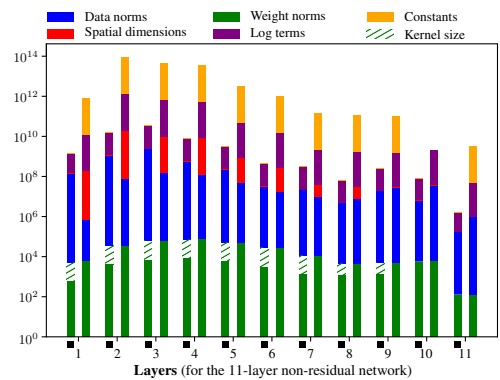

As can be seen from the figure to the right (with our single-layer bound marked by ■), our improvement in the quadratic terms is due to data norms. To be specific, one needs to compare the data norms in our case, to the combination of data norms and spatial dimensions in the bound of Ledent et al. [26]. We also improve in the logarithmic terms and constants.

**Comparison for two-layer networks**

As discussed in (Diff-4), the $l_\infty$ covering approach in [26] allows for a favorable treatment of the last (classification) layer. If the contribution of this last layer to the respective Rademacher complexity bounds is substantial, then the bound in [26] is superior. We evaluate this effect on two-layer networks, where it is most pronounced.

The comparison considers networks of the following architecture. The first layer is convolutional and parametrized by a tensor $K \in \mathbb{R}^{c \times 3 \times k \times k}$. Here $c$ denotes the number of filters (channels of the output) and $k$ the kernel size, which is chosen equal to the stride and the spatial dimensionality of the input, e.g., 32 for images from CIFAR100. Consequently, the spatial dimensionality of the output is 1. This convolutional layer is followed by an activation function with Lipschitz constant 1 (e.g., ReLU) and a linear map $W \in \mathbb{R}^{\kappa \times c}$, with $\kappa$ denoting the number of classes.

Since the quantities and norms appearing in the respective bounds differ, we make the following simplifications, which are motivated by corresponding inequalities and verified empirically.

(1) $\|K\|_{2,1} \approx k \sum_{i=1}^{c} \|K_{i\cdots}\|_2$ ,

(2) $\|W^\top\|_{2,1} \approx \sqrt{c}\|W\|_2$ ,

(3) $\max_{j \leq \kappa} \|W_{j\cdot}\|_2 \approx \mathrm{Lip}(W)$ ,

(4) $\frac{\|X\|_2}{\sqrt{n}} \approx \max_{i \leq n} \|x_i\|_2 =$ maximal norm of convolutional patches from the data ,

(5) $H_{n-1} \approx \log n$ .

Furthermore, just as the single-layer bound in [26] depends on the maximal norm of a patch of the data, ours actually depends only on the maximal norm of particular slices of the data, which we here denote as $|X|_s$. In the special case of the stride being equal to the spatial dimensionality, these slices are over the channels at fixed local position (see Remark C.9 and the last chain of inequalities in the proof of Theorem C.7). Thus, for the contribution of the first layer, we can use

(6) $|X|_s \approx \frac{\|X\|}{k}$ .

Last, we empirically evaluate a maximum operator which appears in the quantity $\mathcal{R}$ in the two layer bound (Theorem 2) in [26], i.e.,

(7) $\frac{1}{\max_{i \leq n} \|\phi_K(x_i)\|_2} \leq \frac{\max_{j \leq \kappa} \|W_{j\cdot}\|_2}{\gamma}$ .

With these simplifications, **our** bound (♣) becomes

$$\frac{48}{\gamma} \frac{\log n}{\sqrt{n}} \sqrt{\log(6ck^2)} \left[ \left( \frac{\mathrm{Lip}(W) \|K-K^{(0)}\|_{2,1}}{k} \right)^{\frac{2}{3}} + \left( \mathrm{Lip}(\phi_K) \|(W-W^{(0)})^\top\|_{2,1} \right)^{\frac{2}{3}} \right]^{\frac{3}{2}} \quad (38)$$

and the bound from Theorem 2 in [26] becomes

$$\frac{768}{\gamma} \frac{\log n}{\sqrt{n}} \sqrt{n^2 \mathcal{D}} \left[ \left( \frac{\mathrm{Lip}(W) \|K-K^{(0)}\|_{2,1}}{k} \right)^{\frac{2}{3}} + \left( \frac{\mathrm{Lip}(\phi_K) \|(W-W^{(0)})^\top\|_{2,1}}{\sqrt{\kappa}} \right)^{\frac{2}{3}} \right]^{\frac{3}{2}} \quad (39)$$

with $\mathcal{D} = \max \left( \frac{\|K - K^{(0)}\|_{2,1}}{k} \frac{\mathrm{Lip}(W)}{\mathrm{Lip}(\phi_K)} c \;,\; \frac{\|X\|_2}{\sqrt{n}} \|W - W^{(0)}\|_2 \frac{\mathrm{Lip}(K)}{\gamma} \kappa \right)$ .

As expected, ignoring constants and log terms, the bound from [26] is better by a factor of $\sqrt{\kappa}$ (square root of number of classes) in the summand corresponding to the last layer.

**Empirically**, we evaluate the bounds for networks of varying width $c \in \{32, 1024, 8192\}$ trained on CIFAR100. Here, we compute exact values and do not use the simplifications (1) - (7). As the models do not fit the training data, we use a margin parameter of $\gamma = 1$ for simplicity. Overall, the models performed rather poorly (as expected) with testing accuracies of 21%, 29%, 31% and training accuracies 32%, 99.8%, 99.9%.

Table 4 lists the computed values of the bounds and how they distribute over the respective factors (weight & data norms, logarithmic term, numerical constant, sample size dependency). Results are presented on a logarithmic scale with base 10. Overall, we observe the following effects:

1. Relatively, the contribution of the second layer in [26] is improved by a factor of 10. This is expected, as 10 is the square root of the number of classes in the CIFAR100 dataset.

2. The wider the network, the more dominant the term corresponding to the first layer becomes. At width 32, the factor from weight and data norms of [26] is clearly superior. This is due to the improved class dependency. However, for a width of 1024, this effect is already negligible.

3. For the wider networks ($c \in \{1024, 8192\}$), we improve over [26] by a factor of approximately $10^{1.5} \approx 30$. Ignoring numerical constants, we improve by a factor $10^{0.3} \approx 2$, which is due to an improvement in the logarithmic term.

4. For fixed width, the bounds and factors do not vary over the random initializations. For the models with widths 1024 and 8192, the standard deviation of the base 10 logarithms are $< 0.005$, which corresponds to a geometric standard deviation of less than a factor 1.01, i.e., 1%.

Last, we consider a network whose first layer has kernel size 3 and stride 1. Reduction of the spatial dimensionality is achieved by a subsequent max pooling layer of window size $32 \times 32$. This setting favors [26] as this work can better account for the pooling layer. In our simplified bound of Eq. (38), the factor $1/k$ in the first summand disappears because of the unit stride; their result improves due to the now smaller convolutional patches. In this setting the contribution of the first layer's weight and data norms is clearly larger in our bound. Yet, our bound is still smaller, but only due to numerical constants.

**Table 4:** Numerical comparison of bounds (Ours and Ledent et al.) and relevant factors computed for two-layer networks trained on CIFAR100 data (5 networks, randomly initialized, per width). Reported are the mean $\pm$ standard deviation of the base 10 logarithms, i.e., the logarithms of the geometric mean and geometric standard deviation. The top table corresponds to an architecture without pooling layers and stride 32; the bottom table corresponds to an architecture with a pooling layer of window size $32 \times 32$.

| | Width 32 | | Width 1024 | | Width 8192 | |
| --- | --- | --- | --- | --- | --- | --- |
| | **Ours** | **Ledent et al.** | **Ours** | **Ledent et al.** | **Ours** | **Ledent et al.** |
| Bound | $5.669 \pm 0.009$ | $6.704 \pm 0.009$ | $6.538 \pm 0.001$ | $7.973 \pm 0.001$ | $7.111 \pm 0.001$ | $8.657 \pm 0.001$ |
| Weight data norms | $4.760 \pm 0.009$ | $4.294 \pm 0.009$ | $5.575 \pm 0.001$ | $5.544 \pm 0.001$ | $6.121 \pm 0.001$ | $6.208 \pm 0.001$ |
| $\rightarrow$ 1st layer | $4.255 \pm 0.012$ | $4.202 \pm 0.009$ | $5.356 \pm 0.001$ | $5.507 \pm 0.001$ | $6.036 \pm 0.001$ | $6.195 \pm 0.001$ |
| $\rightarrow$ 2nd layer | $4.358 \pm 0.008$ | $2.976 \pm 0.011$ | $4.757 \pm 0.001$ | $3.661 \pm 0.002$ | $4.749 \pm 0.001$ | $3.674 \pm 0.002$ |
| Logarithmic term | $0.543 \pm 0.000$ | $0.839 \pm 0.000$ | $0.597 \pm 0.000$ | $0.858 \pm 0.000$ | $0.624 \pm 0.000$ | $0.879 \pm 0.000$ |
| Numerical constants | $1.681 \pm 0.000$ | $2.885 \pm 0.000$ | $1.681 \pm 0.000$ | $2.885 \pm 0.000$ | $1.681 \pm 0.000$ | $2.885 \pm 0.000$ |
| Sample size dependency | $-1.315 \pm 0.000$ | $-1.315 \pm 0.000$ | $-1.315 \pm 0.000$ | $-1.315 \pm 0.000$ | $-1.315 \pm 0.000$ | $-1.315 \pm 0.000$ |

| | Width 1024 (pooling) | |
| --- | --- | --- |
| | **Ours** | **Ledent et al.** |
| Bound | $7.009 \pm 0.002$ | $7.235 \pm 0.001$ |
| Weight data norms | $6.010 \pm 0.002$ | $4.803 \pm 0.001$ |
| $\rightarrow$ 1st layer | $5.823 \pm 0.002$ | $4.118 \pm 0.003$ |
| $\rightarrow$ 2nd layer | $5.408 \pm 0.002$ | $4.523 \pm 0.001$ |
| Logarithmic term | $0.544 \pm 0.000$ | $0.862 \pm 0.000$ |
| Numerical constants | $1.681 \pm 0.000$ | $2.885 \pm 0.000$ |
| Sample size dependency | $-1.315 \pm 0.000$ | $-1.315 \pm 0.000$ |

# B  Additional Experiments

## B.1  Excess capacity in non-residual networks

In addition to the experiments presented in Section 4 of the main manuscript, we performed the same excess capacity experiments on a non-residual convolutional network.

**Architecture.** Essentially, we rely on the same 11-layer convolutional network with ReLU activations as described in Section A.3, only that we substitute each convolutional layer with stride 2 by a convolutional layer with stride 1 followed by a max-pooling layer with kernel size 3 and stride 2. This is done so that we can enforce the constraints on the capacity-driving quantities via the approach described in Section B.2. Consistent with our ResNet18 experiments, the linear classifier is fixed with weights set to the vertices of a (#classes − 1) unit simplex in the output space of the network and kernel sizes of the convolutional layers are not larger than the width of their input.

**Datasets & Training.** Experiments are performed on the CIFAR10 and CIFAR100 benchmark datasets [25]. We minimize the cross-entropy loss using SGD with momentum (0.9) for 200 epochs with batch size 256 and decay the initial learning rate (of 3e-2) with a cosine annealing scheduler after each epoch. *No data augmentation is used.* For projecting onto the constraint sets, we perform one alternating projection step every 10th SGD update. After the final SGD update, we additionally do 15 alternating projection steps to ensure that the trained model is within the capacity-constrained class.

**Results.** We observe similar phenomena as for the residual (ResNet18) network studied in Section 4. When comparing models trained with and without constraint, we see a substantial amount of excess capacity, and this excess capacity increases with task difficulty. In fact, compared to our results with the residual network architecture, this effect is even more pronounced as the capacity-driving quantities in the unconstrained setting are surprisingly large. For instance, the median Lipschitz constant of the model trained on CIFAR100 is 11.53 (cf. Table 5), compared to 2.17 for the ResNet18 results in Table 1. Notably, the capacity-driving quantities can be drastically reduced without a loss of testing accuracy and the constraints can be chosen equally across datasets. This is similar to Section 4 where constraints are not precisely equal, but within a small range. We also observe another manifestation of task difficulty: tightening both constraints beyond the identified operating point leads to a more rapid deterioration of the testing error as task difficulty increases (Fig. 6, middle).

Different to Section 4, the constrained models (almost) fit the training data. However, under slightly stronger constraints, we can still find models with testing accuracy comparable (but slightly worse) to the unconstrained setting, but with noticeably less generalization gap (Fig. 6, bottom). Again, this is primarily due to leaving the zero-training-error regime. We suspect that the constraints could be much stronger, but enforcing the constraints appears to more heavily influence optimization for networks without skip connections. In this context, it is also worth pointing out that the constraints are quite strong for the non-residual network (proportionally much stronger than for the ResNet18 model in Section 4). During training, we projected after every 10th SGD step, which was actually not enough to enforce the constraints throughout the whole training procedure. Only towards the end of training, when the learning rate is already small, do the constraints become satisfied. Increasing the projection frequency might thus allow for even stronger constraints.

## B.2  Projection method

A key aspect of the experiments in Section 4 is to obtain, for each pair of Lipschitz constant and $(2, 1)$ group norm constraints, a model with testing accuracy as high as possible. The quality of such a model depends, to a large extent, on the way the constraints are enforced. This section specifies the projection method used for the experiments and provides additional background information.

As mentioned in Section 4, we utilize *orthogonal projections*. Given $x_0 \in \mathbb{R}^d$ and a nonempty closed convex set $A \subset \mathbb{R}^d$, the orthogonal projection of $x_0$ onto $A$ is defined as the unique

$$x_{\text{orth}} = \underset{x \in A}{\arg \min} \|x - x_0\|_2 \quad . \tag{40}$$

Orthogonal projections have several beneficial properties. First, if $f : \mathbb{R}^d \to \mathbb{R}$ is a strictly convex function, then, for appropriately chosen step sizes, projected gradient descent (i.e., gradient

**Table 5:** Assessment of the capacity-driving quantities for the **non-residual** 11-layer convolutional network of this section. We list the *median* over the Lipschitz constants (**Lip.**) and the $(2,1)$ group norm distances (**Dist.**) across all layers. **Err.** denotes the training/testing error, **Capacity** denotes the measures (♣, ♠) from Theorem 3.5 (adapted to the non-residual setting) and **Gap** the empirical generalization gap. The top part lists results in the unconstrained regime (see ◆ in Fig. 3), the bottom part lists results at the operating point of the most restrictive constraint combination where the testing error is on a par with the unconstrained case. **Mar.** denotes the margin parameter $\gamma$ used for computing the capacity measures, which we choose such that the unconstrained and constrained models have the same ramp loss value.

|          | **Lip.** | **Dist.** | **Mar.** | **Err. (Tst)** | **Err. (Trn)** | **Capacity (♣, ♠)** | **Gap** |
|----------|----------|-----------|----------|----------------|----------------|---------------------|---------|
| CIFAR10  | 4.66     | 370.0     | 16.4     | 0.17           | 0.00           | $1.2 \cdot 10^{12}$ / $8.1 \cdot 10^2$ | 0.17 |
| CIFAR100 | 11.53    | 854.0     | 52.1     | 0.47           | 0.00           | $1.0 \cdot 10^{16}$ / $9.2 \cdot 10^2$ | 0.47 |
| CIFAR10  | 1.80 | 200.0 | 10.0  | 0.17           | 0.00           | $6.6 \cdot 10^8$ / $6.9 \cdot 10^2$ | 0.17 |
| CIFAR100 | 1.80 | 200.0 | 10.0  | 0.47           | 0.03           | $6.6 \cdot 10^8$ / $6.9 \cdot 10^2$ | 0.43 |

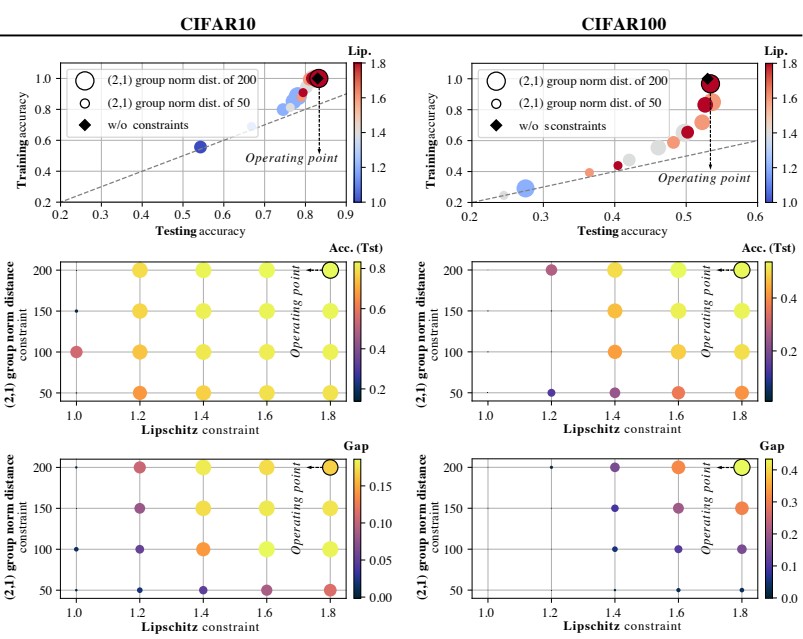

**Figure 6:** Fine-grained analysis of training/testing accuracy in relation to the Lipschitz constraint and the $(2, 1)$ group norm distance to initialization constraint for the 11-layer **non-residual** convolutional network of this section. We see that testing accuracy can be retained (relative to ◆) for a range of fairly restrictive constraints (*top row*), compared to the unconstrained regime (cf. **Lip./Dist.** columns in the top part of Table 5). However, this range noticeably narrows with increasing task difficulty (*middle row*). *Best-viewed in color.*

descent with a subsequent orthogonal projection onto $A$ after each step) converges to the minimizer $\arg\min_{x \in A} f(x)$ [7]. Second, for a tuple $(\mathcal{C}_1, \ldots \mathcal{C}_N)$ of closed convex sets $\mathcal{C}_j \subset \mathbb{R}^d$ with orthogonal projections $P_{\mathcal{C}_j}$, alternating orthogonal projections, i.e., the sequence $x_{i+1} = P_{\mathcal{C}_N} \circ \cdots \circ P_{\mathcal{C}_1}(x_i)$ converges [5] to a point in the intersection $\mathcal{C}_1 \cap \cdots \cap \mathcal{C}_N$ (if it is non-empty). Notably, there are variants of alternating orthogonal projections, e.g., Dykstra's algorithm [12], which converge to the orthogonal projection $P_{\bigcap_i \mathcal{C}_i}$ onto the intersection $\bigcap_i \mathcal{C}_i$. However, there is a key disadvantage of orthogonal projections. Being defined by the optimization problem $x_{\text{orth}} = \arg\min_{x \in A} \|x - x_0\|_2$ of Eq. (40), they often can only be computed numerically and might require a large compute budget.

In Section 4, the convex set is $\mathcal{C} = \{K \in \mathbb{R}^{c_{in} \times c_{out} \times k_h \times k_w} : \|K - K^0\|_{2,1} \leq b, \ \text{Lip}(\phi_K) \leq s\}$. The orthogonal projection onto $\mathcal{C}$ is unknown, but the alternating orthogonal projection onto the sets

$$
\begin{aligned}
\mathcal{C}_1 &= \{K \in \mathbb{R}^{c_{in} \times c_{out} \times h \times w} : \|K - K^0\|_{2,1} \leq b\}, \\
\mathcal{C}_2 &= \{K \in \mathbb{R}^{c_{in} \times c_{out} \times h \times w} : \text{Lip}(\phi_K) \leq s\}, \\
\mathcal{C}_3 &= \{K \in \mathbb{R}^{c_{in} \times c_{out} \times h \times w} : K_{ijkl} = 0 \text{ for } k > k_h, j > k_w\},
\end{aligned}
\tag{41}
$$

still defines a projection onto $\mathcal{C} \subset \mathbb{R}^{c_{in} \times c_{out} \times k_h \times k_w}$ considered as subset of $\mathbb{R}^{c_{in} \times c_{out} \times h \times w}$. Importantly, all three orthogonal projections are known. The projection onto $\mathcal{C}_1$ is due to [30]. The projection onto $\mathcal{C}_2$ requires a singular value decomposition of $M_K$, i.e., the $hwc_{in} \times hwc_{out}$ matrix corresponding to the linear map $\phi_K$. As this matrix can be quite large, this is infeasible in practice. However, [40] show that for strides $1$[4], due to the particular structure of convolutions, it suffices to compute the singular value decomposition of $hw$ matrices of size $c_{in} \times c_{out}$. Still, the computation of the projection onto $\mathcal{C}_2$ is the bottleneck of the training procedure in Section 4. The orthogonal projection onto $\mathcal{C}_3$, which is a plane, is realized by setting the corresponding coordinates to zero.

Another approach is to use *radial projections*. Given $x_0 \in \mathbb{R}^d$ and a norm $\|\cdot\|$, the radial projection of $x_0$ onto the $\|\cdot\|$-ball $B(r, y, \|\cdot\|)$ of radius $r$ centered at $y$, is defined as

$$
x_{\text{rad}} = x_0 - \left(1 - \frac{r}{\|x_0 - y\|}\right)(x_0 - y) \mathbb{1}_{\|x_0 - y\| > r} .
\tag{42}
$$

Such a projection is called radial, as it translates the point $x_0$ in radial direction w.r.t. the ball $B(r, y, \|\cdot\|)$ such that it lands on the boundary (if it is not already in $B(r, y, \|\cdot\|)$).

Notably, $\mathcal{C}_1$ and $\mathcal{C}_2$ are both balls, one with respect to the $(2, 1)$ group norm, the other with respect to the spectral norm of the matrix $M_K$ associated to $K$. Importantly, the spectral norm can be easily estimated by the power method for convolutional layers [15, 27], so radial projections have far less computational overhead. However, alternating radial projections are not guaranteed[5] to converge to a point in the intersection $\mathcal{C}$. Furthermore, by definition, we have $\|x - x_{\text{rad}}\|_2 \geq \|x - x_{\text{orth}}\|_2$, so we expect radial projections to yield inferior results (w.r.t. the constraint strengths that can be enforced).

We evaluated three different approaches to obtain models with constrained capacity: (1) training with a variant of projected gradient descent, where we perform one alternating orthogonal projection step after every 15th SGD update; (2) performing one alternating radial projection step after every SGD update[6]; (3) orthogonal projection onto $\mathcal{C}$ of an already trained unconstrained model, using 100 iterations of Dykstra's algorithm. Our findings are summarized in Fig. 7. We see that, alternating orthogonal projections during training allow for the strongest constraints, without a drop in the testing accuracy. This is expected, because they divert the weights less from the training trajectory than radial projections. By the same logic it is obvious that projecting only at the *end* of training is not feasible, as the weights of the trained network are already too far away from the constraint set. We conclude that alternating orthogonal projections allow for the best estimate of excess capacity.

### B.3 Comparison between constrained and unconstrained models beyond testing error

So far, we have analyzed to which extent the weights of neural networks can be constrained without a loss of testing accuracy. In particular, we have identified the maximal constraint strength (i.e., the operating point) such that the testing error of the constrained models is on a par with the one of unconstrained models. However, this does not imply that constrained models and unconstrained ones can be used interchangeably, as they might differ in other aspects. In this section, we will study how pronounced such differences are with respect to (i) biases to particular classes, (ii) susceptibility to adversarial attacks, and (iii) compressibility in terms of the number of weights (via weight pruning).

For the evaluation, we use **25 unconstrained** and **25 constrained** models trained on CIFAR100, with the same architecture and optimization hyperparameters as listed in the main text, i.e., Section 4. As constraint strength, we choose a layer-wise Lipschitz constant of 0.8 and a distance constraint of 70.

---

[4]Extensions to strides >1 are not straightforward but seem possible.
[5]empirically, we still observed convergence
[6]The increased projection frequency is possible because of the reduced computational overhead of radial projections compared to orthogonal projections.

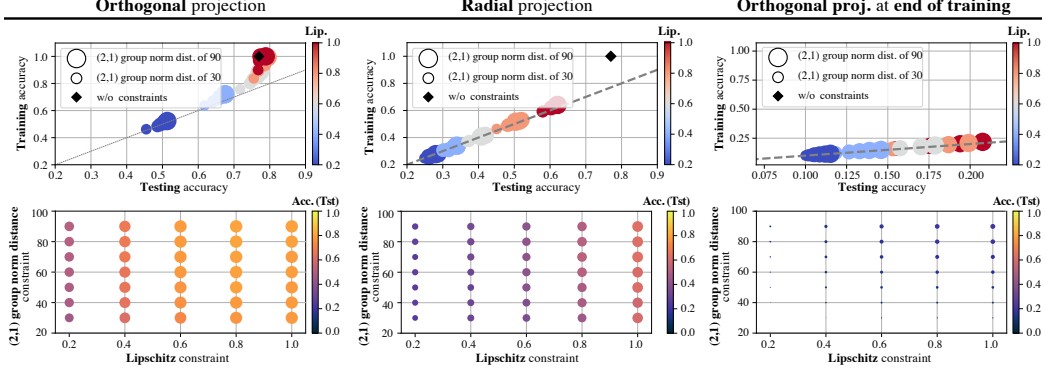

**Figure 7:** Comparison of different *projection techniques* for ResNet18 models trained on CIFAR10. *Best-viewed in color.*

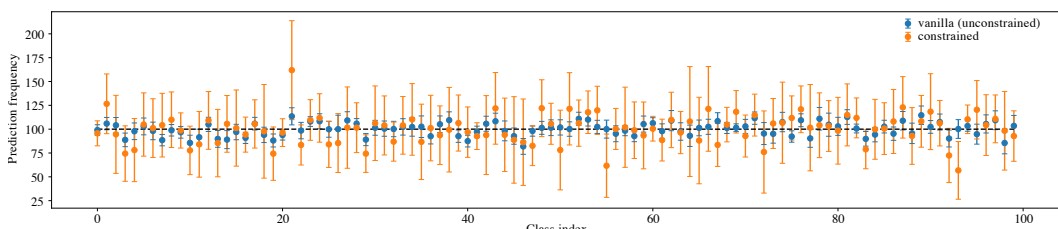

**Figure 8:** Prediction frequencies per class index on the testing portion of CIFAR100, averaged over the predictions of 25 constrained and 25 unconstrained (vanilla) ResNet18 models. *Best-viewed in color.*

### B.3.1 Biases to particular classes

For each of the 50 models, we counted how often each class is predicted on the testing data, which consists of 100 images per class. If there are no biases to particular classes, the counts should be distributed around this value with preferably small spread. Fig. 8 visualizes the results. We immediately see, that for the unconstrained models, the predictions per class are more uniformly distributed with the average class close to 100 and small standard deviations. In contrast, for the constrained models, the standard deviations are much larger. Most striking is the peak at class index 21, indicating that the constrained models are indeed biased to this particular class (chimpanzee). In fact, only one of the 25 models predicted this class less than 100 times. Notably, the unconstrained models are also biased towards this class, as the error region of (mean $\pm$ standard deviation) does not contain 100. Overall, there are more favored/disfavored classes for the unconstrained models (32 *vs.* only 5 for the constrained models), but for the constrained models, the biases are more pronounced.

### B.3.2 Susceptibility to adversarial attacks

We tested several adversarial attacks (FGSM, FGM, L2PGD, LinfPGD, L2DeepFool, L2AdditiveGaussianNoise, L2AdditiveUniformNoise, L2ContrastReduction, GaussianBlur) using the `foolbox` [39] Python package. To compare constrained *vs.* unconstrained models, we extract 1024 images from the testing data, which are correctly classified by all 50 models (25 constrained models, 25 unconstrained models) on which the attacks are evaluated. The fraction of correctly classified images for increasing attack strengths is visualized in Fig. 9. As can be seen from the figure, constrained models are less susceptible to the gradient-based attacks FGSM, FGM, L2PGD, and LinfPGD. For L2DeepFool, contrast reduction (L2ContrastReductionAttack) and Gaussian blur (GaussianBlurAttack), constrained and unconstrained models are equally affected. To our surprise, the constrained models more vulnerable to additive Gaussian and uniform noise.

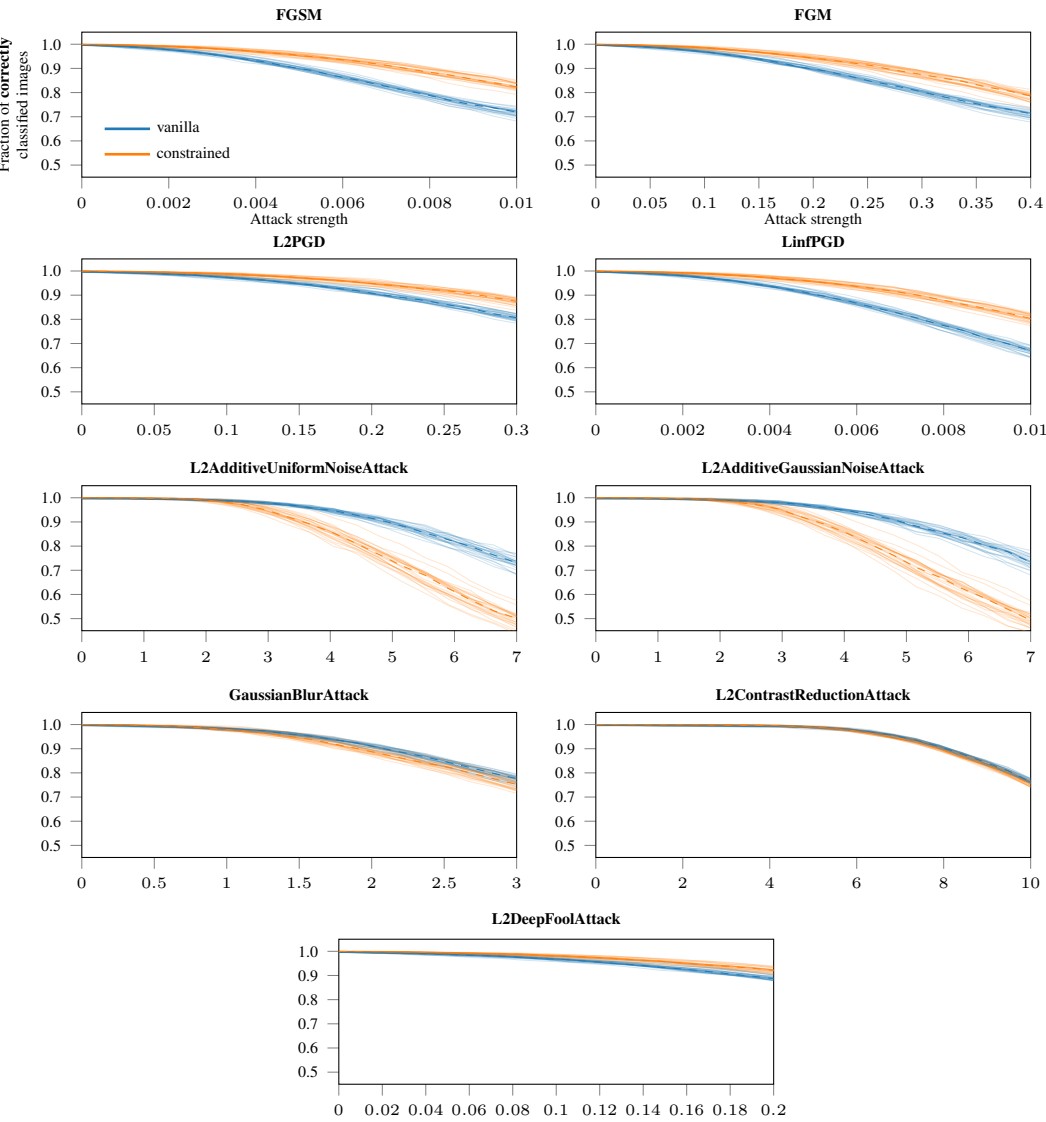

**Figure 9:** Results of running several adversarial attacks against constrained and unconstrained ResNet18 models on CIFAR100 across varying attack strengths ($x$-axis). Each solid line represents one trained model, the dashed lines represent the medians. The $y$-axis shows the fraction of correctly classified images that remain correct as attack strength increases. *Best-viewed in color.*

### B.3.3 Compressibility via weight pruning

We measure compressibility for global (unstructured) $l_1$ weight pruning. This simple pruning technique identifies a predefined fraction of a model's parameters (i.e., elements of the weight tensors/matrices) and sets them to zero. *No subsequent fine-tuning steps were performed.*

As illustrated in Fig. 10, unconstrained models are less affected by this pruning technique than constrained models. Yet, constrained models can still be pruned, and, in the range of pruning strengths 0 to 0.15, the median testing accuracy drops only marginally for both model types. Only at larger pruning strengths do differences between the model types become visible. If we consider a median testing accuracy of 46%, resp. 45%, to be acceptable, then this threshold allows for pruning 20%, resp. 30%, of the weights of constrained models and 30%, resp 40%, of the unconstrained models. Of course, the pruned models might not satisfy the constraints anymore. In particular, the distance constraint gets violated, as with increasing constraint strength, the distance to initialization converges to the norm of the initialization. Typically, the latter is already larger than the distance constraint (of 70). Thus, when combining

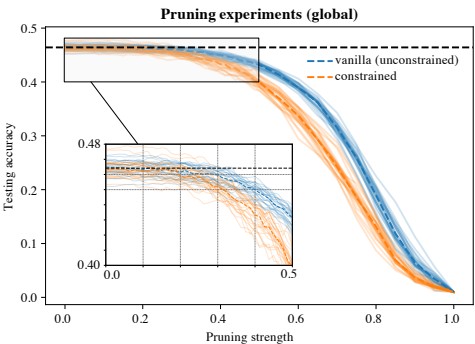

**Figure 10:** Results of global (unstructured) $l_1$ weight pruning on 25 constrained and 25 unconstrained ResNet18 models. Each solid line represents one trained model. *Best-viewed in color.*

norm constraints with weight pruning, constraining the distance with respect to the zero weight appears to be more sensible. This can be done, as the distance constraint in Theorem 3.5 is not required to be with respect to the initialization; it can be chosen relative to any reference weight as long as the reference weight does not depend on the training data.

### B.4 Hardware resources

All experiments were run on an Ubuntu Linux 20.04.4 LTS system with 128 GB of main memory, an Intel®Core™ i9-10980XE processor and two NVIDIA GeForce RTX 3090 graphics cards (24 GB memory, CUDA 11.4, driver version 470.129.06). All models are implemented in Pytorch (v1.10).

# C Proofs

In the following sections, we present proofs for the theoretical results listed in the manuscript as well as additional supplementary results.

## C.1 Preliminaries

In terms of notation, we consider spaces $\mathcal{F}$ of functions $f \colon (\mathcal{X}, \|\cdot\|_{\mathcal{X}}) \to (\mathcal{Y}, \|\cdot\|_{\mathcal{Y}})$ between normed spaces. We write

$$\mathrm{Lip}(f) = \sup_{x_1, x_2 \in \mathcal{X}} \frac{\|f(x_1) - f(x_2)\|_{\mathcal{Y}}}{\|x_1 - x_2\|_{\mathcal{X}}}$$

and

$$\mathrm{Lip}(\mathcal{F}) = \sup_{f \in \mathcal{F}} \mathrm{Lip}(f)$$

for the Lipschitz constant of $f$ and the supremal Lipschitz constant of $\mathcal{F}$, respectively. For the remainder of the section, all function spaces $\mathcal{F}$ will have bounded Lipschitz constants $\mathrm{Lip}(\mathcal{F}) < \infty$. Such function spaces are vector spaces, where addition and scalar multiplication are defined pointwise via the vector space structure on $\mathcal{Y}$, i.e., $(f + g) \colon x \mapsto f(x) + g(x)$ and $(\alpha f) \colon x \mapsto \alpha f(x)$.

We equip $\mathcal{F}$ with a *data-dependent norm*, defined below.

**Definition C.1.** Let $\mathcal{F}$ be a space of functions $f : (\mathcal{X}, \|\cdot\|_{\mathcal{X}}) \to (\mathcal{Y}, \|\cdot\|_{\mathcal{Y}})$ between normed spaces. The *data-dependent norm*, denoted as $\|\cdot\|_X$, on $\mathcal{F}$ restricted to $X = (x_1 \ldots, x_n) \in \mathcal{X}^n$, i.e., $\mathcal{F}|_X$, is defined as

$$\|f\|_X = \sqrt{\sum_{i=1}^n \|f(x_i)\|_{\mathcal{Y}}^2} \ . \tag{43}$$

**Remark C.2.** This norm is a seminorm on $\mathcal{F}$ and a norm on $\mathcal{F}|_X = \{f|_{\{x_1,\ldots,x_n\}} : f \in \mathcal{F}\}$. If $f \in \mathcal{F}$ has norm $\|f\|_X = 0$, then it holds that $\forall i : f(x_i) = 0$. Thus, $f$ is the zero element in $\mathcal{F}|_X$, but not necessarily the zero element in $\mathcal{F}$, as there might exist $v \in \mathcal{X} \setminus \{x_1, \ldots, x_n\}$ with $f(v) \neq 0$.

Two fundamental properties concerning compositions of functions are worth pointing out:

$$\|f \circ g\|_X = \|f\|_{g(X)} \qquad \text{and} \qquad \|f \circ g - f \circ h\|_X \leq \mathrm{Lip}(f) \|g - h\|_X \ . \tag{44}$$

Further, we recall the definition of *covering numbers*.

**Definition C.3.** Let $(\mathcal{H}, \|\cdot\|)$ be a normed space, $S \subset \mathcal{H}$ and $\epsilon > 0$. We call any subset $U \subset S$ an internal $\epsilon$-cover of $S$ if for every $s \in S$ there exists $u \in U$ such that $\|s - u\| \leq \epsilon$. The internal covering number $\mathcal{N}^{\mathrm{int}}(\mathcal{H}, \epsilon, \|\cdot\|)$ is the cardinality of the smallest internal $\epsilon$-cover of $S$, i.e.,

$$\mathcal{N}^{\mathrm{int}}(S, \epsilon, \|\cdot\|) = \min\left(\{|U| : \text{U is an internal } \epsilon\text{-cover of } S\}\right) \tag{45}$$

Dropping the requirement $U \subset S$, we analogously define external $\epsilon$-covers $U \subset \mathcal{H}$ and external covering numbers $\mathcal{N}^{\mathrm{ext}}(S, \epsilon, \|\cdot\|)$.

In the manuscript, if not stated otherwise, covers will *always be internal* and covering numbers will be denoted as $\mathcal{N} = \mathcal{N}^{\mathrm{int}}$.

Internal and external covering numbers are related via the following chain of inequalities:

$$\mathcal{N}^{\mathrm{ext}}(S, \epsilon, \|\cdot\|) \leq \mathcal{N}^{\mathrm{int}}(S, \epsilon, \|\cdot\|) \leq \mathcal{N}^{\mathrm{ext}}(S, \epsilon/2, \|\cdot\|) \ . \tag{46}$$

The first inequality follows directly from the definition of covering numbers; the second one follows from the triangle inequality. Furthermore, for any subset $T \subset S$, it holds that

$$\mathcal{N}^{\mathrm{ext}}(T, \epsilon, \|\cdot\|) \leq \mathcal{N}^{\mathrm{ext}}(S, \epsilon, \|\cdot\|) \ . \tag{47}$$

Notably, this is not true for internal coverings. For example, the unit ball in $\mathbb{R}^d$ defines an internal cover of itself, whereas an annulus cannot be covered internally with only one ball of radius 1.

## C.2 Single-layer covering number bounds

This section contains the covering number bounds for single convolutional layers. For simplicity, we will first present the special case of the single-layer covering number bound for **1D convolutions** with one channel, stride 1, odd kernel size and input size preserving (zero) padding. The proof of the general case then follows along the same line of arguments, but is more tedious, due to the additional notation and subindices.

Let $X = (x_1, \ldots, x_n) \in \mathcal{X}^n$ with $x_i \in \mathcal{X} = \mathbb{R}^h$. Further, let $K \in \mathbb{R}^k$, with $k$ odd, be a convolutional kernel and $\phi_K : \mathbb{R}^h \to \mathbb{R}^h$ the corresponding convolutional map, which is defined coordinate-wise as

$$[\phi_K(x)]_i = \sum_{\alpha=-\frac{k-1}{2}}^{\frac{k-1}{2}} K_\alpha \, x_{i+\alpha} \mathbb{1}_{i+\alpha\in[1,h]} \ . \tag{48}$$

For the norm of the data $X$, we write

$$\|X\| = \sqrt{\sum_{i=1}^{n} \|x_i\|^2} \ . \tag{49}$$

Our covering number bounds hinge on the seminal Maurey sparsification lemma. We state one variant, see [3, Lemma A.6].

**Lemma C.4** (Maurey sparsification lemma). *Fix a Hilbert space $\mathcal{H}$ with norm $\|\cdot\|$. Let $U \in \mathcal{H}$ be given with representation $U = \sum_{i=1}^{d} \alpha_i V_i$ where $V_i \in \mathcal{H}$, $\alpha \in \mathbb{R}_{\geq 0}^d \setminus \{0\}$ and $\sum_i |\alpha_i| \leq 1$. Then, for any positive integer $m$, there exists a choice of non-negative integers $(m_1, \ldots, m_d)$ with $\sum_i m_i = m$, such that*

$$\left\| U - \frac{1}{m} \sum_{i=1}^{d} m_i V_i \right\|^2 \leq \frac{1}{m} \max_{i=1,\ldots,d} \|V_i\|^2 \ . \tag{50}$$

**Theorem C.5** (Single-layer covering number bound – Simple 1D variant). *Let $b > 0$, $k$ odd and let $\mathcal{F} = \{\phi_K \mid K \in \mathbb{R}^k, \|K\|_1 \leq b\}$ be the set of 1D convolutions determined by kernels $K \in \mathbb{R}^k$ with $\|K\|_1 \leq b$. For any $X = (x_1, \ldots, x_n) \in \mathcal{X}^n = \mathbb{R}^{n \times h}$ and $\epsilon > 0$, the covering number $\mathcal{N}(\mathcal{F}, \epsilon, \|\cdot\|_X)$ satisfies*

$$\log \mathcal{N}(\mathcal{F}, \epsilon, \|\cdot\|_X) \leq \left\lceil \frac{\|X\|^2 b^2}{\epsilon^2} \right\rceil \log(2k) \tag{51}$$

*and*

$$\log \mathcal{N}(\mathcal{F}, \epsilon, \|\cdot\|_X) \leq (2k-1) \log\left(1 + \left\lceil \frac{\|X\|^2 b^2}{\epsilon^2} \right\rceil\right) \ . \tag{52}$$

*Proof.* We rewrite the coordinate-wise definition of the 1D convolutional operation from Eq. (48) with unit stride as

$$[\phi_K(x)]_i = \sum_{j=1}^{h} \underbrace{\left( \sum_{\alpha=-\frac{k-1}{2}}^{\frac{k-1}{2}} \mathbb{1}_{j=\alpha+i} K_\alpha \right)}_{=M_{ij}} x_j = \sum_{j=1}^{h} M_{ij} x_j \ .$$

Thus, convolution is a linear map parametrized by a matrix $M \in \mathbb{R}^{h \times h}$ with entries

$$M_{ij} = \sum_{\alpha=-\frac{k-1}{2}}^{\frac{k-1}{2}} \mathbb{1}_{j=\alpha+i} K_\alpha \ .$$

In particular, we can write $M = \sum_{\alpha=-(k-1)/2}^{(k-1)/2} K_\alpha M^{(\alpha)}$ with $M_{ij}^{(\alpha)} = \mathbb{1}_{j-i=\alpha}$ and note that for every $x \in \mathbb{R}^h$ and every $\alpha$, we have

$$\left\| M^{(\alpha)} x \right\| \leq \|x\| \ .$$

For example, if $h = 5$ and $k = 3$, we have

$$M = \begin{pmatrix} K_0 & K_1 & 0 & 0 & 0 \\ K_{-1} & K_0 & K_1 & 0 & 0 \\ 0 & K_{-1} & K_0 & K_1 & 0 \\ 0 & 0 & K_{-1} & K_0 & K_1 \\ 0 & 0 & 0 & K_{-1} & K_0 \end{pmatrix} ,$$

$$M^{(-1)} = \begin{pmatrix} 0 & 0 & 0 & 0 & 0 \\ 1 & 0 & 0 & 0 & 0 \\ 0 & 1 & 0 & 0 & 0 \\ 0 & 0 & 1 & 0 & 0 \\ 0 & 0 & 0 & 1 & 0 \end{pmatrix}, \quad M^{(0)} = \begin{pmatrix} 1 & 0 & 0 & 0 & 0 \\ 0 & 1 & 0 & 0 & 0 \\ 0 & 0 & 1 & 0 & 0 \\ 0 & 0 & 0 & 1 & 0 \\ 0 & 0 & 0 & 0 & 1 \end{pmatrix}, \quad M^{(1)} = \begin{pmatrix} 0 & 1 & 0 & 0 & 0 \\ 0 & 0 & 1 & 0 & 0 \\ 0 & 0 & 0 & 1 & 0 \\ 0 & 0 & 0 & 0 & 1 \\ 0 & 0 & 0 & 0 & 0 \end{pmatrix} .$$

In summary, we have

$$\phi_K(x) = \sum_{\alpha = -(k-1)/2}^{(k-1)/2} K_\alpha M^{(\alpha)} x = \sum_{\alpha = -(k-1)/2}^{(k-1)/2} \frac{K_\alpha}{b} b M^{(\alpha)} x .$$

By assumption, we have $\sum_\alpha \left| \frac{K_\alpha}{b} \right| = \frac{\|K\|_1}{b} \leq 1$ and so we can instantiate Maurey's sparsification lemma (Lemma C.4) on the Hilbert space $(\mathcal{F}|_X, \|\cdot\|_X)$ for $\{V_1, \ldots V_{2d}\} = \{x \mapsto \pm b M^{(\alpha)} x | \, \alpha = 1, \ldots, d\} \subset \mathcal{F}$. As a consequence, for any convolutional kernel $K \in \mathbb{R}^k$ and any $m \in \mathbb{N}$, there exist $(m_1, \ldots, m_{2d})$ with $\sum_{i=1}^{2d} m_i = m$ such that

$$\left\| \phi_K - \frac{1}{m} \sum_{i=1}^{2d} m_i V_i \right\|_X^2 \leq \frac{1}{m} \max_{i=1\ldots,2d} \|V_i\|_X^2 .$$

Thus, for fixed $\epsilon > 0$, if we choose $m \in \mathbb{N}$ such that $\frac{1}{m} \max_i \|V_i\|_X^2 \leq \epsilon^2$, then the solutions (in $m_i$), of $\sum_{i=1}^{2d} m_i = m$, define an $\epsilon$-cover

$$\left\{ \frac{1}{m} \sum_{i=1}^{2d} m_i V_i | \, m_i \in \mathbb{N}_{\geq 0}, \, \sum_{i=1}^{2d} m_i = m \right\} \subset \mathcal{F} .$$

As the number of non-negative $2d$-tuples that add up to $m$ is equal to[7]

$$N(m, d) = \binom{m + 2d - 1}{2d - 1} ,$$

this means that $\mathcal{F}$ has an $\epsilon$-cover of cardinality at most $N(m, d)$, and so $\mathcal{N}(\mathcal{F}, \epsilon, \|\cdot\|_X) \leq N(m, d)$.

Since for all $i \in \{1, \ldots, 2d\}$, the norms $\|V_i\|_X^2$ satisfy

$$\|V_i\|_X^2 = \left\| b M^{(\alpha_i)} \right\|_X^2 = b^2 \sum_{j=1}^{n} \left\| M^{(\alpha_i)} x_j \right\|^2 \leq b^2 \sum_{j=1}^{n} \|x_j\|^2 = b^2 \|X\|^2 ,$$

we can choose $\mathbb{N} \ni m = \left\lceil \frac{b^2 \|X\|^2}{\epsilon^2} \right\rceil$.

The theorem then follows from two particular bounds on $N(m, d)$, see Lemma C.24. These are

$$\binom{m + 2d - 1}{2d - 1} \leq (2d)^m ,$$

which implies Eq. (51), and

$$\binom{m + 2d - 1}{2d - 1} \leq (1 + m)^{2d-1} ,$$

which implies Eq. (52). $\qquad \square$

---

[7]This number equals the number of possibilities to separate $m$ objects by $2d - 1$ delimiters. This corresponds to choosing $2d - 1$ elements (position of delimiters) from a set of $m + 2d - 1$ elements (objects + delimiters).

**Remark C.6.** The definition of the convolution operation in Eq. (48) corresponds to convolutional layers with zero-padding, such that the dimensionality $h$ of the data remains unchanged (i.e., input-size preserving). The covering bound equally holds for other types of padding, corresponding to other matrices $M^{(\alpha)}$, as long as

$$\left\| M^{(\alpha)} x \right\| \leq x$$

for all $x$. In particular, it holds for convolutional layers with circular padding, where the matrices $M^{(\alpha)}$ become *permutation matrices*.

Next, we study the general case of 2D multi-channel convolutions with strides. To that end, let $\phi_K$ be the map determined by a weight tensor $K \in \mathbb{R}^{c_{out} \times c_{in} \times k_h \times k_w}$, where $c_{out}, c_{in}$ denote the number of output and input channels, resp., and $(k_h, k_w)$ is the spatial extension of the kernel. For input images $x \in \mathbb{R}^{c_{in} \times h \times w}$, convolution $\phi_{K,(s_h,s_w)} : \mathbb{R}^{c_{in} \times h \times w} \to \mathbb{R}^{c_{in} \times \lceil h/s_h \rceil \times \lceil w/s_w \rceil}$ with strides $(s_h, s_w)$ is defined coordinate-wise as

$$[\phi_{K,(s_h,s_w)}(x)]_{\sigma\mu\nu} = \sum_{r=1}^{c_{in}} \sum_{i=\lfloor -\frac{k_h-1}{2} \rfloor}^{\lfloor \frac{k_h-1}{2} \rfloor} \sum_{j=\lfloor -\frac{k_w-1}{2} \rfloor}^{\lfloor \frac{k_w-1}{2} \rfloor} K_{\sigma r i j} \, x_{r,1+s_h(\mu-1)+i,1+s_w(\nu-1)+j} \tag{53}$$
$$\cdot \mathbb{1}_{1+s_h(\mu-1)+i \in [1,h]} \mathbb{1}_{1+s_w(\nu-1)+j \in [1,w]}$$

**Theorem C.7** (Single-layer covering number bound – General case). *Let $b > 0$. Define the class of $(s_h, s_w)$-strided 2D convolutions parametrized by tensors $K \in \mathbb{R}^{c_{out} \times c_{in} \times k_h \times k_w}$ with $W = c_{out} c_{in} k_h k_w$ parameters and $(2,1)$ group norm $\|K\|_{2,1} \leq b$ as*

$$\mathcal{F} = \{\phi_{K,(s_h,s_w)} \mid K \in \mathbb{R}^{c_{out} \times c_{in} \times k_h \times k_w}, \|K\|_{2,1} \leq b\} \ .$$

*Then, for any $X = (x_1, \dots, x_n) \in \mathbb{R}^{n \times c_{in} \times h \times w}$ and $\epsilon > 0$, the covering number $\mathcal{N}(\mathcal{F}, \epsilon, \|\cdot\|_X)$ satisfies*

$$\log \mathcal{N}(\mathcal{F}, \epsilon, \|\cdot\|_X) \leq \left\lceil \frac{\|X\|^2 b^2}{\epsilon^2} \right\rceil \log(2W) \tag{54}$$

*and*

$$\log \mathcal{N}(\mathcal{F}, \epsilon, \|\cdot\|_X) \leq (2W - 1) \log \left( 1 + \left\lceil \frac{\|X\|^2 b^2}{\epsilon^2} \right\rceil \right) \ . \tag{55}$$

**Remark C.8.** Recall that convolution with kernel size 1 and input size 1 is a linear map on the input channels, determined by the matrix $M_K = K_{..11}$. In this situation, the convolutional layer reduces to a fully-connected layer and our first bound reduces to [3, Lemma 3.2].

*Proof.* The proof is quite similar to the one of the special case in Theorem C.5. Recall, that the convolution operation is defined coordinate-wise in Eq. (53). Using identities of the form $\mathbb{1}_{t \in \{1,\dots,n\}} = \sum_{i=1}^{n} \mathbb{1}_{i=t}$, we write

$$[\phi_{K,(s_h,s_w)}(x)]_{\sigma\mu\nu} =$$

$$\sum_{\alpha=1}^{c_{in}} \sum_{\beta=1}^{h} \sum_{\gamma=1}^{w} x_{\alpha\beta\gamma} \left( \sum_{p=1}^{c_{out}} \sum_{r=1}^{c_{in}} \sum_{i=\lfloor -\frac{k_h-1}{2} \rfloor}^{\lfloor \frac{k_h-1}{2} \rfloor} \sum_{j=\lfloor -\frac{k_w-1}{2} \rfloor}^{\lfloor \frac{k_w-1}{2} \rfloor} K_{prij} \underbrace{\mathbb{1}_{p=\sigma} \mathbb{1}_{\alpha=r} \mathbb{1}_{\beta=1+s_h(\mu-1)+i} \mathbb{1}_{\gamma=1+s_w(\nu-1)+j}}_{=[M^{(p,r,i,j)}]_{\sigma\mu\nu}^{\alpha\beta\gamma}} \right)$$

and condense this into

$$[\phi_{K,(s_h,s_w)}(x)]_{\sigma\mu\nu} = \sum_{\alpha=1}^{c_{in}} \sum_{\beta=1}^{h} \sum_{\gamma=1}^{w} x_{\alpha\beta\gamma} \underbrace{\left( \sum_{p=1}^{c_{out}} \sum_{r=1}^{c_{in}} \sum_{i=\lfloor -\frac{k_h-1}{2} \rfloor}^{\lfloor \frac{k_h-1}{2} \rfloor} \sum_{j=\lfloor -\frac{k_w-1}{2} \rfloor}^{\lfloor \frac{k_w-1}{2} \rfloor} K_{prij} [M^{(p,r,i,j)}]_{\sigma\mu\nu}^{\alpha\beta\gamma} \right)}_{=M_{\sigma\mu\nu}^{\alpha\beta\gamma}}$$

$$= \sum_{\alpha=1}^{c_{in}} \sum_{\beta=1}^{h} \sum_{\gamma=1}^{w} x_{\alpha\beta\gamma} M_{\sigma\mu\nu}^{\alpha\beta\gamma} \ .$$

Thus, the convolution is a multilinear map $\mathbb{R}^{c_{in} \times h \times w} \to \mathbb{R}^{c_{in} \times \lceil h/s_h \rceil \times \lceil w/s_w \rceil}$ parametrized by

$$M = \sum_{p=1}^{c_{out}} \sum_{r=1}^{c_{in}} \sum_{i=\lfloor -\frac{k_h-1}{2} \rfloor}^{\lfloor \frac{k_h-1}{2} \rfloor} \sum_{j=\lfloor -\frac{k_w-1}{2} \rfloor}^{\lfloor \frac{k_w-1}{2} \rfloor} K_{prij} M^{(p,r,i,j)}$$

$$= \sum_{p=1}^{c_{out}} \sum_{r=1}^{c_{in}} \sum_{i=\lfloor -\frac{k_h-1}{2} \rfloor}^{\lfloor \frac{k_h-1}{2} \rfloor} \sum_{j=\lfloor -\frac{k_w-1}{2} \rfloor}^{\lfloor \frac{k_w-1}{2} \rfloor} \left( K_{prij} \frac{\|X_{\cdot r \cdot \cdot}\|}{\|X\| b} \right) \left( \frac{\|X\| b}{\|X_{\cdot r \cdot \cdot}\|} M^{(p,r,i,j)} \right) \quad .$$

Since

$$\left\| K_{prij} \frac{\|X_{\cdot r \cdot \cdot}\|}{\|X\| b} \right\|_1 = \frac{1}{\|X\| b} \sum_{p=1}^{c_{out}} \sum_{i=\lfloor -\frac{k_h-1}{2} \rfloor}^{\lfloor \frac{k_h-1}{2} \rfloor} \sum_{j=\lfloor -\frac{k_w-1}{2} \rfloor}^{\lfloor \frac{k_w-1}{2} \rfloor} \left( \sum_{r=1}^{c_{in}} |K_{prij}| \, \|X_{\cdot r \cdot \cdot}\| \right)$$

$$\leq \frac{1}{\|X\| b} \sum_{p=1}^{c_{out}} \sum_{i=\lfloor -\frac{k_h-1}{2} \rfloor}^{\lfloor \frac{k_h-1}{2} \rfloor} \sum_{j=\lfloor -\frac{k_w-1}{2} \rfloor}^{\lfloor \frac{k_w-1}{2} \rfloor} \left( \sum_{r=1}^{c_{in}} |K_{prij}|^2 \right)^{1/2} \underbrace{\left( \sum_{r=1}^{c_{in}} \|X_{\cdot r \cdot \cdot}\|^2 \right)^{1/2}}_{=\|X\|}$$

$$= \frac{1}{b} \sum_{p=1}^{c_{out}} \sum_{i=\lfloor -\frac{k_h-1}{2} \rfloor}^{\lfloor \frac{k_h-1}{2} \rfloor} \sum_{j=\lfloor -\frac{k_w-1}{2} \rfloor}^{\lfloor \frac{k_w-1}{2} \rfloor} \left( \sum_{r=1}^{c_{in}} |K_{prij}|^2 \right)^{1/2}$$

$$= \frac{\|K\|_{2,1}}{b} \overset{\text{(by assumption)}}{\leq} 1 \quad,$$

we can instantiate Maurey's sparsification lemma (Lemma C.4) on the Hilbert space $(\mathcal{F}|_X, \|\cdot\|_X)$ for

$$\{V_1, \dots, V_{2W}\} = \left\{ \pm \frac{\|X\| b}{\|X_{\cdot r \cdot \cdot}\|} M^{(p,r,i,j)} \,\middle|\, p \in \{1, \dots, c_{out}\}, r \in \{1 \dots, c_{in}\}, \right.$$

$$i \in \left\{ \left\lfloor -\frac{k_h-1}{2} \right\rfloor, \dots, \left\lfloor \frac{k_h-1}{2} \right\rfloor \right\},$$

$$\left. j \in \left\{ \left\lfloor -\frac{k_w-1}{2} \right\rfloor, \dots, \left\lfloor \frac{k_w-1}{2} \right\rfloor \right\} \right\} \quad.$$

As a consequence, for any convolutional kernel $K \in \mathbb{R}^{c_{out} \times c_{in} \times k_h \times k_w}$ and any $m \in \mathbb{N}$, there exist $(m_1, \dots, m_{2W})$ with $\sum_{i=1}^{2W} m_i = m$ such that

$$\left\| \phi_{K,(s_h,s_w))} - \frac{1}{m} \sum_{i=1}^{2W} m_i V_i \right\|_X^2 \leq \frac{1}{m} \max_i \|V_i\|_X^2 \quad.$$

Thus, for fixed $\epsilon > 0$, if we choose $m \in \mathbb{N}$ such that $\frac{1}{m} \max_i \|V_i\|_X^2 \leq \epsilon^2$, then the solutions in $m_i$ of $\sum_{i=1}^{2W} m_i = m$, define an $\epsilon$-cover

$$\left\{ \frac{1}{m} \sum_{i=1}^{2W} m_i V_i \,\middle|\, m_i \in \mathbb{N}_{\geq 0}, \ \sum_{i=1}^{2W} m_i = m \right\} \subset \mathcal{F} \quad. \tag{56}$$

As the number of non-negative $2W$-tuples which add up to $m$, denoted as $N(m, W)$, is equal to

$$N(m, W) = \binom{m + 2W - 1}{2W - 1} \quad,$$

this means that $\mathcal{F}$ has an $\epsilon$-cover of cardinality at most $N(m, W)$; thus, $\mathcal{N}(\mathcal{F}, \epsilon, \|\cdot\|_X) \leq N(m, W)$.

In order to compute the norms $\|V_i\|_X^2$, we use that for all $(p, r, i, j) \in [c_{out}] \times [c_{in}] \times [k_h] \times [k_w]$ and for all $x \in \mathbb{R}^{c_{in} \times h \times w}$, it holds that

$$
\left\| M^{(p,r,i,j)} x \right\|^2 = \sum_{\sigma,\mu,\nu} [M^{(p,r,i,j)} x]_{\sigma\mu\nu}^2
$$

$$
= \sum_{\sigma,\mu,\nu} \left( \sum_{\alpha=1}^{c_{in}} \sum_{\beta=1}^{h} \sum_{\gamma=1}^{w} [M^{(p,r,i,j)}]_{\sigma\mu\nu}^{\alpha\beta\gamma} x_{\alpha\beta\gamma} \right)^2
$$

$$
= \sum_{\sigma,\mu,\nu} \left( \sum_{\alpha=1}^{c_{in}} \sum_{\beta=1}^{h} \sum_{\gamma=1}^{w} \mathbb{1}_{p=\sigma} \mathbb{1}_{\alpha=r} \mathbb{1}_{\beta=1+s_h(\mu-1)+i} \mathbb{1}_{\gamma=1+s_w(\nu-1)+j} x_{\alpha\beta\gamma} \right)^2
$$

$$
= \sum_{\sigma,\mu,\nu} \left( \mathbb{1}_{p=\sigma} x_{r,1+s_h(\mu-1)+i,1+s_w(\nu-1)+j} \right)^2
$$

$$
= \sum_{\mu=1}^{\lceil h/s_h \rceil} \sum_{\nu=1}^{\lceil w/s_w \rceil} \left( x_{r,1+s_h(\mu-1)+i,1+s_w(\nu-1)+j} \right)^2
$$

$$
= \sum_{\beta=1}^{h} \sum_{\gamma=1}^{w} \left( x_{r\beta\gamma} \right)^2 \mathbb{1}_{\beta \equiv (1+i) \bmod s_h} \mathbb{1}_{\gamma \equiv (1+j) \bmod s_w}
$$

$$
\leq \sum_{\beta=1}^{h} \sum_{\gamma=1}^{w} \left( x_{r\beta\gamma} \right)^2 = \|x_{r\cdot\cdot}\|^2 \quad .
$$

Thus, for any $t \in \{1, \ldots, 2W\}$,

$$
\|V_t\|_X^2 = \left\| \pm \frac{\|X\| b}{\|X_{\cdot r\cdot\cdot}\|} M^{(p_t, r_t, i_t, j_t)} \right\|_X^2 = \frac{\|X\|^2 b^2}{\|X_{\cdot r\cdot\cdot}\|^2} \sum_{k=1}^{n} \left\| M^{(p_t, r_t, i_t, j_t)} x_k \right\|^2
$$

$$
\leq \frac{\|X\|^2 b^2}{\|X_{\cdot r\cdot\cdot}\|^2} \sum_{k=1}^{n} \|X_{kr\cdot\cdot}\|^2 = \|X\|^2 b^2 \quad ,
$$

and we can choose $\mathbb{N} \ni m = \left\lceil \frac{b^2 \|X\|^2}{\epsilon^2} \right\rceil$ to get an $\epsilon$-cover of $\mathcal{F}$ via Eq. (56).

The theorem then follows from two particular bounds on $N(m, W) = \binom{m+2W-1}{2W-1}$. These are

$$
\binom{m+2W-1}{2d-1} \leq (2W)^m \quad ,
$$

which implies Eq. (54), and

$$
\binom{m+2W-1}{2W-1} \leq (1+m)^{2W-1} \quad ,
$$

which implies Eq. (55); see Lemma C.24 for details. $\qquad \square$

**Remark C.9.** In the proof, we bound

$$
\sum_{\beta=1}^{h} \sum_{\gamma=1}^{w} \left( x_{r\beta\gamma} \right)^2 \mathbb{1}_{\beta \equiv (1+i) \bmod s_h} \mathbb{1}_{\gamma \equiv (1+j) \bmod s_w} \leq \sum_{\beta=1}^{h} \sum_{\gamma=1}^{w} \left( x_{r\beta\gamma} \right)^2 \quad .
$$

Under additional assumptions on the data $X$ this result might be improved as, on average, one expects

$$
\sum_{\beta=1}^{h} \sum_{\gamma=1}^{w} \left( x_{r\beta\gamma} \right)^2 \mathbb{1}_{\beta \equiv (1+i) \bmod s_h} \mathbb{1}_{\gamma \equiv (1+j) \bmod s_w} \leq \frac{1}{s_h s_w} \sum_{\beta=1}^{h} \sum_{\gamma=1}^{w} \left( x_{r\beta\gamma} \right)^2 \quad ,
$$

which would reduce the $\frac{\|X\|^2 b^2}{\epsilon^2}$ terms in the bounds by the factor $1/(s_h s_w)$.

## C.3 Whole-network covering number bounds (general form)

In order to prove covering number bounds for residual networks, we utilize the following basic observation: a residual network is a composition of residual blocks and each residual block corresponds to addition of two (compositions of) functions on the same input (one of them is typically the identity function). Thus, if we know the covering numbers of compositions and additions, we can derive whole-network covering number bounds in an inductive way.

*Importantly, the derived covering bounds hold for a broad class of network architectures, including the special cases of non-residual and residual networks.*

### C.3.1 Covering number bounds for compositions and summations

Given normed spaces $(\mathcal{X}_i, \|\cdot\|_{\mathcal{X}_i})$ and function spaces $\mathcal{F}_i$ and $\mathcal{G}_i$ of functions $\mathcal{X}_i \to \mathcal{X}_{i+1}$, we present covering number bounds for the following derived function spaces:

$$\mathrm{Comp}(\mathcal{F}_1, \ldots, \mathcal{F}_L) = \{f_L \circ \cdots \circ f_1 \mid f_i \in \mathcal{F}_i\} \tag{57}$$

$$\mathrm{Sum}(\mathcal{F}_i, \mathcal{G}_i) = \{f_i + g_i \mid f_i \in \mathcal{F}_i, g_i \in \mathcal{G}_i\} \tag{58}$$

**Lemma C.10** (Compositions). *For $i \in \{1, 2, 3\}$, let $(\mathcal{X}_i, \|\cdot\|_{\mathcal{X}_i})$ be normed spaces and let $\mathcal{F}_i$ be classes of functions $\mathcal{X}_i \to \mathcal{X}_{i+1}$ with $\mathrm{Lip}(\mathcal{F}_i) < \infty$. Then, for any $\epsilon_1, \epsilon_2 > 0$ and any $X = (x_1, \ldots, x_n) \in \mathcal{X}_1^n$, the covering number of the class $\mathrm{Comp}(\mathcal{F}_1, \mathcal{F}_2)$ is bounded by*

$$\mathcal{N}\left(\mathrm{Comp}(\mathcal{F}_1, \mathcal{F}_2), \mathrm{Lip}(\mathcal{F}_2)\epsilon_1 + \epsilon_2, \|\cdot\|_X\right) \leq \mathcal{N}(\mathcal{F}_1, \epsilon_1, \|\cdot\|_X) \left( \sup_{f \in \mathcal{F}_1} \mathcal{N}\left(\mathcal{F}_2, \epsilon_2, \|\cdot\|_{f(X)}\right) \right) \tag{59}$$

*If $\mathcal{F}_2 = \{f_2\}$ is a singleton, then*

$$\mathcal{N}(\mathrm{Comp}(\mathcal{F}_1, \mathcal{F}_2), \mathrm{Lip}(\mathcal{F}_2)\epsilon_1, \|\cdot\|_X) \leq \mathcal{N}(\mathcal{F}_1, \epsilon_1, \|\cdot\|_X) \ . \tag{60}$$

**Remark C.11.** There is an analogous result which holds for external covering numbers, i.e.,

$$\mathcal{N}^{\mathrm{ext}}\left(\mathrm{Comp}(\mathcal{F}_1, \mathcal{F}_2), \mathrm{Lip}(\mathcal{F}_2)\epsilon_1 + \epsilon_2, \|\cdot\|_X\right)$$

$$\leq \mathcal{N}^{\mathrm{ext}}(\mathcal{F}_1, \epsilon_1, \|\cdot\|_X) \left( \sup_{f: \mathcal{X}_1 \to \mathcal{X}_2} \mathcal{N}^{\mathrm{ext}}\left(\mathcal{F}_2, \epsilon_2, \|\cdot\|_{f(X)}\right) \right) \ . \tag{61}$$

Notably, in this case, the supremum is taken over all $f \colon \mathcal{X}_1 \to \mathcal{X}_2$. However, this form is unusable for deriving the whole-network covering number bounds in Section C.4 as we want to handle the supremum via an assumption on the Lipschitz constant of the layer.

*Proof.* Fix $\epsilon_1, \epsilon_2 > 0$. Let $\mathcal{U}_{\mathcal{F}_1} \subset \mathcal{F}_1$ be a minimal $\epsilon_1$-cover of $(\mathcal{F}_1, \|\cdot\|_X)$, i.e., $\mathrm{card}(\mathcal{U}_{\mathcal{F}_1}) = \mathcal{N}(\mathcal{F}_1, \epsilon_1, \|\cdot\|_X)$. For any covering element $v \in \mathcal{U}_{\mathcal{F}_1}$, let $\mathcal{U}_{\mathcal{F}_2}(v) \subset \mathcal{F}_2$ be a minimal $\epsilon_2$-cover of $(\mathcal{F}_2, \|\cdot\|_{v(X)})$, i.e., $\mathrm{card}(\mathcal{U}_{\mathcal{F}_2}(v)) = \mathcal{N}(\mathcal{F}_2, \epsilon_2, \|\cdot\|_{v(X)})$ .

Denote $c_2 = \mathrm{Lip}(\mathcal{F}_2)$. We will show that

$$\mathcal{U}_{\mathrm{Comp}(\mathcal{F}_1, \mathcal{F}_2)} = \{w^v \circ v \mid v \in \mathcal{U}_{\mathcal{F}_1}, \ w^v \in \mathcal{U}_{\mathcal{F}_2}(v)\} \subset \mathrm{Comp}(\mathcal{F}_1, \mathcal{F}_2)$$

defines an $(\epsilon_1 c_2 + \epsilon_2)$-cover of $(\mathrm{Comp}(\mathcal{F}_1, \mathcal{F}_2), \|\cdot\|_X)$, i.e., for any $f_1 \in \mathcal{F}_1$ and any $f_2 \in \mathcal{F}_2$, there exist $v \in \mathcal{U}_{\mathcal{F}_1}$ and $w^v \in \mathcal{U}_{\mathcal{F}_2}(v)$ such that

$$\|f_2 \circ f_1 - w^v \circ v\|_X \leq c_2 \epsilon_1 + \epsilon_2 \ .$$

Indeed, since $\mathcal{U}_{\mathcal{F}_1}$ is an $\epsilon_1$-cover of $(\mathcal{F}_1, \|\cdot\|_X)$, we can choose $v \in \mathcal{U}_{\mathcal{F}_1}$ such that $\|f_1 - v\|_X \leq \epsilon_1$, and, since $\mathcal{U}_{\mathcal{F}_2}(v)$ is an $\epsilon_2$-cover of $(\mathcal{F}_2, \|\cdot\|_{v(X)})$, we can choose $w^v \in \mathcal{U}_{\mathcal{F}_2}(v)$ such that $\|f_2 - w^v\|_{v(X)} \leq \epsilon_2$. Thus,

$$\begin{aligned} \|f_2 \circ f_1 - w^v \circ v\|_X &= \|(f_2 \circ f_1 - f_2 \circ v) + (f_2 \circ v - w^v \circ v)\|_X \\ &\leq \|f_2 \circ f_1 - f_2 \circ v\|_X + \|f_2 \circ v - w^v \circ v\|_X \\ &\leq \mathrm{Lip}(f_2) \|f_1 - v\|_X + \|f_2 - w^v\|_{v(X)} \\ &\leq c_2 \epsilon_1 + \epsilon_2 \ , \end{aligned}$$

where the second inequality follows from Eq. (44). Therefore,

$$\mathcal{N}(\mathrm{Comp}(\mathcal{F}_1, \mathcal{F}_2), c_2\epsilon_1 + \epsilon_2, \|\cdot\|_X) \leq \mathrm{card}\left(\mathcal{U}_{\mathrm{Comp}(\mathcal{F}_1, \mathcal{F}_2)}\right)$$

$$= \mathrm{card}\left(\{w^v \circ v \mid v \in \mathcal{U}_{\mathcal{F}_1},\ w^v \in \mathcal{U}_{\mathcal{F}_2}(v)\}\right)$$

$$= \sum_{v \in \mathcal{U}_{\mathcal{F}_1}} \mathrm{card}\left(\mathcal{U}_{\mathcal{F}_2}(v)\right)$$

$$\leq \left(\sup_{v \in \mathcal{U}_{\mathcal{F}_1}} \mathrm{card}\left(\mathcal{U}_{\mathcal{F}_2}(v)\right)\right)\left(\sum_{v \in \mathcal{U}_{\mathcal{F}_1}} 1\right)$$

$$= \left(\sup_{v \in \mathcal{U}_{\mathcal{F}_1}} \mathrm{card}\left(\mathcal{U}_{\mathcal{F}_2}(v)\right)\right) \mathrm{card}(\mathcal{U}_{\mathcal{F}_1})$$

$$\overset{(\star)}{\leq} \left(\sup_{f \in \mathcal{F}_1} \mathrm{card}\left(\mathcal{U}_{\mathcal{F}_2}(f)\right)\right) \mathrm{card}\left(\mathcal{U}_{\mathcal{F}_1}\right)$$

$$\leq \left(\sup_{f \in \mathcal{F}_1} \mathcal{N}\left(\mathcal{F}_2, \epsilon_2, \|\cdot\|_{f(X)}\right)\right) \mathcal{N}\left(\mathcal{F}_1, \epsilon_1, \|\cdot\|_X\right) .$$

For $(\star)$, we used that $\mathcal{U}_{\mathcal{F}_1} \subset \mathcal{F}_1$ is an internal cover.

The special case of $\mathcal{F}_2 = \{f_2\}$ being a singleton is obvious, as we can choose $\mathcal{U}_{\mathcal{F}_2}(v) = \{f_2\}$ for every $v \in \mathcal{U}_{\mathcal{F}_1}$. Then, for every $f_2 \in \mathcal{F}_2$ and every $w^v \in \mathcal{U}_{\mathcal{F}_2}(v)$, it holds that $\mathrm{card}(\mathcal{U}_{\mathcal{F}_2}(v)) = 1$ and $\left\|f_2 - w_j^v\right\|_{v(X)} = 0$. $\qquad\square$

**Lemma C.12** (Summations). *Let $(\mathcal{X}, \|\cdot\|_{\mathcal{X}})$ and $(\mathcal{Y}, \|\cdot\|_{\mathcal{Y}})$ be normed spaces and let $\mathcal{F}, \mathcal{G}$ be classes of functions $\mathcal{X} \to \mathcal{Y}$. Then, for each $\epsilon_{\mathcal{F}}, \epsilon_{\mathcal{G}} > 0$ and each $X = (x_i, \ldots, x_n) \in \mathcal{X}^n$, the covering number of the class $\mathrm{Sum}(\mathcal{F}, \mathcal{G})$ is bounded by*

$$\mathcal{N}(\mathrm{Sum}(\mathcal{F}, \mathcal{G}), \epsilon_{\mathcal{F}} + \epsilon_{\mathcal{G}}, \|\cdot\|_X) \leq \mathcal{N}(\mathcal{F}, \epsilon_{\mathcal{F}}, \|\cdot\|_X)\mathcal{N}(\mathcal{G}, \epsilon_{\mathcal{G}}, \|\cdot\|_X) . \tag{62}$$

*If $\mathcal{G} = \{g\}$ is a singleton, then*

$$\mathcal{N}(\mathrm{Sum}(\mathcal{F}, \mathcal{G}), \epsilon_{\mathcal{F}}, \|\cdot\|_X) = \mathcal{N}(\mathcal{F}, \epsilon_{\mathcal{F}}, \|\cdot\|_X) . \tag{63}$$

*Proof.* Fix $\epsilon_{\mathcal{F}}, \epsilon_{\mathcal{G}} > 0$. Let $\mathcal{U}_{\mathcal{F}} \subset \mathcal{F}$ be a minimal $\epsilon_{\mathcal{F}}$-cover of $(\mathcal{F}, \|\cdot\|_X)$ and let $\mathcal{U}_{\mathcal{G}} \subset \mathcal{G}$ be a minimal $\epsilon_{\mathcal{G}}$-cover of $(\mathcal{G}, \|\cdot\|_X)$, i.e., $\mathcal{N}(\mathcal{F}, \epsilon_{\mathcal{F}}, \|\cdot\|_X) = \mathrm{card}(\mathcal{U}_{\mathcal{F}})$ and $\mathcal{N}(\mathcal{G}, \epsilon_{\mathcal{G}}, \|\cdot\|_X) = \mathrm{card}(\mathcal{U}_{\mathcal{G}})$.

We will show that

$$\mathcal{U}_{\mathrm{Sum}(\mathcal{F}, \mathcal{G})} = \{v + w \mid v \in \mathcal{U}_{\mathcal{F}},\ w \in U_{\mathcal{G}}\} \subset \mathrm{Sum}(\mathcal{F}, \mathcal{G})$$

defines an $(\epsilon_{\mathcal{F}} + \epsilon_{\mathcal{G}})$-cover of $(\mathrm{Sum}(\mathcal{F}, \mathcal{G}), \|\cdot\|_X)$, i.e., for every $f \in \mathcal{F}$ and every $g \in \mathcal{G}$, there exist $v \in \mathcal{U}_{\mathcal{F}}$ and $w \in \mathcal{U}_{\mathcal{G}}$ such that

$$\|(f + g) - (v + w)\|_X \leq \epsilon_{\mathcal{F}} + \epsilon_{\mathcal{G}} .$$

Indeed, since $\mathcal{U}_{\mathcal{F}}$ is an $\epsilon_{\mathcal{F}}$-cover of $(\mathcal{F}, \|\cdot\|_X)$, we can choose $v \in \mathcal{U}_{\mathcal{F}}$ such that $\|f - v\|_X \leq \epsilon_{\mathcal{F}}$ and since $\mathcal{U}_{\mathcal{G}}$ is an $\epsilon_{\mathcal{G}}$-cover of $(\mathcal{G}, \|\cdot\|_X)$, we can choose $w \in \mathcal{U}_{\mathcal{G}}$ such that $\|g - w\|_X \leq \epsilon_{\mathcal{G}}$. Then,

$$\|(f + g) - (v + w)\|_X = \|(f - v) + (g - w)\|_X$$

$$\leq \|f - v\|_X + \|g - w\|_X$$

$$\leq \epsilon_{\mathcal{F}} + \epsilon_{\mathcal{G}} .$$

Therefore, we have

$$\mathcal{N}(\mathrm{Sum}(\mathcal{F}, \mathcal{G}), \epsilon_{\mathcal{F}} + \epsilon_{\mathcal{G}}, \|\cdot\|_X) \leq \mathrm{card}\left(\mathcal{U}_{\mathrm{Sum}(\mathcal{F}, \mathcal{G})}\right)$$

$$= \mathrm{card}\left(\{v + w \mid v \in \mathcal{U}_{\mathcal{F}},\ w \in U_{\mathcal{G}}\}\right)$$

$$\leq \mathrm{card}(\mathcal{U}_{\mathcal{F}})\,\mathrm{card}(\mathcal{U}_{\mathcal{G}})$$

$$= \mathcal{N}(\mathcal{F}, \epsilon_{\mathcal{F}}, \|\cdot\|_X)\mathcal{N}(\mathcal{G}, \epsilon_{\mathcal{G}}, \|\cdot\|_X) .$$

The special case of $\mathcal{G} = \{g\}$ being a singleton is obvious, as we can choose $\mathcal{U}_{\mathcal{G}} = \{g\}$. Then $\mathcal{U}_{\mathrm{Sum}(\mathcal{F}, \mathcal{G})} = \{v + g \mid v \in \mathcal{U}_{\mathcal{F}}\}$ is a cover of $\mathrm{Sum}(\mathcal{F}, \mathcal{G})$ with cardinality $\mathrm{card}(\mathcal{U}_{\mathrm{Sum}(\mathcal{F}, \mathcal{G})}) = \mathrm{card}(\mathcal{U}_{\mathcal{F}})$ and radius $\epsilon_{\mathcal{F}}$. $\qquad\square$

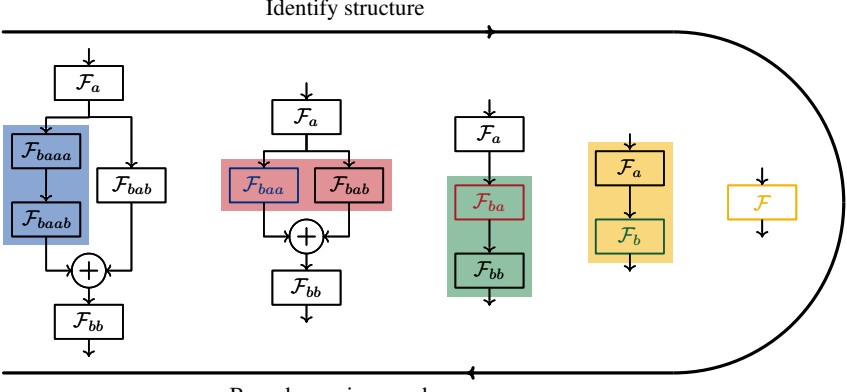

**Figure 11:** Schematic illustration of how to obtain whole-network covering number bounds by first identifying a way to write the network via summations and compositions, and then iteratively applying the respective inequalities. The function classes are systematically denoted by words with characters $a$ and $b$. Starting with $\mathcal{F}$ at the very right, we always add a character when replacing a function class by its building blocks.

Now that we know how to bound the covering numbers of compositions and summations, we can iteratively derive covering number bounds for all function classes obtained from these two operations.

### C.3.2 General strategy for bounding the covering numbers of complex classes

Let $\mathcal{F}$ be a function class whose covering number is *unknown* to us. If $\mathcal{F}$ can be built iteratively by compositions and summations of function classes with known covering number (bounds), then we can derive covering number bounds for $\mathcal{F}$ via the following strategy. In a first step, we identify the structure of $\mathcal{F}$, i.e., how it is built from compositions and summations. In a second step, starting with $\mathcal{F}$, we iteratively replace each function class by its simpler building blocks and the covering number of $\mathcal{F}$ by the respective bound.

To be more specific, we know by Lemma C.10 and Lemma C.12 that for $\mathcal{F} = \mathrm{Comp}(\mathcal{F}_a, \mathcal{F}_b)$, it holds that

$$\mathcal{N}(\mathcal{F}, \mathrm{Lip}(\mathcal{F}_b)\epsilon_{\mathcal{F}_a} + \epsilon_{\mathcal{F}_b}, \|\cdot\|_X) \leq \mathcal{N}(\mathcal{F}_a, \epsilon_{\mathcal{F}_a}, \|\cdot\|_X)\left(\sup_{f \in \mathcal{F}_a} \mathcal{N}(\mathcal{F}_b, \epsilon_{\mathcal{F}_b}, \|\cdot\|_{f(X)})\right) .$$

and for $\mathcal{F} = \mathrm{Sum}(\mathcal{F}_a, \mathcal{F}_b)$, it holds that

$$\mathcal{N}(\mathcal{F}, \epsilon_{\mathcal{F}_a} + \epsilon_{\mathcal{F}_b}, \|\cdot\|_X) \leq \mathcal{N}(\mathcal{F}_a, \epsilon_{\mathcal{F}_a}, \|\cdot\|_X)\mathcal{N}(\mathcal{F}_b, \epsilon_{\mathcal{F}_b}, \|\cdot\|_X) .$$

Now, if (for $x = a$ or $x = b$) some class $\mathcal{F}_x$ is of the form $\mathcal{F}_x = \mathrm{Comp}(\mathcal{F}_{xa}, \mathcal{F}_{xb})$ or $\mathcal{F}_x = \mathrm{Sum}(\mathcal{F}_{xa}, \mathcal{F}_{xb})$, we bound the right-hand side of the equations above by the same argument. We repeat this procedure until the right-hand side contains only terms of known covering number bounds. For an illustration of this stepwise process, see Fig. 11.

### C.3.3 Examples

**Example C.1** (Multi-composition). Let $(\mathcal{X}_1, \|\cdot\|_{\mathcal{X}_1}), \ldots, (\mathcal{X}_L, \|\cdot\|_{\mathcal{X}_L})$ be normed spaces. Let $\mathcal{F}_i$ be classes of functions $\mathcal{X}_i \to \mathcal{X}_{i+1}$ with bounded Lipschitz constants, i.e., $\mathrm{Lip}(\mathcal{F}_i) < \infty$. Denote $\overrightarrow{\mathcal{F}}_i = \mathrm{Comp}(\mathcal{F}_1, \ldots, \mathcal{F}_{i-1})$. Then, for any $\epsilon_i > 0$ and any finite $X = (x_1, \ldots, x_n) \in \mathcal{X}_1^n$, the covering number of the class $\mathcal{F} = \mathrm{Comp}(\mathcal{F}_1, \ldots, \mathcal{F}_L)$ is bounded by

$$\mathcal{N}\left(\mathcal{F}, \sum_{i=1}^{L}\left(\prod_{l=i+1}^{L}\mathrm{Lip}(\mathcal{F}_l)\right)\epsilon_i \mathbb{1}_{\mathrm{card}(\mathcal{F}_i)>1}, \|\cdot\|_X\right) \leq \prod_{i=1}^{L}\sup_{\psi_i \in \overrightarrow{\mathcal{F}}_i}\mathcal{N}\left(\mathcal{F}_i, \epsilon_i, \|\cdot\|_{\psi_i(X)}\right)^{\mathbb{1}_{\mathrm{card}(\mathcal{F}_i)>1}} .$$

$$(64)$$

*Proof.* We have

$$\mathcal{N}\left(\mathcal{F}, \sum_{i=1}^{L}\left(\prod_{l=i+1}^{L}\mathrm{Lip}(\mathcal{F}_l)\right)\epsilon_i\mathbb{1}_{\mathrm{card}(\mathcal{F}_i)>1}, \|\cdot\|_X\right)$$

$$= \mathcal{N}\Bigg(\mathrm{Comp}(\vec{}\mathcal{F}_L, \mathcal{F}_L),$$

$$\mathrm{Lip}(\mathcal{F}_L)\left(\sum_{i=1}^{L-1}\left(\prod_{l=i+1}^{L-1}\mathrm{Lip}(\mathcal{F}_l)\right)\epsilon_i\mathbb{1}_{\mathrm{card}(\mathcal{F}_i)>1}\right) + \epsilon_L\mathbb{1}_{\mathrm{card}(\mathcal{F}_L)>1}, \|\cdot\|_X\Bigg)$$

$$\leq \mathcal{N}\left(\vec{}\mathcal{F}_L, \sum_{i=1}^{L-1}\left(\prod_{l=i+1}^{L-1}\mathrm{Lip}(\mathcal{F}_l)\right)\epsilon_i\mathbb{1}_{\mathrm{card}(\mathcal{F}_i)>1}, \|\cdot\|_X\right)$$

$$\cdot\left(\sup_{\psi_L\in\vec{}\mathcal{F}_L}\mathcal{N}(\mathcal{F}_L, \epsilon_L, \|\cdot\|_{\psi_L(X)})\right)^{\mathbb{1}_{\mathrm{card}(\mathcal{F}_L)>1}}$$

$$\leq \cdots$$

$$\leq \prod_{i=1}^{L}\sup_{\psi_i\in\vec{}\mathcal{F}_i}\mathcal{N}\left(\mathcal{F}_i, \epsilon_i, \|\cdot\|_{\psi_i(X)}\right)^{\mathbb{1}_{\mathrm{card}(\mathcal{F}_i)>1}} \quad.$$

Here, we used Lemma C.10 with $\epsilon_{\vec{}\mathcal{F}_k} = \sum_{i=1}^{k-1}\left(\prod_{l=i+1}^{k-1}\mathrm{Lip}(\mathcal{F}_l)\right)\epsilon_i\mathbb{1}_{\mathrm{card}(\mathcal{F}_i)>1}$ $\qquad\square$

**Remark C.13.** We want to point out that Example C.1 implies the whole-network covering bound as in [3, Lemma A.7]. To see this, let $\mathcal{X}_i = \mathbb{R}^{d_i}$, $X_i = (x_{i_1}, \ldots, x_{i_n}) \in \mathbb{R}^{d_i\times n}$ and let $\sigma_i$ be fixed $\rho_i$-Lipschitz functions. Further, let $\mathcal{A}_i$ be sets of matrices $A \in \mathbb{R}^{d_{i+1}\times d_i}$. Then, the maps

$$\psi_i : (\mathcal{F}_i = \{\sigma_i\circ A|\ A\in\mathcal{A}_i\}, \|\cdot\|_{X_i}) \to (\mathbb{R}^{d_i\times n}, \|\cdot\|_{l_2})$$
$$\sigma_i\circ A \mapsto \sigma_i(AX_i)$$

define isometries, because

$$\begin{aligned}
\|\psi_i(\sigma_i\circ A)\|_{l_2}^2 &= \|\sigma_i(AX_i)\|_{l_2}^2 \\
&= \sum_{k=1}^{n}\|\sigma_i(Ax_{i_k})\|^2 \\
&= \sum_{k=1}^{n}\|(\sigma_i\circ A)(x_{i_k})\|^2 \\
&= \|\sigma_i\circ A\|_{X_i}^2 \quad.
\end{aligned} \tag{65}$$

Consequently,

$$\begin{aligned}
\mathcal{N}(\mathcal{F}_i, \rho_i\epsilon_i, \|\cdot\|_{X_i}) &= \mathcal{N}(\psi_i(\mathcal{F}_i), \rho_i\epsilon_i, \|\cdot\|_{l_2}) \\
&= \mathcal{N}(\{\sigma_i(AX_i)|\ A\in\mathcal{A}_i\}, \rho_i\epsilon_i, \|\cdot\|_{l_2}) \\
&\leq \mathcal{N}(\{AX_i|\ A\in\mathcal{A}_i\}, \epsilon_i, \|\cdot\|_{l_2}) \quad,
\end{aligned}$$

which are the factors on the right hand side of [3, Lemma A.7].

**Example C.2** (Addition block). Let $\mathcal{F}$ be the function class of addition blocks, i.e.,

$$\mathcal{F} = \mathrm{Sum}(\mathcal{G}, \mathcal{H}) \quad,$$

where $\mathcal{G} = \mathrm{Comp}(\mathcal{G}_1, \ldots, \mathcal{G}_{L_\mathcal{G}})$ and $\mathcal{H} = \mathrm{Comp}(\mathcal{H}_1, \ldots, \mathcal{H}_{L_\mathcal{H}})$. For brevity, we write

$$\begin{aligned}
\vec{}\mathcal{G}_i &= \mathrm{Comp}(\mathcal{G}_1, \ldots, \mathcal{G}_{i-1}) \quad, \\
\vec{}\mathcal{H}_i &= \mathrm{Comp}(\mathcal{H}_1, \ldots, \mathcal{H}_{i-1}) \quad.
\end{aligned}$$

The covering number of a block $\mathcal{F} = \mathrm{Sum}(\mathcal{G}, \mathcal{H})$ is bounded by

$$\mathcal{N}\left(\mathcal{F}, \sum_{i=1}^{L_\mathcal{G}} \left(\prod_{l=i+1}^{L_\mathcal{G}} \mathrm{Lip}(\mathcal{G}_l)\right) \epsilon_{\mathcal{G}_i} \mathbb{1}_{\mathrm{card}(\mathcal{G}_i)>1} + \right.$$

$$\left. \sum_{i=1}^{L_\mathcal{H}} \left(\prod_{l=i+1}^{L_\mathcal{H}} \mathrm{Lip}(\mathcal{H}_l)\right) \epsilon_{\mathcal{H}_i} \mathbb{1}_{\mathrm{card}(\mathcal{H}_i)>1}, \|\cdot\|_X \right) \leq \left(\prod_{i=1}^{L_\mathcal{G}} \sup_{\psi \in \vec{\mathcal{G}}_i} \mathcal{N}\left(\mathcal{G}_i, \epsilon_{\mathcal{G}_i}, \|\cdot\|_{\psi(X)}\right)\right)^{\mathbb{1}_{\mathrm{card}(\mathcal{G}_i)>1}}$$

$$\left(\prod_{i=1}^{L_\mathcal{H}} \sup_{\psi \in \vec{\mathcal{H}}_i} \mathcal{N}\left(\mathcal{H}_i, \epsilon_{\mathcal{H}_i}, \|\cdot\|_{\psi(X)}\right)\right)^{\mathbb{1}_{\mathrm{card}(\mathcal{H}_i)>1}}$$

$$(66)$$

*Proof.* From Lemma C.12, we know that

$$\mathcal{N}(\mathcal{F}, \epsilon_\mathcal{G} + \epsilon_\mathcal{H}, \|\cdot\|_X) \leq \mathcal{N}(\mathcal{G}, \epsilon_\mathcal{G}, \|\cdot\|_X) \mathcal{N}(\mathcal{H}, \epsilon_\mathcal{H}, \|\cdot\|_X)$$

holds for every $\epsilon_\mathcal{G} > 0$ and $\epsilon_\mathcal{H} > 0$. Choosing

$$\epsilon_\mathcal{G} = \sum_{i=1}^{L_\mathcal{G}} \left(\prod_{l=i+1}^{L_\mathcal{G}} \mathrm{Lip}(\mathcal{G}_l)\right) \epsilon_{\mathcal{G}_i} \mathbb{1}_{\mathrm{card}(\mathcal{G}_i)>1}$$

and

$$\epsilon_\mathcal{H} = \sum_{i=1}^{L_\mathcal{H}} \left(\prod_{l=i+1}^{L_\mathcal{H}} \mathrm{Lip}(\mathcal{H}_l)\right) \epsilon_{\mathcal{H}_i} \mathbb{1}_{\mathrm{card}(\mathcal{H}_i)>1} ,$$

and bounding each factor on the right-hand side via Example C.1 yields Eq. (66). □

**Example C.3** (Residual network). In the setting of Example C.1, let the function classes $\mathcal{F}_i$ be residual blocks

$$\mathcal{F}_i = \mathrm{Sum}(\mathcal{G}_i, \mathcal{H}_i) ,$$

where

$$\mathcal{G}_i = \mathrm{Comp}(\mathcal{G}_{i1}, \ldots, \mathcal{G}_{iL_{\mathcal{G}_i}}) \quad \text{and} \quad \mathcal{H}_i = \mathrm{Comp}(\mathcal{H}_{i1}, \ldots, \mathcal{H}_{iL_{\mathcal{H}_i}}) .$$

Assume, that $\mathrm{Lip}(\mathcal{G}_{ij}), \mathrm{Lip}(\mathcal{H}_{ij}) < \infty$ and that input data $X$ is given. For brevity, we write

$$\vec{\mathcal{G}}_{ij} = \mathrm{Comp}(\mathcal{G}_{i1}, \ldots, \mathcal{G}_{i,j-1})$$
$$\vec{\mathcal{H}}_{ij} = \mathrm{Comp}(\mathcal{H}_{i1}, \ldots, \mathcal{H}_{i,j-1})$$
$$\vec{\mathcal{F}}_i = \mathrm{Comp}(\mathcal{F}_1, \ldots, \mathcal{F}_{i-1}) .$$

The covering number of the residual network, $\mathcal{F} = \mathrm{Comp}(\mathcal{F}_1, \ldots, \mathcal{F}_L)$, is bounded by

$$\mathcal{N}(\mathcal{F}, \epsilon_\mathcal{F}, \|\cdot\|_X) \leq$$

$$\left(\prod_{i=1}^{L} \prod_{j=1}^{L_i} \sup_{\substack{\psi_{ij} \in \\ \mathrm{Comp}(\vec{\mathcal{F}}_i, \vec{\mathcal{G}}_{ij})}} \mathcal{N}\left(\mathcal{G}_{ij}, \epsilon_{\mathcal{G}_{ij}}, \|\cdot\|_{\psi_{ij}(X)}\right)^{\mathbb{1}_{\mathrm{card}(\mathcal{G}_{ij})>1}}\right)$$

$$(67)$$

$$\left(\prod_{i=1}^{L} \prod_{j=1}^{L_i} \sup_{\substack{\psi_{ij} \in \\ \mathrm{Comp}(\vec{\mathcal{F}}_i, \vec{\mathcal{H}}_{ij})}} \mathcal{N}\left(\mathcal{H}_{ij}, \epsilon_{\mathcal{H}_{ij}}, \|\cdot\|_{\psi_{ij}(X)}\right)^{\mathbb{1}_{\mathrm{card}(\mathcal{H}_{ij})>1}}\right),$$

where

$$\epsilon_\mathcal{F} = \sum_{i=1}^{L} \left(\prod_{l=i+1}^{L} \mathrm{Lip}(\mathcal{F}_i)\right) \epsilon_{\mathcal{F}_i}$$

with

$$\epsilon_{\mathcal{F}_i} = \sum_{j=1}^{L_{\mathcal{G}_i}} \left(\prod_{k=j+1}^{L_{\mathcal{G}_i}} \mathrm{Lip}(\mathcal{G}_{ik})\right) \epsilon_{\mathcal{G}_{ij}} \mathbb{1}_{\mathrm{card}(\mathcal{G}_{ij})>1} + \sum_{j=1}^{L_{\mathcal{H}_i}} \left(\prod_{k=j+1}^{L_{\mathcal{H}_i}} \mathrm{Lip}(\mathcal{H}_{ik})\right) \epsilon_{\mathcal{H}_{ij}} \mathbb{1}_{\mathrm{card}(\mathcal{H}_{ij})>1} .$$

*Proof.* Assuming $\operatorname{card}(\mathcal{F}_i) > 1$, we apply Example C.1 to $\mathcal{F} = \operatorname{Comp}(\mathcal{F}_1, \ldots, \mathcal{F}_L)$ to obtain

$$\mathcal{N}\left(\mathcal{F}, \sum_{i=1}^{L}\left(\prod_{l=i+1}^{L}\operatorname{Lip}(\mathcal{F}_i)\right)\epsilon_{\mathcal{F}_i}, \|\cdot\|_X\right) \leq \prod_{i=1}^{L}\sup_{\psi_i \in \overset{\rightarrow}{}\mathcal{F}_i}\mathcal{N}(\mathcal{F}_i, \epsilon_{\mathcal{F}_i}, \|\cdot\|_{\psi_i(X)}) .$$

Bounding the covering number of each block $\mathcal{F}_i$ via Example C.2 yields Eq. (67). $\qquad\square$

### C.3.4 Covering number bounds for concatenations

The general approach to bounding covering numbers of function classes, obtained from linking simple function classes via summations and compositions, can be easily extended. As an example, we can incorporate *concatenations*, as typically used in DenseNets [22], via the following lemma.

**Lemma C.14** (Concatenations). *Let $(\mathcal{X}, \|\cdot\|_{\mathcal{X}})$ be a normed space and let $(\mathcal{Y}, \|\cdot\|_{\mathcal{Y}}) = (\mathbb{R}^{d_{\mathcal{Y}}}, \|\cdot\|_{l_2})$ and $(\mathcal{Z}, \|\cdot\|_{\mathcal{Z}}) = (\mathbb{R}^{d_{\mathcal{Z}}}, \|\cdot\|_{l_2})$. Let $\mathcal{F}, \mathcal{G}$ be classes of functions $\mathcal{X} \to \mathcal{Y}$, resp. $\mathcal{X} \to \mathcal{Z}$. Define the function class $\operatorname{Cat}(\mathcal{F}, \mathcal{G})$ of concatenations $\mathcal{X} \to \mathcal{Y} \times \mathcal{Z}$ as*

$$\operatorname{Cat}(\mathcal{F}, \mathcal{G}) = \{(f, g) : x \mapsto (f(x), g(x)) \mid f \in \mathcal{F}, \ g \in \mathcal{G}\} . \tag{68}$$

*If we equip $\mathcal{Y} \times \mathcal{Z} = \mathbb{R}^{d_{\mathcal{Y}}d_{\mathcal{Z}}}$ with the $l_2$ norm, then*

$$\mathcal{N}(\operatorname{Cat}(\mathcal{F}, \mathcal{G}), \sqrt{\epsilon_{\mathcal{F}}^2 + \epsilon_{\mathcal{G}}^2}, \|\cdot\|_X) \leq \mathcal{N}(\mathcal{F}, \epsilon_{\mathcal{F}}, \|\cdot\|_X)\mathcal{N}(\mathcal{G}, \epsilon_{\mathcal{G}}, \|\cdot\|_X) . \tag{69}$$

*Proof.* Fix $\epsilon_{\mathcal{F}}, \epsilon_{\mathcal{G}} > 0$. Let $\mathcal{U}_{\mathcal{F}} \subset \mathcal{F}$ be a minimal $\epsilon_{\mathcal{F}}$-cover of $(\mathcal{F}, \|\cdot\|_X)$ and let $\mathcal{U}_{\mathcal{G}} \subset \mathcal{G}$ be a minimal $\epsilon_{\mathcal{G}}$-cover of $(\mathcal{G}, \|\cdot\|_X)$, i.e., $\mathcal{N}(\mathcal{F}, \epsilon_{\mathcal{F}}, \|\cdot\|_X) = \operatorname{card}(\mathcal{U}_{\mathcal{F}})$ and $\mathcal{N}(\mathcal{G}, \epsilon_{\mathcal{G}}, \|\cdot\|_X) = \operatorname{card}(\mathcal{U}_{\mathcal{G}})$.

We will show that

$$\mathcal{U}_{\operatorname{Cat}(\mathcal{F},\mathcal{G})} := \{(v, w) \mid v \in \mathcal{U}_{\mathcal{F}}, \ w \in U_{\mathcal{G}}\} \subset \operatorname{Cat}(\mathcal{F}_1, \mathcal{F}_2)$$

defines an $\sqrt{\epsilon_{\mathcal{F}}^2 + \epsilon_{\mathcal{G}}^2}$-cover of $(\operatorname{Cat}(\mathcal{F}, \mathcal{G}), \|\cdot\|_{l_2})$, i.e., for every $f \in \mathcal{F}$ and every $g \in \mathcal{G}$, there exist $v \in \mathcal{U}_{\mathcal{F}}$ and $w \in \mathcal{U}_{\mathcal{G}}$ such that

$$\|(f, g) - (v, w)\|_X \leq \sqrt{\epsilon_{\mathcal{F}}^2 + \epsilon_{\mathcal{G}}^2} .$$

Indeed, since $\mathcal{U}_{\mathcal{F}}$ is an $\epsilon_{\mathcal{F}}$-cover of $(\mathcal{F}, \|\cdot\|_X)$, we can choose $v$ such that $\|f - v\|_X \leq \epsilon_{\mathcal{F}}$ and since $\mathcal{U}_{\mathcal{G}}$ is an $\epsilon_{\mathcal{G}}$-cover of $(\mathcal{G}, \|\cdot\|_X)$, we can choose $w$ such that $\|g - w\|_X \leq \epsilon_{\mathcal{G}}$. Then

$$
\begin{aligned}
\|(f, g) - (v, w)\|_X^2 &= \sum_{i=1}^{n}\|(f, g)(x_i) - (v, w)(x_i)\|_{l_2}^2 \\
&= \sum_{i=1}^{n}\|(f - v, g - w)(x_i)\|_{l_2}^2 \\
&= \sum_{i=1}^{n}\left(\|(f - v)(x_i)\|_{l_2}^2 + \|(g - w)(x_i))\|_{l_2}^2\right) \qquad \text{(Pythagorean thm.)} \\
&= \|f - v\|_X^2 + \|g - w\|_X^2 \\
&\leq \epsilon_{\mathcal{F}}^2 + \epsilon_{\mathcal{G}}^2 ,
\end{aligned}
$$

Therefore,

$$
\begin{aligned}
\mathcal{N}(\operatorname{Cat}(\mathcal{F}, \mathcal{G}), \sqrt{\epsilon_{\mathcal{F}}^2 + \epsilon_{\mathcal{G}}^2}, \|\cdot\|_X) &\leq \operatorname{card}(\mathcal{U}_{\operatorname{Cat}(\mathcal{F},\mathcal{G})}) \\
&= \operatorname{card}\left(\{(v, w) \mid v \in \mathcal{U}_{\mathcal{F}}, \ w \in U_{\mathcal{G}}\}\right) \\
&= \operatorname{card}(\mathcal{U}_{\mathcal{F}})\operatorname{card}(\mathcal{U}_{\mathcal{G}}) \\
&= \mathcal{N}(\mathcal{F}, \epsilon_{\mathcal{F}}, \|\cdot\|_X)\mathcal{N}(\mathcal{G}, \epsilon_{\mathcal{G}}, \|\cdot\|_X) .
\end{aligned}
$$

$\square$

## C.4 Whole-network covering number bounds (convolutional & fully-connected)

In order to compute covering number bounds for specific residual network architectures, we need to specify the function classes $\mathcal{G}_i$ and $\mathcal{H}_i$. We will present exemplary proofs for a simple residual network with fixed shortcuts (Theorem C.15, which corresponds to Theorem 3.4 in the main text) and the ResNet18 architecture [20] without batch normalization, see Example C.4.

### C.4.1 Bounds for residual networks

**Theorem C.15** (Covering numbers for residual networks). *For $i = 1, \ldots, L$ let $j = 1, \ldots, L_i$, $s_{ij} > 0$ and $b_{ij} > 0$. Further, let $\mathcal{F}$ be the class of residual networks of the form*

$$f = \sigma_L \circ f_L \circ \cdots \circ \sigma_1 \circ f_1 \ , \tag{70}$$

*with $\sigma_i$ fixed $\rho_i$-Lipschitz functions satisfying $\sigma_i(0) = 0$, and $f_i$ residual blocks with fixed shortcuts $g_i$, i.e.,*

$$f_i : g_i + (\sigma_{iL_i} \circ f_{iL_i} \circ \cdots \circ \sigma_{i1} \circ f_{i1}) \ , \tag{71}$$

*where $\sigma_{ij}$ are fixed $\rho_{ij}$-Lipschitz functions with $\sigma_{ij}(0) = 0$ and $g_i$ is Lipschitz with $g_i(0) = 0$.*

*The fully-connected or convolutional layers $f_{ij} \in \mathrm{layer}_{ij}$ are parametrized by matrices $A_{ij}$ or weight tensors $K_{ij}$, respectively. They satisfy Lipschitz constant constraints $s_{ij}$ and $(2,1)$ group norm distance constraints $b_{ij}$ w.r.t. reference weights $M_{ij}$. That is, for convolutions*

$$\mathrm{layer}_{ij} = \left\{ \phi_{K_{ij}} \mid \mathrm{Lip}(\phi_{K_{ij}}) \leq s_{ij}, \ \|K_{ij} - M_{ij}\|_{2,1} \leq b_{ij} \right\}$$

*and for fully-connected layers*

$$\mathrm{layer}_{ij} = \left\{ \phi : x \mapsto A_{ij}x \mid \mathrm{Lip}(\phi) \leq s_{ij}, \ \left\|A_{ij}^{\top} - M_{ij}^{\top}\right\|_{2,1} \leq b_{ij} \right\} \ .$$

*Upon letting $W_{ij}$ denote the number of parameters of each layer and defining*

$$C_{ij} = C_{ij}(X) = 2 \frac{\|X\|}{\sqrt{n}} \left(\prod_{l=1}^{L} s_l \rho_l\right) \frac{\prod_{k=1}^{L_i} \rho_{ik} s_{ik}}{s_i} \frac{b_{ij}}{s_{ij}} \ ,$$

$$\bar{L} = \sum_{i=1}^{L} L_i \ , \quad W = \max_{ij} W_{ij} \ , \quad s_i = \mathrm{Lip}(g_i) + \prod_{j=1}^{L_i} \rho_{ij} s_{ij} \ ,$$

*it holds that*

$$\log \mathcal{N}(\mathcal{F}, \epsilon, \|\cdot\|_X) \leq \log(2W) \left(\sum_{i=1}^{L} \sum_{j=1}^{L_i} \left\lceil C_{ij}^{2/3} \right\rceil\right)^3 \left\lceil \frac{n}{\epsilon^2} \right\rceil \ , \tag{72}$$

*and*

$$\log \mathcal{N}(\mathcal{F}, \epsilon, \|\cdot\|_X) \leq \sum_{i=1}^{L} \sum_{j=1}^{L_i} 2W_{ij} \log \left(1 + \left\lceil \bar{L}^2 C_{ij}^2 \right\rceil \left\lceil \frac{n}{\epsilon^2} \right\rceil\right) \ . \tag{73}$$

*Proof.* As we consider residual networks with fixed shortcuts, the covering number bound from Example C.3 simplifies to

$$\mathcal{N}\left(\mathcal{F}, \sum_{i=1}^{L} \left(\prod_{l=i+1}^{L} \mathrm{Lip}(\mathcal{F}_l)\right) \sum_{j=1}^{L_{\mathcal{H}_i}} \left(\prod_{k=j+1}^{L_{\mathcal{H}_i}} \mathrm{Lip}(\mathcal{H}_{ik})\right) \epsilon_{\mathcal{H}_{ij}}, \|\cdot\|_X\right)$$

$$\leq \prod_{i=1}^{L} \prod_{j=1}^{L_i} \sup_{\substack{\psi_{ij} \in \\ \mathrm{Comp}(\to \mathcal{F}_i, \to \mathcal{H}_{ij})}} \mathcal{N}\left(\mathcal{H}_{ij}, \epsilon_{\mathcal{H}_{ij}}, \|\cdot\|_{\psi_{ij}(X)}\right) \ .$$

In our setting, $\mathcal{F}_i = \{\sigma_i \circ f_i\}$ with $\mathrm{Lip}(\mathcal{F}_i) \leq \rho_i s_i$, $\mathcal{H}_{ij} = \mathrm{Comp}(\mathrm{layer}_{ij}, \{\sigma_{ij}\})$ with $\mathrm{Lip}(\mathcal{H}_{ij}) \leq \rho_{ij} s_{ij}$. Further, $\epsilon_{\mathcal{H}_{ij}} = \rho_{ij} \epsilon_{ij}$ and $L_{\mathcal{H}_{ij}} = L_{ij}$. As covering numbers decrease with the radius, it follows that

$$\mathcal{N}\left(\mathcal{F}, \sum_{i=1}^{L}\left(\prod_{l=i+1}^{L} s_l \rho_l\right)\sum_{j=1}^{L_i}\left(\prod_{k=j+1}^{L_i} s_{ik}\rho_{ik}\right)\rho_{ij}\epsilon_{ij}, \|\cdot\|_X\right)$$

$$\leq \prod_{i=1}^{L}\prod_{j=1}^{L_i} \sup_{\substack{\psi_{ij} \in \\ \mathrm{Comp}(^\to\mathcal{F}_i, ^\to \mathrm{layer}_{ij})}} \mathcal{N}\left(\mathrm{layer}_{ij}, \epsilon_{ij}, \|\cdot\|_{\psi_{ij}(X)}\right) \ .$$

Now, for each $ij$ referring to convolutional layers, we have

$$\mathcal{N}\left(\mathrm{layer}_{ij}, \epsilon_{ij}, \|\cdot\|_{\psi_{ij}(X)}\right)$$

$$= \mathcal{N}\left(\left\{\phi_{K_{ij}} \mid \mathrm{Lip}(\phi_{K_{ij}}) \leq s_{ij}, \ \|K_{ij} - M_{ij}\|_{2,1} \leq b_{ij}\right\}, \epsilon_{ij}, \|\cdot\|_{\psi_{ij}(X)}\right)$$

$$= \mathcal{N}\left(\left\{\phi_{K_{ij}} - \phi_{M_{ij}} \mid \mathrm{Lip}(\phi_{K_{ij}}) \leq s_{ij}, \ \|K_{ij} - M_{ij}\|_{2,1} \leq b_{ij}\right\}, \epsilon_{ij}, \|\cdot\|_{\psi_{ij}(X)}\right)$$

$$= \mathcal{N}\left(\left\{\phi_{K_{ij} - M_{ij}} \mid \mathrm{Lip}(\phi_{K_{ij}}) \leq s_{ij}, \ \|K_{ij} - M_{ij}\|_{2,1} \leq b_{ij}\right\}, \epsilon_{ij}, \|\cdot\|_{\psi_{ij}(X)}\right) \ .$$

In this chain of equalities, we used the translation invariance of covering numbers, i.e., Lemma C.12 with one summand being the singleton $\{-\phi_{M_{ij}}\}$, and the linearity of $\phi$ in the weights to accommodate the distance to initialization. An analogous inequality holds for fully-connected layers.

Theorem C.7 provides bounds for the covering number of the superset

$$\left\{\phi_{K_{ij} - M_{ij}} \mid \|K_{ij} - M_{ij}\|_{2,1} \leq b_{ij}\right\} \ .$$

Hence, to proceed, we need to transition to external covering numbers, which requires halving the radius $\epsilon$. This yields

$$\mathcal{N}\left(\mathrm{layer}_{ij}, \epsilon_{ij}, \|\cdot\|_{\psi_{ij}(X)}\right)$$

$$\leq \ \mathcal{N}\left(\left\{\phi_{K_{ij} - M_{ij}} \mid \mathrm{Lip}(\phi_{K_{ij}}) \leq s_{ij}, \ \|K_{ij} - M_{ij}\|_{2,1} \leq b_{ij}\right\}, \epsilon_{ij}, \|\cdot\|_{\psi_{ij}(X)}\right)$$

$$\overset{\text{Eq. (46)}}{\leq} \mathcal{N}^{\mathrm{ext}}\left(\left\{\phi_{K_{ij} - M_{ij}} \mid \mathrm{Lip}(\phi_{K_{ij}}) \leq s_{ij}, \ \|K_{ij} - M_{ij}\|_{2,1} \leq b_{ij}\right\}, \frac{\epsilon_{ij}}{2}, \|\cdot\|_{\psi_{ij}(X)}\right)$$

$$\overset{\text{Eq. (47)}}{\leq} \mathcal{N}^{\mathrm{ext}}\left(\left\{\phi_{K_{ij} - M_{ij}} \mid \|K_{ij} - M_{ij}\|_{2,1} \leq b_{ij}\right\}, \frac{\epsilon_{ij}}{2}, \|\cdot\|_{\psi_{ij}(X)}\right)$$

$$\overset{\text{Eq. (46)}}{\leq} \mathcal{N}\left(\left\{\phi_{K_{ij} - M_{ij}} \mid \|K_{ij} - M_{ij}\|_{2,1} \leq b_{ij}\right\}, \frac{\epsilon_{ij}}{2}, \|\cdot\|_{\psi_{ij}(X)}\right) \ . \tag{74}$$

Thus, by Eq. (54), it holds that

$$\log \mathcal{N}\left(\mathcal{F}, \sum_{i=1}^{L}\left(\prod_{l=i+1}^{L} s_l \rho_l\right)\sum_{j=1}^{L_i}\left(\prod_{k=j+1}^{L_i} s_{ik}\rho_{ik}\right)\rho_{ij}\epsilon_{ij}, \|\cdot\|_X\right)$$

$$\leq \sum_{i=1}^{L}\sum_{j=1}^{L_i} \sup_{\substack{\psi_{ij} \in \\ \mathrm{Comp}(^\to\mathcal{F}_i, ^\to \mathrm{layer}_{ij})}} \log(2W_{ij})\left\lceil \frac{4\|\psi_{ij}(X)\|^2 b_{ij}^2}{\epsilon_{ij}^2}\right\rceil$$

$$\leq \log(2W)\sum_{i=1}^{L}\sum_{j=1}^{L_i}\left\lceil 4\|X\|^2 \left(\prod_{l=1}^{i-1} s_l \rho_l\right)^2\left(\prod_{k=1}^{j-1} s_{lk}\rho_{lk}\right)^2 \frac{b_{ij}^2}{\epsilon_{ij}^2}\right\rceil \ .$$

Notably, the second inequality requires the assumption that all $g_i$, $f_{ij}$, $\sigma_i$ and $\sigma_{ij}$ map zero to zero. The next step is to choose radii $\epsilon_{ij}$ so that the right-hand side becomes small under the condition that

$$\sum_{i=1}^{L}\left(\prod_{l=i+1}^{L} s_l \rho_l\right)\sum_{j=1}^{L_i}\left(\prod_{k=j+1}^{L_i} s_{ik}\rho_{ik}\right)\rho_{ij}\epsilon_{ij} = \epsilon \ . \tag{75}$$

We choose

$$\epsilon_{ij} = \frac{\epsilon}{\left(\prod_{l=i+1}^{L} s_l \rho_l\right)\left(\prod_{k=j+1}^{L_i} s_{ik}\rho_{ik}\right)\rho_{ij}} \frac{\alpha_{ij}}{\sum_{lk}\alpha_{lk}} \quad, \qquad \alpha_{ij} = \frac{b_{ij}^{2/3}}{s_{ij}^{2/3}} \quad, \tag{76}$$

which would be optimal for the analogous optimization problem without ceiling functions. Then,

$$\log \mathcal{N}\left(\mathcal{F}, \epsilon, \|\cdot\|_X\right)$$

$$\leq \log(2W) \sum_{i=1}^{L}\sum_{j=1}^{L_i}\left\lceil 4\left(\prod_{l=1}^{L} s_l\rho_l\right)^2 \frac{\left(\prod_{k=1}^{L_i} s_{ik}\rho_{ik}\right)^2}{s_i^2}\frac{b_{ij}^2}{s_{ij}^2\alpha_{ij}^2}\frac{\|X\|^2}{\epsilon^2}\left(\sum_{l=1}^{L}\sum_{k=1}^{L_l}\alpha_{lk}\right)^2\right\rceil$$

$$\leq \log(2W) \sum_{i=1}^{L}\sum_{j=1}^{L_i}\left\lceil 4\left(\prod_{l=1}^{L} s_l\rho_l\right)^2 \frac{\left(\prod_{k=1}^{L_i} s_{ik}\rho_{ik}\right)^2}{s_i^2}\frac{b_{ij}^{2/3}}{s_{ij}^{2/3}}\frac{\|X\|^2}{n}\frac{n}{\epsilon^2}\left(\sum_{l=1}^{L}\sum_{k=1}^{L_l}\frac{b_{lk}^{2/3}}{s_{lk}^{2/3}}\right)^2\right\rceil$$

$$\leq \log(2W)\left(\sum_{i=1}^{L}\sum_{j=1}^{L_i}\left\lceil\left(2\frac{\|X\|}{\sqrt{n}}\left(\prod_{l=1}^{L} s_l\rho_l\right)\frac{\left(\prod_{k=1}^{L_i} s_{ik}\rho_{ik}\right)}{s_i}\frac{b_{ij}}{s_{ij}}\right)^{2/3}\right\rceil\right)^3\left\lceil\frac{n}{\epsilon^2}\right\rceil$$

$$= \log(2W)\left(\sum_{i=1}^{L}\sum_{j=1}^{L_i}\left\lceil C_{ij}^{2/3}\right\rceil\right)^3\left\lceil\frac{n}{\epsilon^2}\right\rceil \quad,$$

which establishes the *first* covering number bound, i.e., Eq. (72), from Theorem C.15.

Similarly, Eq. (55) implies

$$\log\mathcal{N}\left(\mathcal{F}, \sum_{i=1}^{L}\left(\prod_{l=i+1}^{L} s_l\rho_l\right)\sum_{j=1}^{L_i}\left(\prod_{k=j+1}^{L_i} s_{ik}\rho_{ik}\right)\rho_{ij}\epsilon_{ij}, \|\cdot\|_X\right)$$

$$\leq \sum_{i=1}^{L}\sum_{j=1}^{L_i} 2W_{ij}\log\left(1+\left\lceil 4\|X\|^2\left(\prod_{l=1}^{i-1} s_l\rho_l\right)^2\left(\prod_{k=1}^{j-1} s_{ik}\rho_{ik}\right)^2\frac{b_{ij}^2}{\epsilon_{ij}^2}\right\rceil\right) \quad.$$

Again, we need to choose the $\epsilon_{ij}$ such that Eq. (75) holds. We choose

$$\epsilon_{ij} = \frac{\epsilon}{\left(\prod_{l=i+1}^{L} s_l\rho_l\right)\left(\prod_{k=j+1}^{L_i} s_{ik}\rho_{ik}\right)\rho_{ij}}\frac{\alpha_{ij}}{\sum_{lk}\alpha_{lk}} \quad \text{with} \quad \alpha_{ij} = 1 \quad. \tag{77}$$

This simple choice yields the optimal solution for the problem of minimizing

$$\sum_{ij}\log\left(\frac{4\|X\|^2\left(\prod_{l=1}^{i-1} s_l\rho_l\right)^2\left(\prod_{k=1}^{j-1}\rho_{ik}\right)^2 b_{ij}^2}{\epsilon_{ij}^2}\right) \quad.$$

Hence, we expect it to be a good choice if the $W_{ij}$ are roughly equal and $\epsilon$ is small. Overall, we get

$$\log\mathcal{N}\left(\mathcal{F}, \epsilon, \|\cdot\|_X\right) \leq \sum_{i=1}^{L}\sum_{j=1}^{L_i} 2W_{ij}\log\left(1+\left\lceil 4\bar{L}^2\left(\prod_{l=1}^{L} s_l\rho_l\right)^2\frac{\left(\prod_{k=1}^{L_i} s_{ik}\rho_{ik}\right)^2}{s_i^2}\frac{b_{ij}^2}{s_{ij}^2}\frac{\|X\|^2}{\epsilon^2}\right\rceil\right)$$

$$\leq \sum_{i=1}^{L}\sum_{j=1}^{L_i} 2W_{ij}\log\left(1+\left\lceil\bar{L}^2 C_{ij}^2\right\rceil\left\lceil\frac{n}{\epsilon^2}\right\rceil\right) \quad,$$

which establishes the *second* covering number bound, i.e., Eq. (73), from Theorem C.15. $\qquad\square$

**Corollary C.16** (Covering numbers for non-residual networks). *For $i \in \{1, \ldots, L\}$, let $\mathrm{layer}_i$ be a function class with Lipschitz constraint $s_i$ and $(2,1)$ group norm distance constraint $b_i$ with respect to a reference weight $M_i$. In particular, if $\mathrm{layer}_i$ is convolutional, then*

$$\mathrm{layer}_i = \left\{ \phi_K \mid \mathrm{Lip}(\phi_K) \leq s_i, \ \|K_i - M_i\|_{2,1} \leq b_i \right\}$$

*and if $\mathrm{layer}_i$ is fully-connected, then*

$$\mathrm{layer}_i = \left\{ \phi : x \mapsto A_i x \mid \mathrm{Lip}(\phi) \leq s_i, \ \|A_i - M_i\|_{2,1} \leq b_i \right\} \ .$$

*We write $W_i$ for the number of parameters of each layer, i.e., the number of elements of each $K_i$, resp. $A_i$. Further, let $\mathcal{F} = \{\sigma_L \circ f_L \circ \cdots \circ \sigma_1 \circ f_1 \mid f_i \in \mathrm{layer}_i\}$, where the maps $\sigma_i$ are $\rho_i$-Lipschitz with $\sigma_i(0) = 0$, and define*

$$C_i = C_i(X) = 2 \frac{\|X\|}{\sqrt{n}} \left( \prod_{l=1}^{L} \rho_l s_l \right) \frac{b_i}{s_i} \ , \qquad W = \max_i W_i \ . \tag{78}$$

*Then, for every input data $X = (x_1, \ldots, x_n)$ and every $\epsilon > 0$, it holds that*

$$\log \mathcal{N}\left(\mathcal{F}, \epsilon, \|\cdot\|_X\right) \leq \log(2W) \left( \sum_{i=1}^{L} \left\lceil C_i^{2/3} \right\rceil \right)^3 \left\lceil \frac{n}{\epsilon^2} \right\rceil \tag{79}$$

*and*

$$\log \mathcal{N}(\mathcal{F}, \epsilon, \|\cdot\|_X) \leq \sum_{i=1}^{L} 2W_i \log \left( 1 + \left\lceil C_i^2 \right\rceil \left\lceil \frac{n}{\epsilon^2} \right\rceil \right) \ . \tag{80}$$

*Proof.* Follows directly from Theorem C.15, as the network can be considered as a single (long) residual block, whose shortcut $g : x \mapsto 0$ is the zero map. □

**Remark C.17.** Similarly, we can derive covering number bounds for networks, where each block $\mathcal{F}_i$ is a sum of $w_i$ parametrized maps, i.e., $\mathcal{F}_i = \mathrm{Sum}(\mathcal{G}_{i1}, \ldots, \mathcal{G}_{iw_i})$, with $\mathcal{G}_{ij} = \mathrm{Comp}(\mathrm{layer}_{ij1}, \sigma_{ij1}, \ldots, \mathrm{layer}_{ijL_{ij}}, \sigma_{ijL_{ij}})$. In this setting, the whole-network covering number is bounded by

$$\log \mathcal{N}(\mathcal{F}, \epsilon, \|\cdot\|_X) \leq \log(2W) \left( \sum_{i=1}^{L} \sum_{j=1}^{w_i} \sum_{k=1}^{L_{ij}} \left\lceil C_{ijk}^{2/3} \right\rceil \right)^3 \left\lceil \frac{n}{\epsilon^2} \right\rceil \tag{81}$$

and

$$\log \mathcal{N}(\mathcal{F}, \epsilon, \|\cdot\|_X) \leq \sum_{i=1}^{L} \sum_{j=1}^{w_i} \sum_{k=1}^{L_{ij}} 2W_{ijk} \log \left( 1 + \left\lceil \bar{L}^2 C_{ijk}^2 \right\rceil \left\lceil \frac{n}{\epsilon^2} \right\rceil \right) \tag{82}$$

for

$$C_{ijk} = 4 \frac{\|X\|}{\sqrt{n}} \left( \prod_{l=1}^{L} s_l \rho_l \right) \frac{\prod_{m=1}^{L_{ij}} \rho_{ijm} s_{ijm}}{s_i} \frac{b_{ijk}}{s_{ijk}}, \qquad \bar{L} = \sum_{i=1}^{L} \sum_{j=1}^{w_i} L_{ij} \ , \tag{83}$$

where $s_i = \sum_{j=1}^{w_i} \left( \prod_{k=1}^{L_{ij}} s_{ijk} \rho_{ijk} \right)$, $\rho_{ijk} = \mathrm{Lip}(\sigma_{ijk})$ and $s_{ijk}, b_{ijk}$ are constraints on the layers $\mathrm{layer}_{ijk}$.

### C.4.2  Application to specific architectures

**Example C.4** (ResNet18). We derive covering number bounds for the ResNet18 architecture [20] *without* batch normalization, illustrated in Fig. 12. We can think of the ResNet18 as a composition of 10 residual blocks, the first and last one having the zero map as shortcut and five blocks having identity shortcuts. The remaining 3 blocks have downsampling shortcuts of the form $\sigma \circ \psi$, where $\psi$ is a 1x1 convolution and $\rho$ is the ReLU activation function. These blocks are handled by Remark C.17. Furthermore, all nonlinearities are 1-Lipschitz and map zero to zero.

For any data $X = (x_1, \ldots x_n)$ and any $\epsilon > 0$, the covering number of the function class $\mathcal{F}$ corresponding to the ResNet18 architecture without batch normalization, with no bias parameters and

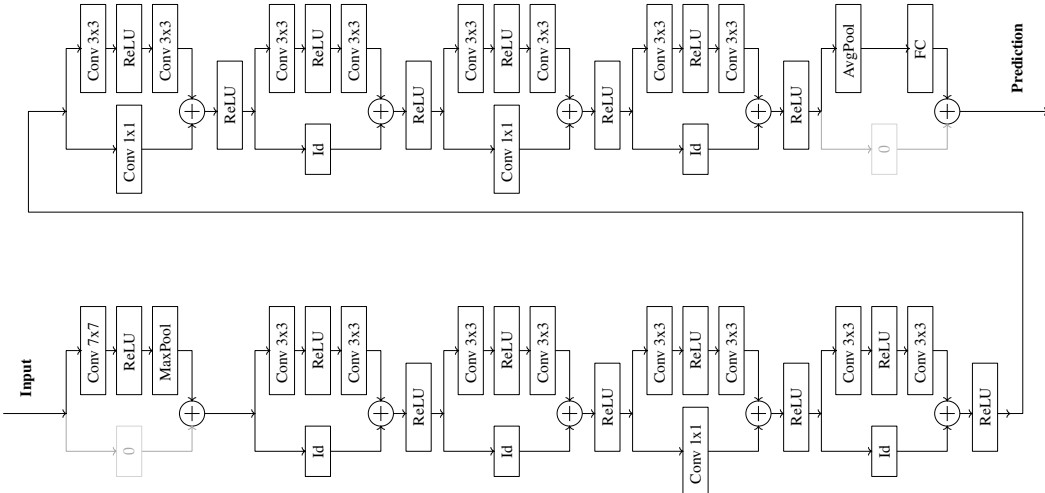

**Figure 12:** ResNet18 [20] architecture *without* batch normalization.

with distance and Lipschitz constrained layers, is approximately (ignoring ceiling functions) bounded by

$$\log \mathcal{N}(\mathcal{F}, \epsilon, \|\cdot\|_X) \lesssim 4 \frac{\|X\|^2}{\epsilon} \log(2W)$$

$$s_1^2 \left( \prod_{i \in \{2,3,5,7,9\}} (1 + s_{i1}s_{i2}) \right)^2 \left( \prod_{i \in \{4,6,8\}} (s_{i,\text{down}} + s_{i1}s_{i2}) \right)^2 s_{10}^2$$

$$\left[ \frac{b_1^{2/3}}{s_1^{2/3}} + \frac{b_{10}^{2/3}}{s_{10}^{2/3}} + \sum_{i \in \{2,3,5,7,9\}} \frac{1}{(1 + s_{i1}s_{i2})^{2/3}} \left( \frac{b_{i1}^{2/3}}{s_{i1}^{2/3}} + \frac{b_{i2}^{2/3}}{s_{i2}^{2/3}} \right) \right.$$

$$\left. + \sum_{i \in \{4,6,8\}} \frac{1}{(s_{i,\text{down}} + s_{i1}s_{i2})^{2/3}} \left( \frac{b_{i,\text{down}}^{2/3}}{s_{i,\text{down}}^{2/3}} + \frac{b_{i1}^{2/3}}{s_{i1}^{2/3}} + \frac{b_{i2}^{2/3}}{s_{i2}^{2/3}} \right) \right]^3 .$$

Here $s_{i1}$, resp $s_{i2}$, denotes the Lipschitz constraint on the first, resp. second, layer in the $i$-th residual block and $s_{i,\text{down}}$ the constraint on the downsampling layer (1x1 convolution). The (2,1)-distance constraints are denoted by $b_{i1}, b_{i2}$ and $b_{i,\text{down}}$.

## C.5 Rademacher complexity & Generalization bounds

The empirical Rademacher complexity can be upper bounded via Dudley's entropy integral. In the following, we restate a variant of this standard result as it appears in Bartlett et al. [3].

**Theorem C.18** (Dudley entropy integral, cf. [3, Lemma A.5]). *Let $\mathcal{F}$ be a class of functions mapping to $[0,1]$ containing the zero function. Then*

$$\hat{\mathfrak{R}}_X(\mathcal{F}) \leq \inf_{0 < t \leq \sqrt{n}} \left( \frac{4t}{\sqrt{n}} + \frac{12}{n} \int_t^{\sqrt{n}} \sqrt{\log \mathcal{N}(\mathcal{F}, \epsilon, \|\cdot\|_X)} \, d\epsilon \right) .$$

We will compute Dudley's entropy integral for the covering number bounds from Section C.4.

**Theorem C.19** (Empirical Rademacher complexity for residual networks). *For $i = 1, \ldots, L$ let $j = 1, \ldots, L_i$, $s_{ij} > 0$ and $b_{ij} > 0$. Further, let $\mathcal{F}$ be the class of residual networks of the form*

$$f = \sigma_L \circ f_L \circ \cdots \circ \sigma_1 \circ f_1 , \tag{84}$$

*with $\sigma_i$ fixed $\rho_i$-Lipschitz functions satisfying $\sigma_i(0) = 0$, and with $f_i$ residual blocks with identity shortcuts, i.e.,*

$$f_i : \text{Id} + (\sigma_{iL_i} \circ f_{iL_i} \circ \cdots \circ \sigma_{i1} \circ f_{i1}) , \tag{85}$$

where $\sigma_{ij}$ are fixed $\rho_{ij}$-Lipschitz functions with $\sigma_{ij}(0) = 0$ and $f_{ij}$ are convolutional or fully-connected layers. The layers $f_{ij}$ satisfy Lipschitz constraints $\mathrm{Lip}(f_{ij}) \leq s_{ij}$ and the corresponding weight tensors $K_{ij}$, respectively weight matrices $A_{ij}$, satisfy distance constraints

$$\left\| K_{ij} - K_{ij}^{(0)} \right\|_{2,1} \leq b_{ij} \ , \quad \text{respectively} \quad \left\| A_{ij} - A_{ij}^{(0)} \right\|_{2,1} \leq b_{ij} \ ,$$

with respect to reference weights $K_{ij}^{(0)}$, respectively $A_{ij}^{(0)}$.

Upon letting $W_{ij}$ denote the number of parameters of each layer and defining

$$\tilde{C}_{ij}(X) = \tilde{C}_{ij} = \frac{4}{\gamma} \frac{\|X\|}{\sqrt{n}} \left( \prod_{l=1}^{L} \mathrm{Lip}(\mathcal{F}_l)\rho_l \right) \frac{\prod_{k=1}^{L_i} \rho_{ik} s_{ik}}{\mathrm{Lip}(\mathcal{F}_i)} \frac{b_{ij}}{s_{ij}} \ ,$$

$$\bar{L} = \sum_{i=1}^{L} L_i \ , \qquad W = \max_{ij} W_{ij} \ ,$$

the empirical Rademacher complexity of the function class $\mathcal{F}_\gamma$, with margin parameter $\gamma > 0$, satisfies

$$\hat{\mathfrak{R}}_X(\mathcal{F}_\gamma) \leq \frac{4}{n} + \frac{12 H_{n-1}}{\sqrt{n}} \sqrt{\log(2W)} \left( \sum_{i=1}^{L} \sum_{i=j}^{L_i} \left\lceil \tilde{C}_{ij}^{2/3} \right\rceil \right)^{3/2} \tag{86}$$

and

$$\hat{\mathfrak{R}}_X(\mathcal{F}) \leq \frac{12}{\sqrt{n}} \sqrt{\sum_{i=1}^{L} \sum_{j=1}^{L_i} 2 W_{ij} \left( \log \left( 1 + \left\lceil \bar{L}^2 \tilde{C}_{ij}^2 \right\rceil \right) + \zeta \left( \frac{3}{2}, 1 \right)^{1/3} \zeta \left( \frac{3}{2}, 1 + 1 / \left\lceil \bar{L}^2 \tilde{C}_{ij}^2 \right\rceil \right)^{2/3} \right)}$$
$$\tag{87}$$

Here, $H_{n-1} = \sum_{m=1}^{n-1} \frac{1}{m}$ denotes the $(n-1)$-th harmonic number and $\zeta(s,q) = \sum_{n=0}^{\infty} \frac{1}{(q+n)^s}$ the Hurwitz zeta function.

**Remark C.20.** The harmonic number satisfies $H_{n-1} < \log(n) + \gamma \approx \log(n) + 0.58$. The function $\psi : x \mapsto \zeta \left( \frac{3}{2}, 1 \right)^{1/3} \zeta \left( \frac{3}{2}, 1 + 1/x \right)^{2/3}$ is monotonically increasing with $\psi(0) = 0$ and upper bounded by $\zeta(\frac{3}{2}) \approx 2.62$. So, for large $\tilde{C} = \max_{ij} \tilde{C}_{ij}$, the second summand is negligible and Eq. (87) scales as $\sqrt{\bar{W} \log(\bar{L}^2 \tilde{C}^2)}$. Here, $\bar{W} = \sum_{ij} W_{ij}$ denotes the number of network parameters.

*Proof.* Both inequalities follow from a combination of Dudley's entropy integral with a covering number bound from Theorem C.15.

Since $\ell_\gamma(-\mathcal{M}(\cdot, \cdot))$ is a fixed $2/\gamma$-Lipschitz function, the covering number of $\mathcal{F}_\gamma$ can be bounded as in Corollary C.16 with

$$C_{ij}(X) = \tilde{C}_{ij}(X) = \frac{4}{\gamma} \frac{\|X\|}{\sqrt{n}} \left( \prod_{l=1}^{L} \rho_l s_l \right) \frac{b_i}{s_i} \ . \tag{88}$$

To prove Eq. (86), we insert Eq. (72) into Dudley's entropy integral, which yields

$$\hat{\mathfrak{R}}_X(\mathcal{F}) \leq \inf_{0 \leq t \leq \sqrt{n}} \left( \frac{4t}{\sqrt{n}} + \frac{12}{n} \sqrt{\log(2W)} \left( \sum_{i=1}^{L} \sum_{j=1}^{L_i} \left\lceil \tilde{C}_{ij}^{2/3} \right\rceil \right)^{3/2} \int_{t}^{\sqrt{n}} \left\lceil \frac{\sqrt{n}}{\epsilon} \right\rceil \, d\epsilon \right)$$

$$= \inf_{0 \leq t \leq \sqrt{n}} \left( \frac{4t}{\sqrt{n}} + \frac{12}{\sqrt{n}} \sqrt{\log(2W)} \left( \sum_{i=1}^{L} \sum_{j=1}^{L_i} \left\lceil \tilde{C}_{ij}^{2/3} \right\rceil \right)^{3/2} \int_{t/\sqrt{n}}^{1} \left\lceil \frac{1}{s} \right\rceil \, ds \right)$$

$$= \inf_{0 \leq t \leq 1} \left( 4t + \frac{12}{\sqrt{n}} \sqrt{\log(2W)} \left( \sum_{i=1}^{L} \sum_{j=1}^{L_i} \left\lceil \tilde{C}_{ij}^{2/3} \right\rceil \right)^{3/2} \int_{t}^{1} \left\lceil \frac{1}{s} \right\rceil \, ds \right) \ .$$

The value of the integral is a harmonic number if $1/t \in \mathbb{N}$, as then

$$\int_t^1 \left\lceil \frac{1}{s} \right\rceil \mathrm{d}s = \sum_{m=1}^{1/t-1} \int_{1/(m+1)}^{1/m} \left\lceil \frac{1}{s} \right\rceil \mathrm{d}s = \sum_{m=1}^{1/t-1} \left( \frac{1}{m} - \frac{1}{m+1} \right)(m+1) = \sum_{m=1}^{1/t-1} \frac{1}{m} = H_{1/t-1} \ .$$

Choosing $t = 1/n$, establishes the inequality in Eq. (86).

To prove Eq. (87), we first observe that, by Jensen's inequality, it holds that

$$\frac{1}{\sqrt{n}} \int_0^{\sqrt{n}} \sqrt{\log \mathcal{N}(\mathcal{F}, \epsilon, \|\cdot\|_X)} \, \mathrm{d}\epsilon \le \sqrt{\frac{1}{\sqrt{n}} \int_0^{\sqrt{n}} \log \mathcal{N}(\mathcal{F}, \epsilon, \|\cdot\|_X) \, \mathrm{d}\epsilon}$$

$$= \frac{1}{\sqrt[4]{n}} \sqrt{\int_0^{\sqrt{n}} \log \mathcal{N}(\mathcal{F}, \epsilon, \|\cdot\|_X) \, \mathrm{d}\epsilon}$$

and thus

$$\hat{\mathfrak{R}}_X(\mathcal{F}) \le \frac{12}{n} n^{1/4} \sqrt{\int_0^{\sqrt{n}} \log \mathcal{N}(\mathcal{F}, \epsilon, \|\cdot\|_X) \, \mathrm{d}\epsilon} \ .$$

Then, recalling Eq. (73), i.e.,

$$\log \mathcal{N}(\mathcal{F}_\gamma, \epsilon, \|\cdot\|_X) \le \sum_{ij} 2W_{ij} \log \left( 1 + \left\lceil \bar{L}^2 \tilde{C}_{ij}^2 \right\rceil \left\lceil \frac{n}{\epsilon^2} \right\rceil \right)$$

yields

$$\int_0^{\sqrt{n}} \log \mathcal{N}(\mathcal{F}, \epsilon, \|\cdot\|_X) \, \mathrm{d}\epsilon$$

$$\le \sum_{ij} 2W_{ij} \int_0^{\sqrt{n}} \log \left( 1 + \left\lceil \bar{L}^2 \tilde{C}_{ij}^2 \right\rceil \left\lceil \frac{n}{\epsilon^2} \right\rceil \right) \, \mathrm{d}\epsilon$$

$$= \sqrt{n} \sum_{ij} 2W_{ij} \int_0^1 \log \left( 1 + \left\lceil \bar{L}^2 \tilde{C}_{ij}^2 \right\rceil \left\lceil \frac{1}{s^2} \right\rceil \right) \, \mathrm{d}s$$

$$\overset{\text{Lemma C.25}}{\le} \sqrt{n} \sum_{ij} 2W_{ij} \left( \log \left( 1 + \left\lceil \bar{L}^2 \tilde{C}_{ij}^2 \right\rceil \right) + \zeta \left( \frac{3}{2}, 1 \right)^{1/3} \zeta \left( \frac{3}{2}, 1 + 1 / \left\lceil \bar{L}^2 \tilde{C}_{ij}^2 \right\rceil \right)^{2/3} \right) \ .$$

The last inequality follows from Lemma C.25 (proof deferred to Section C.6). Overall, this implies

$$\hat{\mathfrak{R}}_X(\mathcal{F}) \le \frac{12}{\sqrt{n}} \sqrt{2 \sum_{i=1}^L \sum_{j=1}^{L_i} W_{ij} \left( \log \left( 1 + \left\lceil \bar{L}^2 \tilde{C}_{ij}^2 \right\rceil \right) + \zeta \left( \frac{3}{2}, 1 \right)^{1/3} \zeta \left( \frac{3}{2}, 1 + 1 / \left\lceil \bar{L}^2 \tilde{C}_{ij}^2 \right\rceil \right)^{2/3} \right)}$$

which establishes the inequality in Eq. (87). $\qquad\qquad\qquad\qquad\qquad\qquad\qquad\qquad \square$

**Corollary C.21.** *Let $\gamma > 0$ and let $\tilde{C}_i = 2C_i/\gamma$. For non-residual networks as specified in Corollary C.16, the empirical Rademacher complexity of $\mathcal{F}_\gamma$ satisfies*

$$\hat{\mathfrak{R}}_X(\mathcal{F}_\gamma) \le \frac{4}{n} + \frac{12 H_{n-1}}{\sqrt{n}} \sqrt{\log(2W)} \left( \sum_{i=1}^L \left\lceil \tilde{C}_i^{2/3} \right\rceil \right)^{3/2} \tag{89}$$

*and*

$$\hat{\mathfrak{R}}_X(\mathcal{F}) \le \frac{12}{\sqrt{n}} \sqrt{\sum_{i=1}^L 2W_i \left( \log \left( 1 + \left\lceil L^2 \tilde{C}_i^2 \right\rceil \right) + \zeta \left( \frac{3}{2}, 1 \right)^{1/3} \zeta \left( \frac{3}{2}, 1 + 1 / \left\lceil L^2 \tilde{C}_i^2 \right\rceil \right)^{2/3} \right)} \ .$$

$$\tag{90}$$

For the sake of completeness, we state the generalization bounds that result from the Rademacher complexity bounds for networks with a priori constrained weights.

**Theorem C.22.** *For $i = 1, \ldots, L$ let $j = 1, \ldots, L_i$, $s_{ij} > 0$ and $b_{ij} > 0$. Let $\mathcal{F}$ be the class of residual networks of the form*

$$f = \sigma_L \circ f_L \circ \cdots \circ \sigma_1 \circ f_1 \ , \tag{91}$$

*with $\sigma_i$ fixed $\rho_i$-Lipschitz functions satisfying $\sigma_i(0) = 0$, and with $f_i$ residual blocks with identity shortcuts, i.e.,*

$$f_i : \mathrm{Id} + (\sigma_{iL_i} \circ f_{iL_i} \circ \cdots \circ \sigma_{i1} \circ f_{i1}) \ , \tag{92}$$

*where $\sigma_{ij}$ are fixed $\rho_{ij}$-Lipschitz functions with $\sigma_{ij}(0) = 0$ and $f_{ij}$ are convolutional or fully-connected layers whose weight tensors $K_{ij}$, resp. weight matrices $A_{ij}$, satisfy the distance constraints*

$$\left\| K_{ij} - K_{ij}^{(0)} \right\|_{2,1} \leq b_{ij} \quad and \quad \left\| A_{ij} - A_{ij}^{(0)} \right\|_{2,1} \leq b_{ij} \ ,$$

*with respect to reference weights $K_{ij}^{(0)}$, resp. $A_{ij}^{(0)}$, and the Lipschitz constraints $\mathrm{Lip}(f_{ij}) \leq s_{ij}$.*

*Let $W_{ij}$ denote the number of parameters of each layer and define*

$$\tilde{C}_{ij}(X) = \tilde{C}_{ij} = \frac{4}{\gamma} \frac{\|X\|}{\sqrt{n}} \left( \prod_{l=1}^{L} \mathrm{Lip}(\mathcal{F}_l)\rho_l \right) \frac{\prod_{k=1}^{L_i} \rho_{ik} s_{ik}}{\mathrm{Lip}(\mathcal{F}_i)} \frac{b_{ij}}{s_{ij}} \ ,$$

$$\bar{L} = \sum_{i=1}^{L} L_i \ , \qquad W = \max_{ij} W_{ij} \ .$$

*Then, for fixed margin parameter $\gamma > 0$, every network $f \in \mathcal{F}$ satisfies*

$$\mathbb{P}[\arg\max_{i=1,\ldots,k} f(x)_i \neq y] \leq \hat{\mathcal{R}}_\gamma(f) + \frac{8}{n} + 24\sqrt{\log(2W)} \left( \sum_{i=1}^{L} \sum_{i=j}^{L_i} \left\lceil \tilde{C}_{ij}^{2/3} \right\rceil \right)^{3/2} \frac{H_{n-1}}{\sqrt{n}} + 3\sqrt{\frac{\log(\frac{2}{\delta})}{2n}} \tag{93}$$

*and*

$$\mathbb{P}[\arg\max_{i=1,\ldots,k} f(x)_i \neq y]$$

$$\leq \hat{\mathcal{R}}_\gamma(f)$$

$$+ \frac{24}{\sqrt{n}} \sqrt{2 \sum_{i=1}^{L} \sum_{i=j}^{L_i} W_{ij} \left( \log\left(1 + \left\lceil \bar{L}^2 \tilde{C}_{ij}^2 \right\rceil\right) + \zeta\left(\frac{3}{2}, 1\right)^{1/3} \zeta\left(\frac{3}{2}, 1 + 1/\left\lceil \bar{L}^2 \tilde{C}_{ij}^2 \right\rceil\right)^{2/3} \right)}$$

$$+ 3\sqrt{\frac{\log\left(\frac{2}{\delta}\right)}{2n}} \tag{94}$$

*with probability of at least $1 - \delta$ over an i.i.d. draw $((x_1, y_1), \ldots, (x_n, y_n))$.*

*Proof.* Recall Lemma 3.1, i.e.,

$$\mathbb{P}[\arg\max_{i=1,\ldots,k} f(x)_i \neq y] \leq \hat{\mathcal{R}}_\gamma(f) + 2\hat{\mathfrak{R}}_S(\mathcal{F}_\gamma) + 3\sqrt{\frac{\log(\frac{2}{\delta})}{2n}} \ , \tag{95}$$

where

$$\mathcal{F}_\gamma = \{(x, y) \mapsto \ell_\gamma(-\mathcal{M}(f(x), y)) : f \in \mathcal{F}\} \ . \tag{96}$$

Bounding the empirical Rademacher complexity $\hat{\mathfrak{R}}_S(\mathcal{F})$ via Theorem C.19 proves the theorem. □

**Remark C.23.** By a union bound argument over the constraint sets and the margin parameter, the generalization bound above can be transformed to a generalization bound which depends directly on the norms of the network weights and the Lipschitz constants instead of a priori constraints, see for example [3, Lemma A.9]. Furthermore, Lipschitz augmentation [43] allows to replace the product of Lipschitz constants by empirical equivalents, i.e., norms of activations and norms of Jacobians.

## C.6 Calculations

This section contains postponed calculations.

**Lemma C.24** (used in Theorem C.5 and Theorem C.7). *For any $n \in \mathbb{N}$, it holds that*

$$\binom{n+k}{k} \leq \min\left((k+1)^n, (n+1)^k\right) \tag{97}$$

*Proof.* To prove the first inequality, note that

$$\binom{n+k}{k} = \frac{(n+k)!}{k!\,n!} = \frac{(k+1)\cdots(n+k)}{1\cdots n} = \prod_{j=1}^{n} \frac{k+j}{j} \leq \prod_{j=n}^{k}(k+1) = (k+1)^n \quad.$$

Similarly,

$$\binom{n+k}{k} = \frac{(n+k)!}{k!\,n!} = \frac{(n+1)\cdots(n+k)}{1\cdots k} = \prod_{i=1}^{k} \frac{n+i}{i} \leq \prod_{i=1}^{k}(n+1) = (n+1)^k \quad.$$

$\square$

**Lemma C.25** (used in Theorem C.19). *For any $\alpha > 0$, it holds that*

$$\int_0^1 \log\left(1 + \alpha\left\lceil \frac{1}{s^2} \right\rceil\right) \, ds \leq \log(1 + \alpha) + \zeta\left(\frac{3}{2}, 1\right)^{1/3} \zeta\left(\frac{3}{2}, \frac{1+\alpha}{\alpha}\right)^{2/3} \tag{98}$$

*Proof.* The function

$$\mathbb{1}_{[0,1]}(s) \log\left(1 + \alpha\left\lceil \frac{1}{s^2} \right\rceil\right) = \sum_{m=1}^{\infty} \mathbb{1}_{\left[\frac{1}{\sqrt{m+1}}, \frac{1}{\sqrt{m}}\right]}(s) \log\left(1 + \alpha(m+1)\right)$$

is piecewise constant and so its integral is defined as

$$\int_0^1 \log\left(1 + \alpha\left\lceil \frac{1}{s^2} \right\rceil\right) \, ds = \lim_{M \to \infty} \sum_{m=1}^{M} \log(1 + \alpha(m+1))\left(\frac{1}{\sqrt{m}} - \frac{1}{\sqrt{m+1}}\right) \quad.$$

For any $M \in \mathbb{N}$, the partial sums are

$$\sum_{m=1}^{M}\left(\frac{1}{\sqrt{m}} - \frac{1}{\sqrt{m+1}}\right) \log\left(1 + \alpha(m+1)\right)$$

$$= \sum_{m=1}^{M} \frac{1}{\sqrt{m}} \log\left(1 + \alpha(m+1)\right) - \sum_{m=1}^{M} \frac{1}{\sqrt{m+1}} \log\left(1 + \alpha(m+1)\right)$$

$$= \sum_{m=1}^{M} \frac{1}{\sqrt{m}} \log\left(1 + \alpha(m+1)\right) - \sum_{m=2}^{M+1} \frac{1}{\sqrt{m}} \log\left(1 + \alpha m\right)$$

$$= \sum_{m=1}^{M} \frac{1}{\sqrt{m}} \log\left(1 + \alpha(m+1)\right) - \sum_{m=1}^{M} \frac{1}{\sqrt{m}} \log\left(1 + \alpha m\right) + \log\left(1 + \alpha\right) - \frac{\log\left(1 + \alpha(M+1)\right)}{\sqrt{M+1}}$$

$$= \sum_{m=1}^{M} \frac{1}{\sqrt{m}} \log\left(\frac{1 + \alpha(m+1)}{1 + \alpha m}\right) + \log\left(1 + \alpha\right) - \frac{\log\left(1 + \alpha(M+1)\right)}{\sqrt{M+1}}$$

$$= \sum_{m=1}^{M} \frac{1}{\sqrt{m}} \log\left(1 + \frac{1}{1/\alpha + m}\right) + \log\left(1 + \alpha\right) - \frac{\log\left(1 + \alpha(M+1)\right)}{\sqrt{M+1}} \quad.$$

Since $\lim_{M \to \infty} \frac{\log(1+\alpha M)}{\sqrt{M}} = 0$ for every $\alpha > 0$, we conclude

$$\int_0^1 \log\left(1 + \alpha\left\lceil \frac{1}{s^2} \right\rceil\right) \, ds = \sum_{m=1}^{\infty} \frac{1}{\sqrt{m}} \log\left(1 + \frac{1}{1/\alpha + m}\right) + \log\left(1 + \alpha\right) \quad. \tag{99}$$

Since $\frac{1}{1/\alpha+m} \in (0,1)$ for any $\alpha > 0$ and $m \in \mathbb{N}$, the logarithm is given by the Mercator series

$$\log\left(1 + \frac{1}{1/\alpha+m}\right) = \sum_{k=1}^{\infty} \frac{(-1)^{k+1}}{k}\left(\frac{1}{1/\alpha+m}\right)^k . \tag{100}$$

Inserting this into the series from above and exchanging the order of summation, we get

$$\sum_{m=1}^{\infty} \frac{1}{\sqrt{m}} \log\left(1 + \frac{1}{1/\alpha+m}\right) = \sum_{k=1}^{\infty} \frac{(-1)^{k+1}}{k} \sum_{m=1}^{\infty} \frac{1}{\sqrt{m}} \frac{1}{(1/\alpha+m)^k} .$$

This is an alternating convergent series in $k$, so its first summand

$$\sum_{m=1}^{\infty} \frac{1}{\sqrt{m}} \frac{1}{1/\alpha+m} \leq \left(\sum_{m=1}^{\infty} \frac{1}{m^{3/2}}\right)^{1/3} \left(\sum_{m=1}^{\infty} \frac{1}{(1/\alpha+m)^{3/2}}\right)^{2/3} \tag{101}$$
$$= \zeta(3/2,1)^{1/3} \zeta(3/2,1+1/\alpha)^{2/3}$$

(using Hölder inequality) already provides an upper bound, i.e.,

$$\int_0^1 \log\left(1 + \alpha\left\lceil\frac{1}{s^2}\right\rceil\right) \, ds \leq \log(1+\alpha) + \zeta(3/2,1)^{1/3}\zeta(3/2,1+1/\alpha)^{2/3} .$$

$\square$