# OpenReview forum: "On Measuring Excess Capacity in Neural Networks"
_NeurIPS.cc/2022/Conference — NeurIPS 2022 Accept_

### Official Review · Reviewer_BA91 · 2022-07-09

**Rating:** 6
**Confidence:** 1
**Soundness:** 3 good
**Presentation:** 3 good
**Contribution:** 3 good

**Summary:**

The authors propose an extension to prior Rademacher complexity bounds to account for more realistic neural network architectures used in practice, specifically convolution and skip connections, as found in Residual Networks, a modified form of which are also used in the empirical results. The extension relies on two capacity terms: the Lipschitz constants of each of the layers, and the (2, 1) group norm distance to the initialization of the conv filter weights. The authors show experiments based on training a modified ResNet-18 model without batch normalization across the CIFAR-10, CIFAR-100 and Tiny ImageNet classification datasets, and show that constrained models (i.e. models that have lower group norm and Lipschitz capacity terms) can achieve similar performance to models with excess capacity (reaching 100% training accuracy), while having similar capacity on what are presumed to be increasingly difficult tasks. The authors propose this indicates that, like known with parameter counts, neural networks are compressible w.r.t. weight norms.

**Questions:**

- The authors point to the fact that compressibility w.r.t. parameter counts is well known, but claim that their experimental results show a novel result — that neural networks are compressible w.r.t. weight norms. This presumes that weight norm and parameter counts are independent (or “orthogonal”). However, the authors use weight decay in their experiments, i.e. an L2 weight norm regularization term, and the variance of the common random initialization methods for NN layers (i.e. Xavier/He) explicitly depends on the parameter count of a layer. Can we really assume that weight norms and parameter counts are independent in the context of real-world neural network training?
- While the authors acknowledge that a significant problem in the field is that complexity bounds are often vacuous, it’s not obvious to me why complexity measures and related capacity bounds such as those explored in this paper would be a good method of analyzing excess capacity?
- I don’t see any details on how the Lipschitz constant for all layers in the results section (i.e. Table 1) is calculated empirically, making it difficult to repeat these results.
- Training ResNets without batch norm often is problematic (increasing with depth), without using methods such as those outlined in the Fixup paper (Zhang et al., 2019). The authors don’t mention any such workarounds, and I’d just like to ask if the authors could comment on this?
- It’s not clear to me why layer-wise Lipschitz constant is important intuitively, as we do not perform layer-wise optimization? Isn’t the Lipschitz constant over all parameters (i.e. dimensions in the loss landscape) more relevant?
- Isn’t 1x1 convolution on a 1x1 spatial feature map a linear combination or dot product over the channels rather than matrix multiplication?

**Limitations:**

I saw no discussion of societal impact of this work. I think one potential question is if only looking at generalization, as done in this work, is an appropriate measure of compressibility. For example how different is the function learned in the constrained solution? Achieving similar accuracy over the entire test set doesn’t mean functions learned are similar or predictions on the same samples is similar (e.g. as analyzed in the Deep Ensembles work for different solutions). In particular, are the constrained models more/less biased in predictions to particular classes in a classification task?

**Strengths And Weaknesses:**

Strengths:
* The paper is well written overall, with the motivation for the proposed capacity terms made clear, and appropriate background references
* The authors extend a complexity measure to more realistic settings — convolution all layers and skip connections are used in most contemporary neural network architectures for learning from images.
* The authors present a derivation of their proposed complexity measures

Weaknesses
* The intuition behind the capacity terms proposed was not obvious to me
* There are some reproducibility issues in the experimental results presented, notably it’s not obvious to me how the Lipschitz constant is calculated for different layers (only the classifier is explicitly addressed by construction from what I can see).
* The claim of independence of weight norm and parameter count doesn’t hold up in the real-world experimental setting. In particular, in the stated example of neural network pruning, it is often done in a manner explicitly attempting to preserve the weight norm (i.e. removing smallest magnitude weights).

---

> ### Author Response · Authors · 2022-08-02
> **Response to Reviewer BA91 (3/3)**
>
> #### **(7) Considering convolution as matrix multiplication**
> > Isn’t 1x1 convolution on a 1x1 spatial feature map a linear combination or dot product over the channels rather than matrix multiplication?
>
> This is correct if we consider the convolution operation of a single filter. Stacking the outputs of all filters thus corresponds to matrix multiplication (each row of the matrix contains one filter). We stuck with the matrix multiplication terminology to highlight that our result subsumes the result of [Bartlett et al., NeurIPS '17] as a special case (see the discussion after Theorem 3.2).
>
> #### **(8) Societal impact**
> > I saw no discussion of societal impact of this work. I think one potential question is if only looking at generalization, as done in this work, is an appropriate measure of compressibility. For example how different is the function learned in the constrained solution? Achieving similar accuracy over the entire test set doesn’t mean functions learned are similar or predictions on the same samples is similar (e.g. as analyzed in the Deep Ensembles work for different solutions). In particular, are the constrained models more/less biased in predictions to particular classes in a classification task?
>
> Thanks for pointing this out! This should have been included and will be in a final version. Despite the more theoretical nature of our work, it is worth pointing out that any potential bias-amplifying effects due to capacity reduction should be investigated.
>
> As a starting point, we evaluated the number of predictions per class on the testing datasets for constrained and unconstrained models trained on CIFAR-100.
> When considering models in an isolated manner, we observed that predictions were more uniformly distributed over the classes for the unconstrained models as indicated by a smaller standard deviation
> of the number of predictions per class. The average of these standard deviations over 10 trained models is $11.2$ in the unconstrained setting and $32.5$ in the constrained setting.
> However, when comparing class predictions across models, we did not observe any inherently favored classes in either setting. That is, for any fixed class, the mean of the number of predictions across models approaches 100.
>
> Further, from preliminary experiments motivated by the reviewer's comment, we find that constrained models tend to be less susceptible to certain adversarial attacks. More precisely, out of the studied adversarial attacks (FGSM, FGM, L2PGD, LinfPGD, L2Deepfool, additive noise, contrast reduction, Gaussian blur), constrained models are less susceptible to the gradient-based attacks FGSM, FGM, L2PGD, and LinfPGD. For L2Deepfool, contrast reduction and Gaussian blur, constrained and unconstrained models are equally affected. To our surprise, the unconstrained models are less susceptible to additive Gaussian and uniform noise.
> However, these experiments should be taken with caution, since our collection included only a small subset of possible adversarial attacks.
> We look forward to continuing this specific research branch in the future.

---

> > ### Comment · Reviewer_BA91 · 2022-08-09
> > **Rebuttal Response**
> >
> > (7) Yes, this is correct, for the set of filters it is a matrix multiplication still.
> >
> > (8) Thanks for being open to considering the ethical dimensions of these questions, despite this being a very theoretical paper! Indeed I think we as a research community (including myself) often focus on generalization alone and neglect to compare things in function space. Your initial experiments sound very interesting, and sound like they could potentially lead to more theoretical & practical insights. I look forward to seeing your final paper, and your future work on this.

---

> ### Author Response · Authors · 2022-08-02
> **Response to Reviewer BA91 (2/3)**
>
> #### **(4) Relevance of complexity bounds**
>
> > While the authors acknowledge that a significant problem in the field is that complexity bounds are often vacuous, it’s not obvious to me why complexity measures and related capacity bounds such as those explored in this paper would be a good method of analyzing excess capacity?
>
> We argue that Rademacher complexity bounds are particularly informative concerning which quantities play a significant role. Exerting control over these quantities shows the expected effects, as evidenced by our results in Table 1 and Fig. 2. Yet, from a sample complexity point of view, we agree that the bounds are not yet sufficient.
>
> #### **(5) Training without batchnorm**
> > Training ResNets without batch norm often is problematic (increasing with depth), without using methods such as those outlined in the Fixup paper (Zhang et al., 2019). The authors don’t mention any such workarounds, and I’d just like to ask if the authors could comment on this?
>
> In our experiments, we did not observe any issues while training the ResNet18 models without batch normalization. Yet, admittedly, a ResNet18 model is not very deep, and problems will most likely arise if one would, e.g., seek to train a ResNet50 (or larger) model (as remarked by the reviewer).
>
> **We think our work is a critical step towards a better learning-theoretic understanding of state-of-the-art models that include operations such as convolutions and shortcuts.** We agree that the reviewer is undoubtedly right that eventually, one also needs to handle normalization techniques or, for example, the multiplier strategy in Fixup (Zhang et al., ICLR'19), which eliminates the need for batch normalization. For the latter, capacity control would also require constraining the magnitudes of the additional multipliers. We suspect that, to some extent, this will be possible while preserving training stability, but we did not implement this.
>
> #### **(6) Importance of layer-wise Lipschitz constants**
> > It’s not clear to me why layer-wise Lipschitz constant is important intuitively, as we do not perform layer-wise optimization? Isn’t the Lipschitz constant over all parameters (i.e. dimensions in the loss landscape) more relevant?
>
> We want to clarify, that in our work, Lipschitz constants are with respect to the data for *fixed weights W*, e.g., $\Vert W x - W y \Vert \le L \Vert x - y \Vert$. Notably, in [Long & Sedghi, ICLR '20], the authors study Lipschitz continuity of a map from $\mathbb{R}^\text{number of parameters}$ to the hypothesis class. We assume that this is most likely the Lipschitz constant the reviewer refers to and it essentially also factorizes as the product of layer-wise Lipschitz constants wrt. the data; see Eq. (26) in the comparison to this work in our supplementary material. Of course, the product of these layer-wise Lipschitz constants is just the worst-case upper bound on the whole-network Lipschitz constant wrt. the data. Existing *lower bounds* on the Rademacher complexity, see [Bartlett et al., NeurIPS '17], show that this worst-case estimate cannot be avoided without further assumptions on the data. It can however be replaced with empirical estimates [Wei & Ma, NeurIPS '19], [Ledent et al., AAAI '21].

---

> > ### Comment · Reviewer_BA91 · 2022-08-09
> > **Rebuttal Response**
> >
> >
> > (4) I think this is all I was looking for as a reader. I think this is a question that will occur to any reader of the current text given you bring up the vacuous nature of most complexity bounds, and so you may want to consider addressing it in some form.
> >
> > (5) Yes, this is what I figured. I think this is the largest limitation of the current work, especially for a reader hoping to apply the analysis to their contemporary model, but there is nothing wrong with acknowledging this and marking it as something for future work and it certainly should not affect the acceptance of this work.
> >
> > (6) I see, yes I completely misunderstood this and appreciate this clarification. This is likely because of my research background and the more typical usage of Lipschitz constants in that area rather than your paper's explanation.

---

> ### Author Response · Authors · 2022-08-02
> **Response to Reviewer BA91 (1/3)**
>
> We like to thank reviewer **BA91** for the detailed comments and the time spent on this thorough review. Below, we can hopefully clarify any raised issues.
>
> #### **(1) Intuition behind capacity terms**
> >The intuition behind the capacity terms proposed was not obvious to me
>
> Our capacity terms are motivated directly by theory, i.e., from the derived Rademacher complexity bounds (Theorem 3.5). Overall, the two bounds both consist of several factors, corresponding to (i) sample size dependency, (ii) (weak) data dependency via the average norm of the samples, (iii) dependency on the number of parameters ($W$) and (iv) dependency on weight norms ($\tilde C$). For the same data distribution and sample size, the first capacity measure ($\clubsuit$) is mainly determined by the weight norms, whereas the number of parameters mainly determines the second one ($\spadesuit$).
> We would be more than happy to incorporate any suggestions on providing further intuition in the manuscript and which specific points need clarification. However, we also very much appreciate the positive sentiment provided by our previous comment "The paper is well written overall, with the motivation for the proposed capacity terms made clear, and appropriate background references".
>
> #### **(2) Computation of Lipschitz constants / Reproducibility**
>
> > There are some reproducibility issues in the experimental results presented, notably it’s not obvious to me how the Lipschitz constant is calculated for different layers (only the classifier is explicitly addressed by construction from what I can see).
>
> > I don’t see any details on how the Lipschitz constant for all layers in the results section (i.e. Table 1) is calculated empirically, making it difficult to repeat these results.
>
> We want to point the reviewer to our supplementary material, particularly the provided source code and Appendix B.2, which details the projection method and lists relevant prior work. Informally, after an update step during stochastic optimization, we (orthogonally) project the parameters (per layer) back onto the constraint set. To this end, a layer's singular values (including the Lipschitz constant) are computed via the method introduced in [Sedghi et al., ICLR '19]. At evaluation time, Lipschitz constants are computed via a power iteration method for convolutional layers [Gouk et al., Machine Learning '20], [Li et al., NeurIPS '19]. When making the code public, we will provide clean encapsulated layers to be used by the community to easily reproduce the results.
>
> #### **(3) Parameter counts and weight norms**
> > The authors point to the fact that compressibility w.r.t. parameter counts is well known, but claim that their experimental results show a novel result — that neural networks are compressible w.r.t. weight norms. This presumes that weight norm and parameter counts are independent (or “orthogonal”). However, the authors use weight decay in their experiments, i.e. an L2 weight norm regularization term, and the variance of the common random initialization methods for NN layers (i.e. Xavier/He) explicitly depends on the parameter count of a layer. Can we really assume that weight norms and parameter counts are independent in the context of real-world neural network training?
>
> Thank you for raising that point. We think the wording ''orthogonal'' was poorly chosen and will be adjusted. We did not mean to imply that the parameter count and weight norms are fully independent. To provide more insight, we ran a series of (additional) parameter pruning experiments (eliminating 10%, 20%, etc., of the smallest weights) for unconstrained and constrained models. Interestingly, these experiments indicate that constrained models are as compressible as unconstrained ones (in terms of #parameters). Overall, we see evidence that neither a large number of parameters nor large weight norms are necessary to obtain well generalizing models. *Notably, both can be reduced at the same time*. There is, however, one crucial difference: parameter pruning is typically done on the *final* model, whereas capacity reduction (via weight norms) is done during training and fails if done on the final model only (see Appendix, Fig. 6 - rightmost column).
>
> Also, our additional experiments with LARS/LAMB were run *without* any weight decay, yielding essentially the same results in the constrained regime; see our response to Reviewer vwiQ (Regarding Questions).

---

> > ### Comment · Reviewer_BA91 · 2022-08-09
> > **Rebuttal Response**
> >
> > I thank the authors for their clarifications and rebuttal. I'll proceed through each of your numbered rebuttal points and respond individually to them.
> >
> > (1) This is good, I think it's still unclear to me intuitively why weight norms should be related to capacity (except perhaps the presence of non-zero /approximately zero weights indicating the weights are not being used).
> >
> > (2) While making code public is great, I do believe this should be described somewhere in the paper/appendix (even if it's just, we calculated with the method introduced by [Sedghi et al., ICLR '19]).
> >
> > (3) The authors did address my concerns on the relationship between parameter count and weight norms, and I hope they can adjust the text to make this clear to future readers.

---

> > > ### Author Response · Authors · 2022-08-09
> > > **Thank you**
> > >
> > > We thank you for very thoughtful response. We are glad that the issues could be resolved.
> > > For a final version, we will update our manuscript to incorporate your comments and extend our preliminary experiments regarding point (8).

---

### Official Review · Reviewer_vwiQ · 2022-07-10

**Rating:** 6
**Confidence:** 3
**Soundness:** 3 good
**Presentation:** 4 excellent
**Contribution:** 3 good

**Summary:**

This paper provides new bounds for Rademacher complexity for neural networks and analyses their effect empirically. Most importantly, skip connect and convolutions are handled jointly, whereas normalisation techniques are not considered. A comparison to the existing literature is implemented. Exp[eriments try to emphasise how there is an excess in capacity and how it can be reduced. However, implementations seems to be cumbersome for now and not fully practical.

**Questions:**

Why can we apply Lipshitz constraints for h around line 215? I would have expected h to be unconstrained in normal deep learning.

How can Rademacher boundaries be established for the different normalisation techniques like group and batch normalisation, which are crucial parts of the algorithm. Is it possible at all to address them? If not, why? If yes, how?

Why is it not possible to run the experiments with larger networks and more importantly larger datasets?

Why are the experiments run for 200 epochs, even though even ResNet50 is able to converge in 35 epochs?

Can the limitation of norms for reducing the capacity also be achieved by different optimisers that consider layers separately like LARS or LAMB?

**Limitations:**

Complex networks, require normalisation techniques like batch norm or layer norm. These are not addressed by the paper, which is a major limitation. Whereas skip connections and convolutions are crucial for generating meaningful features and getting a large network depth, normalisation techniques are crucial for convergence.

Another limitation is that the reduced capacity in the experiments does not really have an impact on the model size and expressivity and seems more artificially related to the provided new Rademacher complexity bounds. They could be however more a sign of the insufficiency of the bounds and the benefits of regularization than the real power of the new bounds.

**Strengths And Weaknesses:**

Significance
************

The relation to the state of the art is well presented. Especially, the inclusion of convolutions and skip connections is making a difference as well as the paper addressing how the respective Rademacher Complexities can be actually used for real world decision like changing the capacity of networks.

I really like Figure 1 and how it is representing the state of the art. Also, the "Relation to prior work." sections are very valuable, because they put the theorems in relation to other work and discuss advantages and disadvantages. Thus providing some valuable kind of literature overview.

According to my understanding of ResNet18, it is possible to linearly scale layers without impacting the decision boundaries. Hence, it is unclear how limiting the weight norm would impact capacity at all. What are the benefits of this "compression" and why is it called "compression" at all?

Previous work addressed convolutions and skip connections separately and it seems none addressed them sufficiently jointly.

A major caveat Is the lack of addressing normalisation techniques in the network to have ResNet18 fully addressed.

I feel like disagreeing on the necessity of data dependent bounds. I belief, model complexity and data complexity are strongly related.


Clarity
*******


The paper is very well structured and the separate steps are clear. References to the appendix are provided where required.

Whereas I am not experienced in Rademacher complexity proofs, I feel that the proof sketch fro Theorem 3.2 and the whole motivation are described very clearly.

The word "vacuous" should be replaced with a more common and less archaic word.
The figure on page 1 is missing a legend for the read dots and it is missing a caption/title/number/reference.

Tiny-ImageNet-200 is not sufficiently referenced and explained in line 304.

The experiments state that every 15th SGD update step, the projections procedure is applied but then the authors "increase the alternating projection steps to 15". 15 is not an increase of 15.

Table 1 mentions "the testing error is on a par with the unconstrained case". However, the error is no the same. So I assume that it means "the testing error is equal or better than the unconstrained case"?

Some more technical details on the experiments would be great.

It might be useful to quickly mention what the "empirical generalization gap" is and why it is important to measure.

The experiments need further clarification.

Originality
***********

The paper points out quite clear the relation to existing state of the art and how its boundaries are better than theirs. This is a great contribution in itself.

The derived boundaries seem to be non-trivial, especially since previous literature resulted in higher bounds.

It is great effort of the authors, not being satisfied with some new bounds but implementing a comparison to related work as well as addressing the respective values in empirical evaluations.


Quality
*******


The first figure needs some fixing. Overall, the paper is well structured and readable.
The paper states "Notably, modifications (2) and (3) do not harm performance, with
302 empirical testing errors on a par with a standard ResNet18 w/o batch normalization.", however, respective references are missing. Also, it is not clear, how much the quality of ResNet18 is reduced due to the lack of normalisation. This part needs much more discussion.

---

> ### Author Response · Authors · 2022-08-02
> **Reviewer vwiQ (4/4)**
>
> ## $\blacktriangleright$ **Regarding Limitations**
>
> > Another limitation is that the reduced capacity in the experiments does not really have an impact on the model size and expressivity and seems more artificially related to the provided new Rademacher complexity bounds. They could be however more a sign of the insufficiency of the bounds and the benefits of regularization than the real power of the new bounds.
>
> We agree that model size (in terms of #parameters) is not affected by our constraints, but expressivity is. For sufficiently strong constraints, models are unable to fit to the training data, which suggests a reduced expressivity. We also want to point toward our earlier comment on weight scaling.
>
> Furthermore, controlling Lipschitz constants *and* the distance to initialization could indeed be considered a form of *regularization*, which is motivated by our bound-driving terms. Conceptually similar techniques have been used before, but only in isolation, see, e.g., [Miyato et al., ICLR'18] or [Gouk et al., ICLR '21]. If both quantities are controlled *jointly*, as suggested by our bound, we see an apparent effect on the difference between training and testing error (i.e., the empirical generalization gap). In contrast, just enforcing a Lipschitz constraint of 0.8 on CIFAR100 (with no constraint on the distance to initialization) yields an empirical generalization gap of 43 percentage points. Similarly, only enforcing a distance to initialization constraint of 70 (with no Lipschitz constraint) gives a gap of 44 percentage points. As seen in Table 1, controlling *both* constraints reduces this gap to 16 percentage points without deteriorating the testing error. We argue that this is strong evidence that norm-based Rademacher complexity bounds are quite informative, despite being insufficient to provide acceptable sample size guidelines. Yet, this is more of a deficiency related to having to account for the worst case in classic uniform convergence type analyses.

---

> ### Author Response · Authors · 2022-08-02
> **Response to Reviewer vwiQ (3/4)**
>
> ## $\blacktriangleright$ **Regarding Questions**
>
> > Why can we apply Lipschitz constraints for h around line 215? I would have expected h to be unconstrained in normal deep learning.
>
> Yes, in ''normal'' deep learning $h$ would be unconstrained. At this point (`l215`) in the paper, introducing the Lipschitz constraint is a mere technicality. However, later in Section 4, we do need the results derived in this regime, as we specifically run experiments with hypothesis classes that are a priori constrained wrt. Lipschitz constants. A subsequent union bound argument (e.g., Lemma A.9 in [Bartlett et al., NeurIPS '17]) over varying constraint strengths implies generalization bounds for the unconstrained setting.
>
> ---
> > How can Rademacher boundaries be established for the different normalisation techniques like group and batch normalisation, which are crucial parts of the algorithm. Is it possible at all to address them? If not, why? If yes, how?
>
> This is an excellent question without a satisfactory answer (yet):
> at evaluation time, batch normalization is an affine map, parametrized by the running statistics. Rademacher complexity bounds can then be established under the assumption of Lipschitz and norm constraints. As discussed above, when deriving generalization bounds, these assumptions can be considered a technical tool and be removed eventually.
>
> Nevertheless, when studying the excess capacity of networks with normalization layers, we would have to impose such constraints on them, potentially removing their ability to normalize the inputs/representations properly. As this defeats the very purpose of these layers, we decided to omit them for our empirical evaluation. We hope to properly account for these techniques in future work.
>
> ---
> > Why is it not possible to run the experiments with larger networks and more importantly larger datasets?
>
> There is no conceptual limitation to running our experiments on larger datasets or larger models. We chose CIFAR10, CIFAR100, and TinyImageNet-200 as these three datasets cover the regime of increasing dataset complexity (which we defined as an increase in the number of classes).
>
> ---
> > Why are the experiments run for 200 epochs, even though even ResNet50 is able to converge in 35 epochs?
>
> Minimizing the loss under the restriction that our constraints are met is a more challenging optimization problem that converges more slowly. For consistency, we ran all experiments with the same hyperparameters. We found that optimizing over 200 epochs ensured convergence even for the strongest constraints we evaluated. Unconstrained models indeed converge much faster, but (importantly) the testing error does not deteriorate in case of more update steps. Equally, the Lipschitz constants and the distance to initialization do not continue to increase much once the close-to-zero training error regime is reached (which happens way before 200 epochs).
>
> ---
> > Can the limitation of norms for reducing the capacity also be achieved by different optimisers that consider layers separately like LARS or LAMB?
>
> To answer that question, we ran additional experiments with LAMB *and* LARS but did not observe any noticeable differences. The norm constraints were still met at the end of training, and model performance was comparable. E.g., on CIFAR10 at 0.8 layer-wise Lipschitz constraints and a distance to initialization constraint of 50 (i.e., the operating point in Table 1), LARS and LAMB yield models with training/testing errors comparable to the models trained with SGD. That is, for LARS, the testing accuracy was $78.6 \pm .1$ and training accuracy was $95.9 \pm .1$. For LAMB, $78.6 \pm .3$ resp. $95.8 \pm .3$. We will add a remark in a final version of the manuscript.

---

> ### Author Response · Authors · 2022-08-02
> **Response to Reviewer vwiQ (2/4)**
>
> ## $\blacktriangleright$ **Regarding Clarity**
>
> We thank you for your suggestions to improve clarity. We will incorporate them in a final version of the manuscript.
>
> >The experiments state that every 15th SGD update step, the projections procedure is applied but then the authors "increase the alternating projection steps to 15". 15 is not an increase of 15.
>
> Here, the 15 steps refer to two different quantities; one refers to the frequency of how often we project onto the contraint sets, the other corresponds to the quality of such projections. Choosing the value of 15 for both was certainly suboptimal from a clarity perspective, but cannot be changed retrospectively. We will indicate the different nature of these quantities more clearly in a final version. Below, we explain the nature of both quantities in more detail.
>
> Recall from the last paragraph on page 7, that during training we want to project onto the constraint set $\mathcal C = \mathcal C_1 \cap \mathcal C_2$. This is the intersection of the sets $\mathcal C_1$ and $\mathcal C_2$, which correspond to the distance and Lipschitz constraints, respectively. We consider the operation of first projecting a weight onto $\mathcal C_1$ and then onto $\mathcal C_2$ as one "*alternating projection step*". A weight, on which repeated applications of such alternating projection steps are performed, converges to a point in the intersection $\mathcal C$.
> For efficiency, we do not ensure that the weights satisfy the constraints after every SGD update and make the following adaptions.
>
>   - We project only after every 15th *SGD update*.
>
>   - Even after 15 SGD steps, we do not necessarily project sufficiently often, so that the weights satisfy the constraints. Instead, we perform only *one alternating projection step* (project once on $\mathcal C_1$ and once on $\mathcal C_2$).
>
> Because of the decrease of gradient norms and learning rates during training, this suffices that the final model (almost) satisfies the constraints. To ensure that the constraints are actually satisfied, we perform additional projections after training. These are the mentioned 15 alternating projection steps. That is, we first project the weights onto $\mathcal C_1$ and then on $\mathcal C_2$, and repeat this procedure 15 times. We will clarify this in a final version. We also like to point the reviewer toward our supplementary material (B.2), where we provide further details on the projection method.
>
> ---
> >Table 1 mentions "the testing error is on a par with the unconstrained case". However, the error is no the same. So I assume that it means "the testing error is equal or better than the unconstrained case"?
>
> This is correct.
>
> ---
> > Some more technical details on the experiments would be great.
>
> > The experiments need further clarification.
>
> We address your specific question below ($\blacktriangleright$ Questions). If further clarifications are needed, we would be more than happy to provide them.
>
> ## $\blacktriangleright$ **Regarding Quality**
>
> > The paper states "Notably, modifications (2) and (3) do not harm performance, with 302 empirical testing errors on a par with a standard ResNet18 w/o batch normalization.", however, respective references are missing.
>
> We would like to point to the beginning of the corresponding paragraph, which includes relevant references regarding modifications (2) and (3). Further, we empirically verified, that the modifications do not harm performance, i.e., models achieved the same testing accuracy irrespective if they were modified via (2) and (3).
>
> ---
> > Also, it is not clear, how much the quality of ResNet18 is reduced due to the lack of normalisation. This part needs much more discussion.
>
> To address this point, we ran additional experiments with a pre-activation ResNet18 with batch normalisation but without data augmentation (as done in our experiments). These models achieved a testing accuracy of $\sim$ 89%. So from a performance perspective, the contribution of normalisation techniques cannot be ignored. Nevertheless, we are convinced, that running experiments without batch normalisation is appropriate, as we reason in the comment addressing your second question (see below).

---

> ### Author Response · Authors · 2022-08-02
> **Response to Reviewer vwiQ (1/4)**
>
> We like to thank reviewer **vwiQ** for the detailed comments and the time spent on this thorough review. Below, we can hopefully clarify any raised issues.
>
> ## $\blacktriangleright$ **Regarding Significance**
>
> > According to my understanding of ResNet18, it is possible to linearly scale layers without impacting the decision boundaries. Hence, it is unclear how limiting the weight norm would impact capacity at all. What are the benefits of this "compression" and why is it called "compression" at all?
>
> First, it is correct that there are architectures for which you can linearly scale the layers without impacting decision boundaries, e.g., homogeneous networks.
> However, we like to clarify that our capacity measure considers the magnitude of weight norms *relative* to the classification margin $\gamma$ (see the definition of $\tilde C_{ij} = 2C_{ij}/\gamma$ in Theorem 3.5). Thus, for homogeneous networks, such a linear scaling would *not* lead to any change in hypothesis class capacity. Our notion of "compression" can be understood as reducing the layer-wise Lipschitz constants while *at the same time* retaining the classification margin. This is different from the aforementioned linear scaling of weights: our experiments confirm that this notion of compression can actually be achieved, as the Lipschitz constants can be reduced to a much larger extent than the margin.
> For example, for the models trained on CIFAR-10, the product of Lipschitz constants can be reduced from $4.4\cdot 10^5$ (unconstrained) to $2.1  \cdot 10^2$ (operating point), while the mean of the margin distribution only reduces from $15.2$ (unconstrained) to $3.2$ (operating point). Similarly, to achieve a ramp loss $\hat R_\gamma < 0.1$ on CIFAR-10, we can choose $\gamma= 11.4$ for the unconstrained model and $\gamma=0.8$ for the constrained one. For CIFAR-100 this effect is even more pronounced, with a reduction of the product of Lipschitz constants from $2.5\cdot 10^7$ to $2.1  \cdot 10^2$ and of the average margin from $13.5$ to $0.7$.
>
> Addressing your remark regarding *residual networks*. For them, the property of decision boundary invariance wrt uniform layer scaling is due to batch normalization, however, only during training. During evaluation, batch normalization is just an affine map, and the argument from above applies. Handling the map to which batch normalization corresponds to during training, i.e., a non-parametrized map that simply normalizes each batch to zero mean and unit variance, is more challenging and can (currently) not be handled by our approach (as it is a fixed, but non-Lipschitz map). The decision boundaries of the modified architecture we used in experiments (no batch normalization) are actually affected by weight scaling; such scaling would change the "distance" of each residual block to the identity map.
>
> We emphasize that our primary goal was to analyze (excess) capacity for architectures close to what is used in practice. With convolutions and skip connections handled, we believe this is the right direction. Still, as pointed out by the reviewer, normalization techniques need to be addressed due to their abundance in state-of-the-art networks.
>
> ---
> > I feel like disagreeing on the necessity of data dependent bounds. I belief, model complexity and data complexity are strongly related.
>
> We agree with your comment on the strong relationship between data and model complexity. In fact, we would argue that our experiments highlight this relation. While Fig. 1 (unconstrained ResNet18's) already hints toward such a relationship, it is inconclusive to some extent, as it does not show whether the increase in the weight norms is actually required to handle the more complex datasets. Our experiments with constrained models address this deficiency. At first sight, Table 1 suggests that the strength of possible constraints is not strongly coupled to dataset complexity, but Table 1 only focuses on *one* combination of constraints, i.e., the identified *operating point*. Taking the whole range of constraint strengths into account in Fig. 2 (bottom) reveals more information: with decreasing data complexity, there is a broader range of stronger constraints within which the testing error only marginally deteriorates.

---

### Official Review · Reviewer_RqdG · 2022-07-10

**Rating:** 7
**Confidence:** 4
**Soundness:** 4 excellent
**Presentation:** 3 good
**Contribution:** 3 good

**Summary:**

This paper proves generalization bounds for both convolutional and ResNet networks. In both cases, both the parameter counting approach and the norm-based approach are explored.

CNN-Norm based (Theorem 3.5, first equation): Similar to [1] expect that it relies on L2 norms instead of L\infty norms and therefore scales slightly differently depending on the regime. The main aspects are that the dependence on the input to each layer is larger than in [1] since it involves the whole of the input rather than a single convolutional patch, but on the other hand there is a factor of the post pooling width from [1] which disappears. The two could compensate each other or be in favour of one or the other bound. The norm in the batch direction is L2 instead of L\infty as well, which is an advantage in the presence of outliers, but makes the dependence on the number of classes worse.

CNN-Parameter counting (Theorem 3.5, second equation): Similar to [3], better management of constants (it becomes explicit), cleaner proof. Other claimed improvements are not significant. Just like [2] the bound basically says that the sample complexity grows as the number of parameters.


Resent- Parameter counting: Similar to [3] in terms of the approach, but solves a different problem and offers some of the first bounds on resnets. Neat exercise but no significantly original content. Just like [2] the bound basically says that the sample complexity grows as the number of parameters.

Resnet- Norm based: similar extension of [1] but still a very long and tedious exercise (due to the larger number of connections in the architecture) which was successfully executed.











[1] Antoine Ledent, Waleed Mustafa, Yunwen Lei and Marius Kloft. " Norm-based Generalisation bounds for convolutional neural networks"

[2] Peter Bartlett, Dylan J. Foster, Matus Telgarsky. "Spectrally-normalized margin bounds for neural networks". NeurIPS 2017.

[3] Philip Long and Hanie Sedghi. "Generalisation bounds for deep convolutional neural networks". ICLR 2020.

**Questions:**



**Limitations:**

Originality, though I don't think this should be viewed too negatively as the community also needs papers like this one, especially if the claims of originality are further toned down.

**Strengths And Weaknesses:**


Strengths:

++This is a long paper with a lot of work and is quite clean and well organised. It contains a good encyclopaedic summary of the field and of many frequently misunderstood issues.

+ A lot of progress since previous submission

Weaknesses:

- No substantial originality in the techniques.
- Some issues not corrected since last submission.



NUMERICAL CONSTANTS:

There is some slight dishonesty going on, including an error, in the treatment of numerical constants here: the authors results involve a constant of 12, which looks better than the corresponding constant of 36 in Bartlett et al [2]. This is only partially true.

First of all, there is an error in the constant in [2], and it should be 72 instead of 36 (the authors correctly divide the constant of 72 from [2] in Lemma A.8 because they are looking at Rademacher complexity without explicitly writing the results in terms of generalisation bounds. However, the correct version of Lemma A8 should be with a constant of 144. Indeed, the function class which is studied in Lemma 3.2 in [2] includes only a (2,1) norm constraint, but no constraint on the spectral norm. In the proof of Theorem 3.3, a cover on a function class including both constraints is used instead (such a cover can be obtained up to a factor of 2 by a simple extra argument).  This is a very subtle error, which is inherited in the present paper (not in [1]), so a priori anyone could be forgiven for overseeing this. However, I reviewed this paper for ICML this year, pointed out the error to the authors, and they agreed with me in the rebuttal and agreed to fix it, which they haven't done!


Next, the improvement in the constant compared to [2] (and to a lesser extent to [1]) is partially illusory. Note that there is a factor of 2 in both [2] and [1] which comes from the 2-Lipschitzness of margin-based loss functions. Here the authors do not work with a last layer loss function and work directly with the Rademacher complexity of the function class defined by the network, so the improvement in the constant is just due to solving a slightly different problem.

Regarding the comparison to [1] in terms of numerical constants, it is correctly pointed out in line 654 (sup) that some may be unavoidable but that the bound from [4], where, according to the current paper "the constants are nor optimized". This is quite a bold statement. The results in [4] are very different and much much harder to prove that the Maurey type results used here and in [2], since they rely on L\infty norms. There is a good chance the constants are unavoidable if one goes for the L\infty approach.


IMPROVEMENT COMPARED TO [3]


I guess we just have to agree to disagree here but again, the improvement in the parameter counting result compared to [3] is purely in terms of the proof technique and clarity of exposition (and maybe slightly in the constant). Both [3] and the present paper somewhat misleadingly claim that the results "take distance to initialization into account", with the present paper also claiming that they do so "more naturally" (see line 264), but fundamentally, in both bounds, the square root of the number of parameters is both conceptually and numerically the dominant term, and distance to initialization is only present in the logarithmic factors. The authors explain this quite clearly in other parts of the paper, so I think they understand the point.



TWO LAST POINTS REGARDING COMPARISON TO [1]

I would like to see a comparison with the two layer version of [1]. Indeed, the factor which comes from changing between norms and scales similarly in [1] and the present paper depending on the regime is intuitively parasitic (though it seems difficult to get rid of it entirely and neither the current paper nor [1] succeed). However, in the two layer case, it is absent from [1]. I would expect the bound from [1] to be uniformly superior in this case (of course since it only applies to 2 layers it doesn't make the present paper irrelevant in general).
Note also that the parasitic  factor can be reasonably removed from [1] in the general case at the cost of extreme cost in terms of computations (rather than loss of statistical power) by simply certifying the Lipschitz constant of the network with respect the the appropriate norms in a sufficiently large neighbourhood of all inputs.









SMALL TYPOS

Line 1089, capital letter for "For"
 Line 213 "we further fixate..." --> "we further fix"






[1] Antoine Ledent, Waleed Mustafa, Yunwen Lei and Marius Kloft. " Norm-based Generalisation bounds for convolutional neural networks"

[2] Peter Bartlett, Dylan J. Foster, Matus Telgarsky. "Spectrally-normalized margin bounds for neural networks". NeurIPS 2017.

[3] Philip Long and Hanie Sedghi. "Generalisation bounds for deep convolutional neural networks". ICLR 2020.


[4] Tony Zhang, "Covering number bounds of certain regularised linear function classes".

---

> ### Author Response · Authors · 2022-08-02
> **Response to Reviewer RqdG (3/3)**
>
> **Empirically**, we computed the bounds for networks of varying width $c\in \{32, 1024, 8192\}$ trained on CIFAR-100. Here, we compute exact values and do not use the simplifications (1) - (7). As all models did not fit to the training data, we used $\gamma=1$ for the margin parameter for simplicity. Overall, these models performed rather poorly with testing accuracies 21%, 29%, 31% and training accuracies 32%, 99.8%, 99.9%.
>
>   The following tables list the computed values and how they compare over the respective factors on a logarithmic scale with base $10$ (the sample size dependency is negative, as $\log_{10}(\frac{\log(n)}{\sqrt{n}})<0$).
>
>   | Width = 32 | Ours ($\clubsuit$) $\small{(\log_{10}[~~])}$  | Ledent et al. $\small{(\log_{10}[~~])}$ |
>   |---|---|---|
>   |Bound                 | ${5.7}$ |  ${6.7}$
>   |Weight & Data norms   | ${4.8}$ |  ${4.3}$
>   |$\to$ 1st layer       | ${4.3}$ |  ${4.2}$
>   |$\to$ 2nd layer       | ${4.3}$ |  ${3.0}$
>   |Logarithmic term      | ${0.5}$ |  ${0.8}$
>   |Numerical constant    | ${1.7}$ |  ${2.9}$
>   |Sample size dependency| ${-1.3}$|  ${-1.3}$
>
> | Width = 1024  | Ours ($\clubsuit$) $\small{(\log_{10}[~~])}$  | Ledent et al. $\small{(\log_{10}[~~])}$ |
>   |---|---|---|
>   |Bound                 | ${6.5}$ |  ${8.0}$
>   |Weight & Data norms   | ${5.6}$ |  ${5.5}$
>   |$\to$ 1st layer       | ${5.4}$ |  ${5.5}$
>   |$\to$ 2nd layer       | ${4.7}$ |  ${3.6}$
>   |Logarithmic term      | ${0.6}$ |  ${0.9}$
>   |Numerical constant    | ${1.7}$ |  ${2.9}$
>   |Sample size dependency| ${-1.3}$|  ${-1.3}$
>
> | Width = 8192  | Ours ($\clubsuit$) $\small{(\log_{10}[~~])}$  | Ledent et al. $\small{(\log_{10}[~~])}$ |
>   |---|---|---|
>   |Bound                 | ${7.1}$ |  ${8.6}$
>   |Weight & Data norms   | ${6.1}$ |  ${6.2}$
>   |$\to$ 1st layer       | ${6.0}$ |  ${6.2}$
>   |$\to$ 2nd layer       | ${4.6}$ |  ${3.7}$
>   |Logarithmic term      | ${0.6}$ |  ${0.9}$
>   |Numerical constant    | ${1.7}$ |  ${2.9}$
>   |Sample size dependency| ${-1.3}$|  ${-1.3}$
>
>   *Overall*, we observe the following effects:
>   1. Relatively, the contribution of the second layer in [Ledent et al., AAAI '21] is improved by a factor of 10. This is expected, as 10 is the square root of the number of classes.
>   2. The wider the network, the more dominant the term corresponding to the first layer becomes. At width 32, the factor from weight and data norms of [Ledent et al., AAAI '21] is clearly superior. This is due to the improved class dependency. However, for a width of 1024, this effect is already negligible.
>   3. For the wider networks, we improve over [Ledent et al., AAAI '21] by a factor of $\sim 10^{1.5} \approx 30$. Ignoring numerical constants, we improve by a factor $\sim 10^{0.3} \approx 2$, which is due to an improvement in the logarithmic term.
>
> Last, we consider a network whose first layer has kernel size 3 and stride 1. Reduction of the spatial dimensionality is achieved by a subsequent max pooling layer of window size $32 \times 32$. This setting favors [Ledent et al., AAAI '21] as they can better account for the pooling layer; our result ($\clubsuit$) loses the factor $1/k$ in the first summand in the simplified bound from above ($k$ is now 1); their result improves due to the now smaller convolutional patches.
>
> Width = 1024 (pooling)  | Ours ($\clubsuit$) $\small{(\log_{10}[~~])}$  | Ledent et al. $\small{(\log_{10}[~~])}$ |
>   |---|---|---|
>   |Bound                 | ${7.0}$ |  ${7.3}$
>   |Weight & Data norms   | ${6.1}$ |  ${4.8}$
>   |$\to$ 1st layer       | ${5.8}$ |  ${4.1}$
>   |$\to$ 2nd layer       | ${5.4}$ |  ${4.6}$
>   |Logarithmic term      | ${0.5}$ |  ${0.9}$
>   |Numerical constant    | ${1.7}$ |  ${2.9}$
>   |Sample size dependency| ${-1.3}$|  ${-1.3}$
>
> We see that in this setting the contribution of weight and data norms is clearly larger in our bound, due to the first layer. Yet, our bound is still smaller, but only due to *numerical constants*.

---

> ### Author Response · Authors · 2022-08-02
> **Response to Reviewer RqdG (2/3)**
>
> ### $\blacktriangleright$ **Comparison to [Ledent et al., AAAI '21]**:
>
> Just as the reviewer, we expect the result of [Ledent et al., AAAI '21] to be favorable in the case of two-layer networks. To some extent, we discuss this in the supplementary material (A.5), particularly at point (4) *Dependency on number of classes*. Both single-layer covering number bounds can be decomposed into 4 factors, corresponding to (i) data norms, (ii) weight norms, (iii) logarithmic terms, and (iv) numerical constants. Overall, we conclude that the factors from data and weight norms are approximately equal, whereas we improve in the logarithmic terms and numerical constants. However, the $l_\infty$ covering approach in [Ledent et al., AAAI '21] allows for favorable treatment of the last (classification) layer. Its covering number bound only depends on the Frobenius norm of its weight instead of the (2,1) group norm. This yields an implicit improvement by a factor $\sqrt{\text{number of classes}}$. If the contribution of this last layer to the respective Rademacher complexity bounds is substantial, then the bound of [Ledent et al., AAAI '21] should be superior. This effect is most pronounced for two-layer networks on tasks with a large number of classes.
>
> **Detailed two-layer comparison**. As requested, we provide an empirical and theoretical comparison of the two-layer case in the following setting: the first layer is a convolutional layer, parametrized by a tensor $K\in \mathbb R^{c \times 3 \times k \times k}$ with $c$ denoting the number of filters (output channels). The kernel sizes $k$ and strides are equal to the spatial dimensionality of the input, e.g., 32 for images from CIFAR100, so that the spatial dimensionality of the output is already 1. This is followed by an activation function with Lipschitz constant 1 (e.g., ReLU & flatten) and a linear map parametrized by a matrix $W\in \mathbb R^{m\times c}$, with $m$ denoting the number of classes.
>
> As the quantities/norms appearing in the respective bounds differ, we make the following simplifications, which are motivated by corresponding inequalities and we verified empirically.
>
>   (1) $\Vert K \Vert_{2,1} \approx k \sum_{i=1}^c \Vert K_{i\cdot\cdot\cdot}\Vert$,
>
>   (2) $\Vert W^T \Vert_{2,1} \approx \sqrt{c} \Vert W\Vert_2$,
>
>   (3) $\max_{k\le m} \Vert W_{k\cdot}\Vert_2 \approx \operatorname{Lip}(W)$,
>
>   (4) $\frac{\|X\|}{\sqrt n} \approx \max_{i\le n} \|x_i\|_2 =$ maximal norm of convolution patches from the data $= |X|_0$,
>
>   (5) $H_{n-1}/n \approx \log(n)/n$.
>
> Furthermore, just as the single-layer bound in [Ledent et al., AAAI '21] depends on the maximal norm of a patch of the data, ours actually depends only on the maximal norm of particular slices of the data, which we here denote as $|X|_s$. In the special case of the stride being equal to the spatial dimensionality, these slices are over the channels at fixed local position (see Remark C.9 and the last chain of inequalities in the proof of Theorem C.7). Thus, for the contribution of the first layer, we can use
>
>   (6) $|X|_s \approx \frac{\|X\|}{k}$.
>
> Last, we empirically verified that
>
>   (7)
>   $\frac 1 {\max_{i\le n} \Vert \phi_K(x_i) \Vert_2}
>   \le \frac{\max_{k\le m} \Vert W_{k\cdot}\Vert_2 }{\gamma}\enspace,
>   $
>
> The last point (7) is relevant for the maximum operator appearing in the quantity $\mathcal R$ in Theorem 2 of [Ledent et al., AAAI '21].
>
> With these simplifications, **our** bound ($\clubsuit$) becomes
>
> $$\frac {48} {\gamma}  \frac{\log n}{\sqrt {n}}\frac{\lVert X\rVert }{\sqrt n}\sqrt{\log(6 c k^2)}\bigg[\bigg(\frac{\operatorname{Lip}(W) \lVert K - K^{(0)} \rVert _{2,1}}{k}\bigg)^{2/3}+ \bigg(\operatorname{Lip}(\phi_K) \lVert(W - W^{(0)})^\top\rVert _{2,1} \bigg)^{2/3}\bigg]^{3/2}$$
>
>   and the bound from Theorem 2 in [Ledent et al., AAAI '21] becomes
> $$\frac{768}{\gamma}
> \frac{\log n}{\sqrt n}
> \sqrt{\log_2\bigg(n^2 \mathcal D\bigg)} \bigg[\bigg(\frac{\operatorname{Lip}(W) \lVert K - K^{(0)}\rVert _{2,1}}{k}\bigg)^{2/3} + \bigg(\frac{\operatorname{Lip}(\phi_K) \lVert (W - W^{(0)})^T\rVert _{2,1}}{\sqrt{m}}\bigg)^{2/3}\bigg]^{3/2}$$
>
>   with the argument of the logarithm being
>   $$ \mathcal D = \max\bigg(
>       \frac{\Vert K-K^{(0)}\Vert_{2,1}}{k} \frac {\operatorname{Lip}(W)}{\operatorname{Lip}(K)} c,
>     ~
>     \frac{\Vert X\Vert}{\sqrt n}
>     \Vert W-W^{(0)}\Vert_2
>     \frac{\operatorname{Lip}(K)}{\gamma}  m\bigg)\enspace.$$
>
>   As expected, ignoring constants and log terms, the bound from [Ledent et al., AAAI '21] is better by a factor of $\sqrt m$ (square root of number of classes) in the summand corresponding to the last layer.

---

> ### Author Response · Authors · 2022-08-02
> **Response to Reviewer RqdG (1/3)**
>
> We like to thank reviewer **RqdG** for the detailed comments and the time spent on this thorough review. Below, we can hopefully clarify any raised issues.
>
> ### $\blacktriangleright$ **Regarding numerical constants:**
>
> We agree that the result in our earlier submission of this manuscript missed a factor of 2. **We corrected this error in the current submission.** In particular, Eq. 14 (i.e., the definition of $C_{ij}(X)$), now includes this factor. The supplementary material (on page 37, `l1003`) contains the corresponding argument, where we switch from internal to external covering numbers.
>
> Not transferring this correction to the result of [Bartlett et al., NeurIPS '17] (which we list in Table 3 and to which the correction also applies, as pointed out by the reviewer) was an honest mistake and will obviously be corrected.
>
>
> > Next, the improvement in the constant compared to [2] (and to a lesser extent to [1]) is partially illusory. Note that there is a factor of 2 in both [2] and [1] which comes from the 2-Lipschitzness of margin-based loss functions. Here the authors do not work with a last layer loss function and work directly with the Rademacher complexity of the function class defined by the network, so the improvement in the constant is just due to solving a slightly different problem.
>
> First, we like to clarify that we never claim an improvement in constants wrt. [Bartlett et al., NeurIPS '17]. In fact, our discussion in `l141-l144` highlights that we actually recover the same single-layer covering number bound for fully-connected layers when specifying our Theorem 2 to spatial inputs of size 1 and kernel size 1.
>
> Second, for the Rademacher complexity bound in our Theorem 3.5, we actually do incorporate the last layer with the last layer loss function (i.e., ramp loss) whose Lipschitz constant $\frac{2}{\gamma}$ enters via $\tilde C_{ij}=\frac{2}{\gamma}C_{ij}$. We will add a remark to point this out more clearly. Taking into account the constants from the definition of $\tilde{C}_{ij}$ and the additional factor of 2 from our earlier comment, the Rademacher complexity bound in Theorem 3.5 (second summand) has a leading factor of 48 vs. 72 from [Bartlett et al., NeurIPS '17]. This difference (a factor of $3/2$) originates from the proof of Lemma A.8 in [Bartlett et al., NeurIPS '17], where the minimizer $\alpha$ for the Dudley entropy integral argument is chosen as $1/n$, yielding a factor of $\log(n^{3/2}) = \frac{3}{2}\log(n)$. In principle, choosing $\alpha=1/\sqrt{n}$ would be equally possible, which then yields a Rademacher complexity bound scaling as $\frac 4 n + 48 \mathcal\ O(\frac{\log n }{\sqrt{n}})$, just as ours.
>
> We further want to clarify our comment regarding ''non-optimized'' constants in [Zhang, JMLR '02]. In no way did we intend to criticize any of the contributions in the reference or in [Ledent et al., AAAI '21] for that matter. On the contrary, we wanted to highlight that the difference in numerical constants between our result and [Ledent et al., AAAI '21] might actually be less pronounced than what appears at first sight. In fact, it *might* be possible to obtain bounds with similar numerical constants, despite the different approaches regarding norms. We apologize for the poor wording and will adjust it accordingly. In essence, our ''non-optimized constants'' remark reflects the discussion in the original work of Zhang (following the proof of Theorem 4 on page 538), which states ''Note that we have made no attempt to optimize the constants in the proof of Theorem 4''.
>
> ### $\blacktriangleright$ **Comparison to [Long & Sedghi, ICLR '20]**
>
> Concerning your comment on the differences to [Long & Sedghi, ICLR '20], we agree that in the final result, the $\sqrt{\text{number of parameters}}$ term dominates in both bounds and that the distance to initialization enters only in the logarithmic factors in our work *and* theirs.
>
> Yet, we are still convinced that this dependency (in the logarithmic factor) enters more naturally in our result. In fact, any bound which depends on weight $\|K\|$ norms can incorporate the distance to initialization as $\|K\|\le \|K- K_0\| + \|K_0\|$, however at the cost of the "looseness" in the triangle inequality. [Long & Sedghi, ICLR '20] utilize exactly this very strategy; compare the proof of Lemma 3.2 in the reference. On the contrary, when proving our result, we use the translation invariance of covering numbers (as also done in [Ledent et al., AAAI '21], [Bartlett et al., NeurIPS '17], ...) to replace a function class centered at $0$ with function classes centered at the initialization. This allows to replace constraints on weight norms $\| K\|$ with constraints on $\|K - K_0 \|$ *without* introducing any additional looseness in the bound.

---

### Meta-Review · Area_Chair_uZK6 · 2022-08-29

**Recommendation:** Accept
**Confidence:** Less certain

**Metareview:**

This paper extends prior Rademacher complexity bounds to neural network architectures that include convolutions and skip connections. The paper is well-organized with a clear summary of the field and related work. The reviewers found that, although the techniques were not novel, the new results enhance the practical relevance of prior bounds and will be of interest to the theory community. A few technical points were raised and addressed during the discussion phase, but the consensus remains that this is a solid paper and should be published, and so I recommend acceptance.

**Award:**

No

---

### Decision · Program_Chairs · 2022-09-14

Accept